# Regularized Discriminative Alignment for Deep Representations under Label Shift

Hengchao Shi[* 1]  Boen Jiang[* 1]  Guanhua Fang[1]  Wen Yu[1]  Ming Zheng[1]

## Abstract

Label shift refers to the distribution shift scenario where the marginal label distribution changes while the class-conditional distribution remains invariant. To address this challenge in complex real-world settings, we propose **Regularized Discriminative Alignment for Label Shift (RDALS)**, a novel framework that adapts to target domains by aligning distributions within the deep latent space. By shifting the focus from raw inputs to learned representations, RDALS effectively operates under a weaker and more practical invariance assumption. Specifically, we construct a moment-matching linear system using Linear Discriminant Analysis (LDA) and show that this choice maximizes numerical stability. We further provide rigorous theoretical analysis, establishing finite-sample error bounds for the importance weight estimation and the generalization bounds for the adapted classifier. Extensive experiments on standard benchmarks demonstrate that RDALS significantly outperforms state-of-the-art baselines, achieving superior robustness and accuracy in both data-scarce and extreme-shift regimes.

## 1. Introduction

### 1.1. Background

The remarkable success of modern deep learning across various supervised learning tasks is often contingent on the fundamental assumption that training and test data are independent and identically distributed (i.i.d.). However, in real-world applications, this assumption is frequently violated due to the distribution shift (Quionero-Candela et al., 2009; Moreno-Torres et al., 2012; Ovadia et al., 2019; Smith & Adams, 2025), where the target distribution differs from the source distribution, leading to significant performance degradation of predictive models. Among various types of shift, label shift (also known as prior probability shift) (Saerens et al., 2002; Storkey, 2009; Schölkopf et al., 2012) is a prevalent scenario where the marginal distribution of class labels $y$ changes across domains, while the class-conditional distribution of features $x$ remains invariant. Specifically, denote the source and target marginal distribution of class labels by $P_y$ and $Q_y$, and the source and target class-conditional distribution of features by $P(x|y)$ and $Q(x|y)$, respectively. The label shift mentioned above means that $P_y \neq Q_y$ while $P(x|y) = Q(x|y)$. This setting is ubiquitous in real-world applications across diverse domains, such as medical diagnosis(Zhang et al., 2013; Tasche, 2017; Roschewitz et al., 2025), financial fraud detection (Lucas et al., 2019; Zhu et al., 2024), and ecological monitoring (Norouzzadeh et al., 2018; Oliver et al., 2023).

### 1.2. Prior Work

To address label shift, existing literature generally explores two primary paradigms. The first one, feature distribution matching (Gretton et al., 2012; Long et al., 2015), aligns feature densities directly, using techniques such as kernel mean matching (KMM) (Zhang et al., 2013) or generative adversarial training (LTF) (Guo et al., 2020). Due to the high computational cost of these approaches, the second paradigm, importance re-weighting (Garg et al., 2020; Maity et al., 2022; Roberts et al., 2022), has become more prevalent. This strategy corrects the shift by estimating a density ratio, defined as $w(y) = Q_y/P_y$. For the estimation, there exist two main approaches, moment matching and maximum likelihood.

**Moment matching.** Within the importance re-weighting framework, moment matching methods (also known as prediction probability matching) utilize the source classifier's confusion matrix, denoted by $C$, to efficiently estimate the weights, denoted by $w$, by solving the linear system $Cw = \mu$, where $\mu$ represents the target marginal predictions. Representative approaches range from direct matrix inversion methods like Black Box Shift Estimation (BBSE) (Lipton et al., 2018) and Regularized Learning under Label Shift (RLLS) (Azizzadenesheli et al., 2019) Recently, CPMCN (Wen et al., 2024) proposes to match the class probability distributions directly using calibrated networks.

*Equal contribution [1]School of Management, Fudan University, Shanghai, China. Correspondence to: Guanhua Fang <fanggh@fudan.edu.cn>.

*Proceedings of the 43$^{rd}$ International Conference on Machine Learning*, Seoul, South Korea. PMLR 306, 2026. Copyright 2026 by the author(s).

**Maximum likelihood.** The other major category in re-weighting framework, maximum likelihood estimation (MLE), estimates the importance weights by maximizing the likelihood of observed target features. Classic methods, such as the EM-based maximum likelihood algorithm (Saerens et al., 2002; Chan & Ng, 2005), iteratively update class priors to fit the target distribution. However, these methods heavily rely on the calibration of the predictive model $P(y|x)$, as miscalibrated networks often lead to estimation bias (Guo et al., 2017). Consequently, recent approaches explicitly integrate calibration techniques to mitigate this issue. Representative method such as bias-corrected calibration maximum likelihood label shift (MLLS) (Alexandari et al., 2020).

**The Gap & Challenges.** Despite the progress of existing label shift estimators, two critical challenges remain for reliable deployment. First, most approaches rely heavily on high-quality source classifiers, which become bottlenecks in data-scarce settings. Moment matching requires stable confusion matrix inversion (Lipton et al., 2018; Garg et al., 2020), while likelihood-based methods are highly sensitive to miscalibration (Guo et al., 2017; Alexandari et al., 2020). Although recent efforts attempt to refine calibration (Popordanoska et al., 2024) or estimation robustness (Fan et al., 2025), leveraging general-purpose pre-trained representations offers a more practical alternative (Wu et al., 2024). Second, standard estimators are often fragile to high-dimensional nuisances. While the standard label shift setting assumes strict conditional invariance $P(x|y) = Q(x|y)$, real-world data often contains domain-specific variations (e.g., sensor artifacts) that disrupt this pixel-level alignment. Recent benchmarks (Garg et al., 2023) and empirical studies (Park et al., 2023; Jang & Chung, 2024) demonstrate that SOTA estimators degrade substantially when this strict invariance is even slightly perturbed, as their reliance on raw inputs or fragile source classifiers limits robustness (Lee et al., 2025).

### 1.3. Our Method and Contributions

To bridge these gaps, we propose *regularized discriminative alignment for label shift (RDALS)*, a novel framework that rethinks label shift adaptation through the lens of *discriminative latent invariance*. Our method relies on three key pillars to ensure robustness and stability. Firstly, instead of relying on unstable raw inputs, we leverage the general-purpose semantic representations of large-scale pre-trained backbones (e.g., ResNet or ViT) (He et al., 2016; Dosovitskiy, 2020) as a sturdy foundation. Secondly, to further mitigate the impact of domain-specific nuisances, we introduce *Linear Discriminant Analysis (LDA)* to project these high-dimensional latent features into a compact, discriminative subspace. This step is crucial: by maximizing the between-class variance relative to the within-class variance, LDA effectively filters out non-semantic noise, ensuring that

the conditional invariance assumption ($P(z|y) \approx Q(z|y)$ with $z$ being a lower-dimensional representation of $x$) holds more rigorously in the feature space. Finally, to recover the importance weights $\boldsymbol{w}$, we adopt a *constrained optimization* objective inspired by RLLS (Azizzadenesheli et al., 2019). Rather than solving an unconstrained linear system, we minimize the distribution mismatch subject to simplex constraints, which guarantees numerically stable and non-negative weight estimates even in data-scarce regimes.

**Contributions.** The contributions of this paper are summarized as follows.

(i) Novel Framework: We propose a robust label shift adaptation framework that integrates pre-trained representations with LDA-based alignment and constrained optimization. This approach constructs a discriminative latent space that preserves conditional invariance more effectively than methods operating on raw inputs.

(ii) Theoretical Guarantees: We provide theoretical analysis establishing three key results: (1) the optimality of LDA in preserving shift information while reducing dimensionality; (2) the finite-sample error bounds for both the estimated label distribution $\hat{q}$ and importance weights $\hat{w}$; and (3) a generalization bound for downstream re-weighted classifier.

(iii) Empirical Validation: Extensive experiments on standard benchmarks demonstrate that our method significantly outperforms the state-of-the-art baselines. It exhibits superior stability and accuracy, particularly in challenging scenarios with **limited target supervision** or **severe distribution shifts**.

## 2. Preliminaries

### 2.1. Basic setting

We consider the unsupervised domain adaptation setting with an input space $\mathcal{X} \subset \mathbb{R}^D$ and a label space $\mathcal{Y} = \{1, \ldots, K\}$. Let $P$ and $Q$ denote the source and target distribution on $\mathcal{X} \times \mathcal{Y}$, respectively. We are given a labeled source dataset $S = \{(x_i, y_i)\}_{i=1}^{n_S}$ being an independent and identically distributed (i.i.d.) sample from $P$, and an unlabeled target dataset $T = \{x_j^{\text{tgt}}\}_{j=1}^{n_T}$ being an i.i.d. sample from the target marginal distribution of $x$, $Q_x$. Our goal is to learn a hypothesis $h : \mathcal{X} \to \mathcal{Y}$ that minimizes the target risk $\mathcal{R}_Q(h) \triangleq \mathbb{E}_Q[\ell(h(X), Y)]$, where $\ell$ is a bounded loss function and $(X, Y)$ stands for the population-level variable of $(x, y)$.

### 2.2. Label Shift

In this work, we focus on the label shift scenario. The following assumption is standard in the literature (Lipton et al., 2018; Azizzadenesheli et al., 2019; Garg et al., 2020).

**Assumption 2.1** (Standard Label Shift)**.** The marginal label distributions differ between the source and target domains

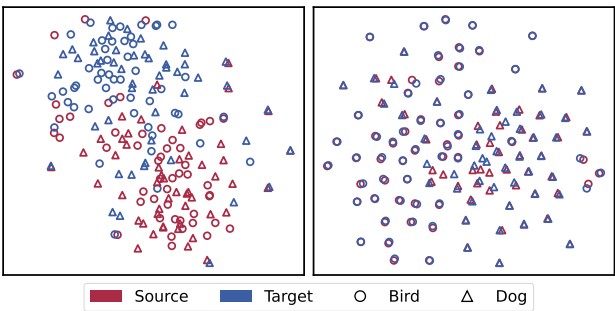

Source    Target    O Bird    △ Dog

*Figure 1.* t-SNE visualization of feature distributions on CIFAR-10 vs. CIFAR-10-C. Red and Blue denote Source and Target samples respectively. Left: Raw input space shows significant domain separation due to nuisances, making strict invariance fragile. Right: Pre-trained latent space preserves class alignment.

(i.e., $P_y \neq Q_y$), but the class-conditional distributions of the raw inputs remain invariant:

$$P(x|y) = Q(x|y), \quad \forall x \in \mathcal{X}, y \in \mathcal{Y}.$$

Under Assumption 2.1, the target joint distribution decomposes as $Q(x,y) = w(y)P(x,y)$, where $w(y) \triangleq Q_y/P_y$ is the importance weight for class $y$. Consequently, the target risk can be expressed as a re-weighted source risk:

$$\mathcal{R}_Q(h) = \mathbb{E}_P\left[w(Y)\ell(h(X), Y)\right].$$

The goal is to estimate the weight vector $\boldsymbol{w} = [w(1), \ldots, w(K)]^\top$. However, directly verifying or utilizing Assumption 2.1 in the high-dimensional raw input space is often intractable and sensitive to noise.

### 2.3. Robustness via Latent Conditional Invariance

To address the fragility of pixel-level matching, we propose to ground the estimation in the latent space. While raw inputs are sensitive to domain nuisances (Garg et al., 2023), deep representations extracted by pre-trained backbones demonstrate superior robustness. As illustrated in Figure 1, latent features maintain class-conditional alignment even when the raw input distribution shifts, providing a more reliable basis for estimation.

Formally, let $\phi : \mathcal{X} \to \mathcal{Z} \subset \mathbb{R}^d$ be a feature extractor mapping inputs $x$ to representations $z = \phi(x)$, where $\mathcal{Z}$ denotes the latent space. Accordingly, let $Z = \phi(X)$ denote the corresponding population-level random variable in $\mathcal{Z}$. We assume the dimension satisfies $d \geq K$ to ensure the feature space is sufficiently expressive.[1] Instead of relying on strict input invariance, RDALS operates under the following working assumption:

---

[1]Specifically, this condition ensures that the between-class scatter matrix in our LDA analysis avoids rank deficiency (rank $< K - 1$).

**Assumption 2.2** (Latent Conditional Invariance). Let $\phi$ be a fixed feature extractor. The class-conditional distributions of the latent features are invariant across domains:

$$P(z|y) = Q(z|y), \quad \forall z \in \mathcal{Z}, y \in \mathcal{Y}.$$

Assumption 2.2 serves as a robust proxy for Assumption 2.1. By leveraging the semantic consistency of deep representations, it enables RDALS to accurately estimate weights for standard label shift while maintaining resilience against input-level perturbations that break standard estimators.

## 3. Regularized Discriminative Alignment for Label Shift

In this section, we present our proposed method, Regularized Discriminative Alignment for Label Shift (RDALS), in four stages: (i) extracting a feature representation from raw inputs, (ii) constructing a linear system linking source and target data, (iii) optimizing summary functions via LDA, and (iv) solving $Q_y$ by a robust regularized estimator.

### 3.1. Nonlinear Feature Extraction

For a high-dimensional input $x \in \mathbb{R}^D$, directly applying moment matching in the raw input space is typically ineffective. We therefore introduce a (fixed) pretrained deep neural network feature extractor, denoted by

$$\phi : \mathcal{X} \to \mathbb{R}^d, \qquad z = \phi(x).$$

Because $\phi$ is a deterministic mapping, the Assumption 2.1 in the input space implies the corresponding Assumption 2.2 in the representation space: $P(z \mid y) = Q(z \mid y), \forall y \in \mathcal{Y}$. All subsequent computations are carried out in the feature space $\mathcal{Z} \subset \mathbb{R}^d$.

### 3.2. Moment-Matching Linear System

To connect the source and target domains, we utilize the law of total expectation. Suppose we have a set of $K - 1$ scalar summary functions $\{h_m\}_{m=1}^{K-1}$. For each $m$, the expectation on the target domain can be decomposed as:

$$\mathbb{E}_Q[h_m(Z)] = \sum_{j=1}^{K} \mathbb{E}_Q[h_m(Z) \mid Y = j]\, q_j,$$

where $q_j = Q_y(Y = j)$, $j \in [K]$. Under Assumption 2.2, the class-conditional expectations are invariant across domains. Substituting this yields:

$$\underbrace{\mathbb{E}_Q[h_m(Z)]}_{b_m} = \sum_{j=1}^{K} \underbrace{\mathbb{E}_P[h_m(Z) \mid Y = j]}_{A_{mj}}\, q_j. \tag{1}$$

Collecting these equations for $m = 1, \ldots, K - 1$ gives the linear moment system:

$$b = Aq, \qquad q = (q_1, \ldots, q_K)^\top \in \Delta^{K-1}, \tag{2}$$

with

$$\Delta^{K-1} = \left\{ q \in \mathbb{R}^K : q_j \geq 0, \sum_{j=1}^{K} q_j = 1 \right\} \subset \mathbb{R}^K. \quad (3)$$

Here, $b \in \mathbb{R}^{K-1}$ is the target moment vector, and $A \in \mathbb{R}^{(K-1)\times K}$ is the source class-conditional moment matrix defined by the summary functions. Theoretical analysis (see Section 4) suggests that the estimation error of $q$ is inversely proportional to the minimum singular value of the matrix associated with $A$, i.e., $\sigma_{\min}(\tilde{A})$. Therefore, the choice of summary functions $h(\cdot)$ is critical: arbitrarily chosen functions may lead to an ill-conditioned system (i.e., rows of $A$ are linearly dependent), making the solution unstable.

### 3.3. Optimal Summary Functions via LDA

To ensure the linear system (2) is well-conditioned and numerically stable, the summary functions should maximize the separability between classes. We adopt multiclass Fisher's Linear Discriminant Analysis (LDA), which seeks directions that maximize the ratio of between-class scatter to within-class scatter.

Recall that the LDA directions are obtained from the generalized eigenvalue problem $S_B w = \lambda S_W w$, where $S_B$ and $S_W$ are defined in Section 4.2. Let $w_1, \ldots, w_{K-1}$ be the top $K - 1$ generalized eigenvectors. We define the summary functions as the projection of features onto these discriminative directions:

$$h_m(z) = w_m^\top z, \qquad m = 1, \ldots, K - 1.$$

where $h_m(z)$ denotes the $m$-th component of $h(z)$. In matrix form, we have $h(z) = W^\top z \in \mathbb{R}^{K-1}$ where $W = [w_1, \ldots, w_{K-1}]$. By construction, these functions extract the most discriminative information. As we show in Theorem 4.1, this choice is theoretically optimal for maximizing the minimum singular value of the system, providing the most robust basis for recovering the label distribution.

**Remark.** In high-dimensional regimes (e.g., $d \geq n_S$) where the empirical within-class scatter $\hat{S}_W$ may be singular or ill-conditioned, we use a shrinkage regularization $\hat{S}_{W,\epsilon} := \hat{S}_W + \epsilon I$ with a small $\epsilon > 0$ to ensure numerical well-conditioning. We then compute the discriminant directions by solving $\hat{S}_B v = \lambda \hat{S}_{W,\epsilon} v$, and form $\widehat{W}$ from the leading $K - 1$ generalized eigenvectors.

### 3.4. Regularized Estimation of Label Ratios

Given the linear system derived above, a direct solution based on inversion $\hat{q} \approx \hat{A}^\dagger \hat{b}$ is often unstable due to sampling noise and ill-conditioning. Instead of solving the equation directly, we reformulate the problem as a constrained optimization task. By enforcing the probability simplex constraint and introducing the regularization, we solve:

$$\hat{q} \in \arg \min_{q \in \Delta^{K-1}} \left\{ \|\hat{A}q - \hat{b}\|_2 + \lambda \|q\|_2 \right\}, \qquad (4)$$

---

**Algorithm 1** Regularized Discriminative Alignment for Label Shift (RDALS)

---

**Require:** Source data $S = \{(x_i, y_i)\}_{i=1}^{n_S} \sim P(x, y)$, Target data $T = \{x_j^{\text{tgt}}\}_{j=1}^{n_T} \sim Q_x(x)$, feature extractor $\phi$, loss function $\ell$, regularization parameter $\lambda$.

1: Extracting features $z_i = \phi(x_i)$, and $z_j^{\text{tgt}} = \phi(x_j^{\text{tgt}})$.
2: Fitting LDA summary functions. Using $\{(z_i, y_i)\}_{i=1}^{n_S}$ to fit an LDA model and attain the transformation matrix $\hat{W}$. Let $\hat{h}(z) = \hat{W}^\top z$ and construct $\hat{A}, \hat{b}$ by plug-in estimators.
3: Solve the SOCP problem

$$\hat{q} \in \arg \min_{q \in \Delta^{K-1}} \left\{ \|\hat{A}q - \hat{b}\|_2 + \lambda \|q\|_2 \right\}.$$

4: Obtain $\hat{Q}_y$ from $\hat{q}$ and $\hat{P}_y$ from the empirical label distribution of $S$. Compute the importance weights $\hat{w}(y) = \hat{Q}_y / \hat{P}_y, \quad y \in \mathcal{Y}$.
5: Train the final weighted ERM classifier

$$\hat{f}_{\hat{w}} = \arg \min_{f \in \mathcal{F}} \frac{1}{n_S} \sum_{i=1}^{n_S} \hat{w}(y_i) \ell(f(x_i), y_i).$$

**Ensure:** Predictive model $\hat{f}_{\hat{w}}$.

---

where $\Delta^{K-1}$ enforces the probability simplex constraint, and $\lambda > 0$ is a regularization parameter for preventing the overfitting to sampling noise.

### 3.5. Weighted Empirical Risk Minimization

We use $\hat{q}$ to estimate the target label distribution, with the probability estimation for each class $y \in \mathcal{Y}$ being denoted by $\hat{Q}_y$. Then the corresponding importance weight is computed as $\hat{w}(y) = \hat{Q}_y / \hat{P}_y$, where $\hat{P}_y$ is the empirical label distribution of the source domain.

Finally, we adapt the classifier to the target domain by minimizing the importance-weighted empirical risk in some function class $\mathcal{F}$ on the source data:

$$\hat{f}_{\hat{w}} = \arg \min_{f \in \mathcal{F}} \frac{1}{n_S} \sum_{i=1}^{n_S} \hat{w}(y_i) \ell(f(x_i), y_i).$$

By re-weighting the source samples, this objective approximates the risk on the target domain, thereby correcting for the label shift. For readers' convenience, the full algorithm is summarized in Algorithm 1.

## 4. Theoretical results

In this section, we establish the theoretical foundations of our proposed method. Firstly, we show that LDA is the *optimal* choice of linear summary function for identifying the label-shift vector $q$, in the sense that it maximizes the minimum singular value of the resulting estimating-equation

system and hence minimizes sensitivity to estimation noise. Secondly, we provide finite-sample error bounds for the estimated weights $\hat{q}$ and $\hat{w}$. Thirdly, we translate these weight-estimation errors into an excess-risk bound for the downstream ERM classifier. All formal assumptions and complete proofs are collected in Appendix B.

### 4.1. Construction of Estimating Equations

For any integrable function $h$, the latent conditional invariance assumption implies (1). Both sides of (1) can be estimated empirically, where $\mathbb{E}_Q[h(Z)]$ can be approximated by the target sample average $\mathbb{Q}_{n_T}[h(Z)]$ and $\mathbb{E}_P[h(Z) \mid Y = j]$ by the class-conditional source average $\mathbb{P}_{n_S}[h(Z) \mid Y = j]$. The system is solved under the simplex constraint with an additional regularization term, as formalized in (4).

To study numerical stability, it is useful to remove the simplex constraint and work in a $K - 1$ dimensional contrast space around the source label distribution $p$. Specifically, after a standard contrast reparametrization of the simplex, the constrained system $b = Aq, q \in \Delta^{K-1}$, can be written as the square linear system

$$\tilde{b} = \tilde{A}\tilde{q}, \qquad \tilde{b} := b - Ap, \qquad \tilde{A} := AD^{1/2}C_p,$$

where $D = \text{diag}(p)$, $C_p \in \mathbb{R}^{K \times (K-1)}$ is a contrast matrix orthogonal to $D^{1/2}\mathbf{1}_K$, and $\tilde{q} \in \mathbb{R}^{K-1}$ is the corresponding label-shift coordinate. The precise construction of $C_p$ and the equivalence between the simplex-constrained system and this square system are given in Appendix B.2.

This reparametrization separates the baseline source label distribution $p$ from the label-shift perturbation. The matrix $\tilde{A}$ describes how perturbations in the label distribution are translated into changes in the observed moments. Hence the stability of estimating $q$ is governed by the conditioning of $\tilde{A}$, i.e., a small $\sigma_{\min}(\tilde{A})$ makes the recovered weights sensitive to perturbations in the empirical moments and class-conditional estimates.

### 4.2. LDA optimality for moment stability

First, we denote $\mu_j := \mathbb{E}_P[Z \mid Y = j], \Sigma_j := \text{Cov}_P(Z \mid Y = j)$, and corresponding divergence matrices $S_W := \sum_{j=1}^{K} p_j \Sigma_j$, $S_B := \sum_{j=1}^{K} p_j (\mu_j - \bar{\mu})(\mu_j - \bar{\mu})^\top$, where $\bar{\mu} := \sum_{j=1}^{K} p_j \mu_j$.

Classical multiclass Fisher linear discriminant analysis (LDA) (Fisher, 1936; McLachlan, 2005; Hart et al., 2001) seeks a projection matrix $W \in \mathbb{R}^{d \times (K-1)}$ that maximizes the between-class dispersion relative to the within-class dispersion:

$$J(W) = \text{tr}\left\{ \left(W^\top S_W W\right)^{-1} \left(W^\top S_B W\right) \right\}.$$

Equivalently, the LDA subspace is spanned by the leading $K - 1$ generalized eigenvectors of $S_B u = \lambda S_W u$.

For later use, it is convenient to rewrite this generalized eigenvalue problem in a whitened coordinate system. Define $\tilde{S}_B := S_W^{-1/2} S_B S_W^{-1/2}$, and let $\tilde{S}_B = \tilde{U} \Lambda \tilde{U}^\top$ be its eigen-decomposition, where $\Lambda = \text{diag}(\lambda_1, \ldots, \lambda_d), \lambda_1 \geq \lambda_2 \geq \cdots \geq \lambda_d \geq 0$, and $\tilde{U} = [u_1, \ldots, u_d]$ is orthonormal. The corresponding LDA directions are $w_m = S_W^{-1/2} u_m, m = 1, \ldots, K - 1$.

Thus, classical LDA selects the directions along which the class-conditional means are most separated after normalizing for within-class variation. In our label-shift moment system, this same geometric criterion also controls numerical stability. Theorem 4.1 shows that the LDA subspace maximizes the smallest singular value of the moment matrix $\tilde{A}(W)$, and therefore yields the best-conditioned linear system for estimating $q$ within the normalized class of projections.

**Theorem 4.1.** *Suppose Assumption 2.2 holds. For any* $W \in \mathbb{R}^{d \times (K-1)}$*, the moment matrix satisfies*

$$\sigma_{\min}(\tilde{A}(W))^2 = \lambda_{\min}(W^\top S_B W),$$

*where* $\sigma_{\min}$ *denotes the smallest singular value and* $\lambda_{\min}$ *denotes the smallest eigenvalue. Consider the scale-normalized class*

$$\mathcal{W} := \left\{ W \in \mathbb{R}^{d \times (K-1)} : W^\top S_W W = I_{K-1} \right\}.$$

*Then*

$$\max_{W \in \mathcal{W}} \sigma_{\min}(\tilde{A}(W))^2 = \lambda_{K-1}(\tilde{S}_B).$$

*Moreover, the maximum is attained by the multiclass LDA subspace, i.e., any optimizer* $W^*$ *satisfies*

$$\text{col}(W^*) = \text{col}\left( S_W^{-1/2} U_{1:(K-1)} \right),$$

*where* $U_{1:(K-1)} = [u_1, \ldots, u_{K-1}]$ *contains the leading* $K - 1$ *eigenvectors of* $\tilde{S}_B$.

### 4.3. Generalization Bound

We employ sample splitting to decouple the construction of the moment functions from the estimation of the moment equations. Randomly split the labeled source sample $S$ into two disjoint subsets $S_1$ and $S_2$, with index sets $I_1$ and $I_2$, such that $I_1 \cup I_2 = \{1, \ldots, n_S\}$ and $I_1 \cap I_2 = \emptyset$. The first split $S_1$ is used to estimate the LDA projection matrix $\hat{W}$, while the second split $S_2$, together with the unlabeled target sample $T$, is used to estimate the moment system. Conditional on $S_1$, the functions $\hat{h}(z) = \hat{W}^\top z$ can be treated as fixed linear moment functions.

Let $n_{i,j}$ be the number of observations in $S_i$ with label $j$, for $i \in \{1, 2\}$ and $j \in [K]$. Using $S_1$, we estimate the class-conditional mean and covariance by

$\hat{\mu}_j = n_{1,j}^{-1} \sum_{i \in I_1 : Y_i = j} Z_i$ and $\hat{\Sigma}_j = n_{1,j}^{-1} \sum_{i \in I_1 : Y_i = j} (Z_i - \hat{\mu}_j)(Z_i - \hat{\mu}_j)^\top$. Throughout the analysis, we condition on the empirical source label proportions $p_j = n_{S,j}/n_S$ and treat $p = (p_1, \ldots, p_K)^\top$ as fixed. Define $\hat{\bar{\mu}} = \sum_{j=1}^K p_j \hat{\mu}_j$, $\hat{S}_W = \sum_{j=1}^K p_j \hat{\Sigma}_j$, and $\hat{S}_B = \sum_{j=1}^K p_j (\hat{\mu}_j - \hat{\bar{\mu}})(\hat{\mu}_j - \hat{\bar{\mu}})^\top$. Let $\widehat{\overline{S}}_B = \hat{S}_W^{-1/2} \hat{S}_B \hat{S}_W^{-1/2}$. If $\hat{U} = [\hat{u}_1, \ldots, \hat{u}_{K-1}]$ contains the leading $K-1$ eigenvectors of $\widehat{\overline{S}}_B$, then the empirical LDA projection is $\hat{W} = \hat{S}_W^{-1/2} \hat{U}$.

Given $\hat{W}$, we estimate the moment matrix $A$ using the independent source split $S_2$, and estimate the target moment vector $b$ using the target sample $T$:

$$\hat{A}_{mj} = \frac{1}{n_{2,j}} \sum_{i \in I_2 : Y_i = j} \hat{h}_m(Z_i), \qquad \hat{b}_m = \frac{1}{n_T} \sum_{i=1}^{n_T} \hat{h}_m(Z_i),$$

for $m = 1, \ldots, K-1$ and $j \in [K]$. We then solve the regularized SOCP in Algorithm 1 to obtain the estimated target label distribution $\hat{q}$. The simplex constraints ensure that $\hat{q}$ is a valid label distribution, while the regularization improves numerical stability, similarly in spirit to RLLS (Azizzade-nesheli et al., 2019).

After estimating $\hat{q}$, we form the importance weights $\hat{w}(j) = \hat{q}_j/p_j$, $j \in [K]$, and train the final classifier by weighted empirical risk minimization:

$$\hat{f}_{\hat{w}} = \arg\min_{f \in \mathcal{F}} \frac{1}{n_S} \sum_{i=1}^{n_S} \hat{w}(y_i) \ell(f(X_i), Y_i).$$

At the population level, let $q^*$ denote the oracle solution to the population moment system $Aq = b$. The following theorem provides a high-probability upper bound for the estimation error $\|\hat{q} - q^*\|_2$.

**Theorem 4.2.** *Let $p_{\max} := \max_{j \in [K]} p_j$. Suppose that Assumption B.1, B.2 and the conditions in B.3 hold. For $\hat{q}$ as defined in equation (4), we have with probability at least $1 - \delta$,*

$$\|\hat{q} - q^*\|_2 \leq \epsilon_{q^*}(n_S, n_T, \delta, \lambda),$$

*where*

$$\epsilon_{q^*}(n_S, n_T, \delta, \lambda) := \frac{\sqrt{p_{\max}}}{\sigma_{\min}(\tilde{A})} \left( (\Delta_A(\delta/10) + \lambda) \|q^*\|_2 + 2\Delta_b(\delta/10) \right).$$

*Here $\Delta_A(\delta)$ and $\Delta_b(\delta)$ are the probability upper bounds for $\|\hat{A} - A\|$ and $\|\hat{b} - b\|$ respectively.*

The proofs and the explicit forms of $\Delta_A(\delta)$ and $\Delta_b(\delta)$ are given in Appendix B.

The dependence on $\sigma_{\min}(\tilde{A})$ reflects the stability requirement. Since $q^*$ is characterized through a linear system,

the sensitivity of the solution to perturbations in $(A, b)$ is governed by the conditioning of $\tilde{A}$. Accordingly, a larger $\sigma_{\min}(\tilde{A})$ attenuates the impact of estimation noise in $(\hat{A}, \hat{b})$, making the resulting solution less sensitive. The term $(\Delta_A + \lambda)\|q^*\|_2$ captures the combined effect of stochastic error in $\hat{A}$ and the explicit regularization $\lambda$, while the additive $2\Delta_b$ accounts for the stochastic error in $\hat{b}$.

Under the definition $w^*(j) = q_j^*/p_j$ and $\hat{w}(j) = \hat{q}_j/p_j$, for each $j \in [K]$ we have $|\hat{w}_j - w_j^*| = |\hat{q}_j - q_j^*|/p_j \leq \|\hat{q} - q^*\|_\infty/p_{\min}$, where $p_{\min} := \min_{j \in [K]} p_j > 0$. It follows that $\|\hat{w} - w^*\|_\infty \leq p_{\min}^{-1} \|\hat{q} - q^*\|_2$, and substituting the bound from Theorem 4.2 yields the claimed result.

**Corollary 4.3.** *Assume that the conditions in 4.2 hold. Then with probability greater than $1 - \delta$,*

$$\|\hat{w} - w^*\|_\infty \leq \frac{1}{p_{\min}} \epsilon_{q^*}(n_S, n_T, \delta, \lambda).$$

**Theorem 4.4** (Generalization bound). *Let $n_S$, $n_T$ be the source and target sample sizes, $\mathcal{F}$ a hypothesis class, and $\ell$ a loss function. Define $\mathcal{G} := \{(x, y) \mapsto w^*(y)\ell(f(x), y) : f \in \mathcal{F}\}$. Under Assumption 2.1 and the conditions of Theorem 4.2, the following generalization bound holds: with probability at least $1 - 2\delta$,*

$$\mathcal{R}_Q(\hat{f}_{\hat{w}}) - \mathcal{R}_Q(f^*) \leq$$
$$\frac{2}{p_{\min}} \epsilon_{q^*}(n_S, n_T, \delta, \lambda) + 4\operatorname{Rad}_n(\mathcal{G}) + \frac{2}{p_{\min}} \sqrt{\frac{\log(2/\delta)}{n_S}}.$$

Theorem 4.4 bounds the target excess risk of the predictor trained with estimated importance weights, $\hat{f}_{\hat{w}}$, in terms of (i) the error incurred by estimating the shift weights and (ii) the usual statistical complexity of the learning problem. The leading term $\epsilon_{q^*}(n_S, n_T, \delta, \lambda)/p_{\min}$ is the contribution of weight estimation error. The term $\operatorname{Rad}_n(\mathcal{G})$ is a uniform convergence term controlling the gap between empirical and population risks over class $\mathcal{G}$. The concentration term $\sqrt{\log(2/\delta)/n_S}/p_{\min}$ reflects sampling fluctuation from the finite labeled source sample used to estimate the weighted risk.

## 5. Experiments

In this section, we empirically evaluate the effectiveness of our proposed method **RDALS** against state-of-the-art label shift adaptation baselines. We focus on two key metrics, (1) the estimation error of the importance weights, and (2) the classification accuracy on the target domain.

### 5.1. Experimental Setup

**Datasets and Shift Types.** We evaluate RDALS on three standard benchmarks: MNIST, CIFAR-10, and CIFAR-100. We simulate label shift using a pre-trained ResNet-18/ViT-B-16 backbone under two types: (1) Dirichlet Shift, where the target label distribution is drawn from $Dir(\alpha)$, with

*Table 1.* Performance comparison of RDALS against baselines on MNIST, CIFAR-10, and CIFAR-100. We report the Mean Squared Error (MSE) for weight estimation and Accuracy/Macro-F1 for downstream classification. All results are averaged over 100 independent trials.

| Method | MNIST | | | CIFAR-10 | | | CIFAR-100 | | |
|---|---|---|---|---|---|---|---|---|---|
| | MSE | ACC(%) | Macro F1(%) | MSE | ACC(%) | Macro F1(%) | MSE | ACC(%) | Macro F1(%) |
| BBSL | 3.195e-1 | 96.95 | 96.96 | 1.322e+0 | 84.60 | 84.44 | 7.917e+0 | 44.92 | 37.71 |
| | (6.995e-1) | (0.01) | (0.01) | (2.343e+0) | (0.01) | (0.02) | (1.423e+0) | (0.02) | (0.03) |
| RLLS | 3.195e-1 | 96.95 | 96.96 | 1.058e+0 | 84.71 | 84.61 | 5.340e+0 | 51.81 | 46.78 |
| | (6.995e-1) | (0.01) | (0.01) | (1.718e+0) | (0.01) | (0.01) | (1.118e+0) | (0.02) | (0.02) |
| MLLS | 5.895e-2 | 96.98 | 96.99 | 8.222e-2 | 85.01 | 84.97 | 9.415e+0 | 40.23 | 36.52 |
| | (2.977e-1) | (0.00) | (0.00) | (1.177e-1) | (0.01) | (0.01) | (2.498e+0) | (0.03) | (0.03) |
| CPMCN | 7.190e-2 | 96.99 | 96.99 | 1.829e-1 | 85.02 | 84.98 | 5.251e+0 | 51.59 | 46.97 |
| | (1.205e-1) | (0.00) | (0.00) | (2.867e-1) | (0.01) | (0.01) | (1.063e+0) | (0.02) | (0.02) |
| RDALS | **2.339e-2** | **97.00** | **97.01** | **5.179e-2** | **85.06** | **85.02** | **2.679e+0** | **58.03** | **54.55** |
| | (2.720e-2) | (0.00) | (0.00) | (5.912e-2) | (0.01) | (0.01) | (9.319e-1) | (0.02) | (0.02) |

$\alpha \in (0, 1]$ representing severe shifts; and (2) Tweak-One Shift, where a single class probability $\rho$ is varied to simulate extreme imbalance.

**Baselines.** We compare RDALS against representative state-of-the-art methods, including moment-matching approaches BBSL, RLLS and CPMCN(Lipton et al., 2018; Azizzadenesheli et al., 2019; Wen et al., 2024) and maximum likelihood-based methods MLLS(Alexandari et al., 2020). For calibration-sensitive methods (MLLS and CPMCN), we report results across multiple calibration strategies (e.g., TS, NBVS, BCTS) to ensure a comprehensive comparison.

**Implementation and Metrics.** Unlike baselines that require validation splits for calibration, RDALS maximizes data efficiency by utilizing the full source dataset. We evaluate performance using Mean Squared Error (MSE) for weight estimation and Accuracy/Macro F1 for downstream adaptation. Full implementation details and hyperparameter settings are deferred to Appendix A.

### 5.2. Main Results: Estimation and Adaptation

**Superior Performance on Main Benchmarks.** Table 1 benchmarks RDALS against other baselines under *Dirichlet shift* ($\alpha = 1$) with a sample size of $N = 10,000$. RDALS consistently achieves the lowest MSE and highest ACC/Macro-F1 across all datasets. Notably, on the challenging CIFAR-100 benchmark, RDALS reduces the MSE by nearly half compared to the best-performing baseline.

We further extend evaluations to varying sample sizes ($N \in \{1000, 5000, 10000\}$) and shift intensities. Detailed results are provided in Appendix A.4: Tables 7, 8, and 9 report the estimation MSE under varying Dirichlet concentrations ($\alpha$); Tables 10 and 11 present the MSE under Tweak-One shift (varying $\rho$); and Tables 12, 13, and 14 detail the downstream Accuracy and Macro-F1 scores under Dirichlet shift. Finally, we conducted experiments on CIFAR-10 and CIFAR-10-C

with different *Dirichlet Shift* $\alpha$ and sample size in Table 15 to empirically validate the *Latent Conditional Invariance* assumption (Section 2.3), further verifying the robustness of our method against input perturbations.

**Visualization of Distribution Alignment.** We further provide a qualitative visualization of feature alignment in Figure 2 by projecting features onto the 1D LDA subspace. While the original source distribution (gray) deviates significantly from the target (blue), the source distribution reweighted by RDALS (red) aligns almost perfectly with the target. This visually confirms that our derived weights effectively correct the label shift and recover the target marginal distribution.

**Real world style domain shift on PACS.** We further conduct experiments on the real world PACS dataset to evaluate the effectiveness of RDALS under more practical domain variations. Specifically, we use three stylistic domains, Art, Cartoon, and Sketch, and randomly select one domain as the source domain and another as the target domain. This setting introduces natural style variation between source and target data, which weakens the strict conditional invariance assumption in the raw input space and therefore provides a more realistic test for our latent conditional invariance assumption. As shown in Table 2, RDALS consistently achieves the lowest MSE for importance weight estimation and the lowest $\ell_1$ error for target label distribution estimation across all evaluated domain shifts. This shows that aligning distributions in the discriminative latent space remains robust under real world style variations.

### 5.3. Ablation Study: Dissecting the Source of Gains

For ablation studies, we focus on CIFAR-10 with *Dirichlet shift* ($\alpha = 1.0$), as it represents a sufficiently challenging scenario to distinguish the contribution of each component.

**Effectiveness of Discriminative Projection.** We evaluate

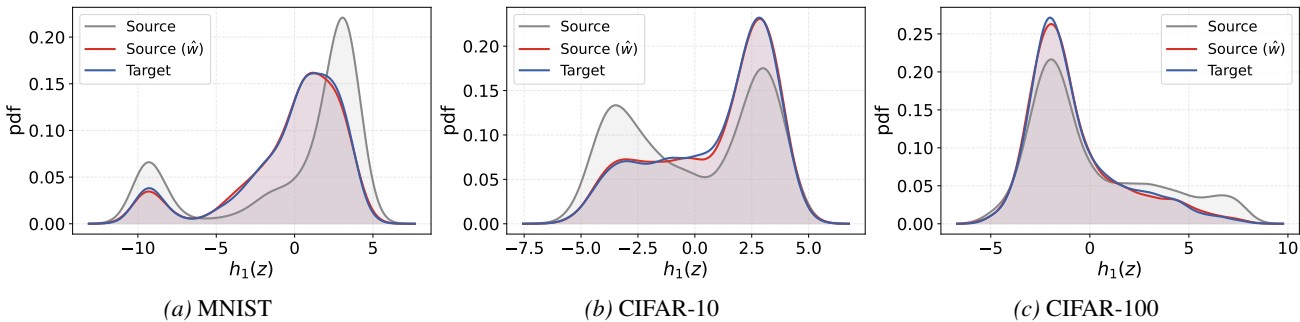

*(a)* MNIST    *(b)* CIFAR-10    *(c)* CIFAR-100

*Figure 2.* Visualization of feature alignment in the 1D LDA subspace using Kernel Density Estimation (KDE). We use Dirichlet shift with $\alpha = 1$ and sample size 10000. We plot the probability density function (pdf) of the Target distribution (Blue), the original Source distribution (Gray, unweighted), and the Source distribution re-weighted by $\hat{w}$ (Red).

*Table 2.* Performance comparison of RDALS and baselines on the real world PACS dataset. We report the MSE for importance weight estimation and the $\ell_1$ error for target label distribution estimation across different stylistic domain shifts.

| Source | Target | Method | MSE($\hat{w}$) | $\ell_1(\hat{q})$ |
|---|---|---|---|---|
| Art | Cartoon | BBSL | 2.3864 | 1.0701 |
| | | RLLS | 2.3864 | 1.0701 |
| | | MLLS | 3.5883 | 1.3759 |
| | | CPMCN | 3.8850 | 1.3840 |
| | | **RDALS** | **1.4327** | **0.9144** |
| Art | Sketch | BBSL | 4.5176 | 1.3583 |
| | | RLLS | 4.5469 | 1.3627 |
| | | MLLS | 4.6235 | 1.4227 |
| | | CPMCN | 3.2068 | 1.3175 |
| | | **RDALS** | **1.3747** | **0.7735** |
| Cartoon | Sketch | BBSL | 1.1877 | 0.7043 |
| | | RLLS | 1.1866 | 0.7041 |
| | | MLLS | 1.7530 | 0.9038 |
| | | CPMCN | 2.3306 | 1.1906 |
| | | **RDALS** | **0.8661** | **0.6025** |

*Table 3.* Ablation study on projection methods for summary functions $h(z)$ (MNIST, CIFAR-10, and CIFAR-100, Dirichlet shift with $\alpha = 1$, sample size 10000). We compare LDA (Ours) with PCA and Random Projection (orthogonalized).

| Dataset | Projection | Mechanism | MSE |
|---|---|---|---|
| MNIST | Random | Random $\mathcal{N}(0,1)$ | $2.195 \pm 1.866$ |
| | PCA | Max Variance | $2.938 \pm 2.303$ |
| | LDA | Max Separability | $\mathbf{0.018 \pm 0.020}$ |
| CIFAR-10 | Random | Random $\mathcal{N}(0,1)$ | $0.736 \pm 0.647$ |
| | PCA | Max Variance | $0.573 \pm 0.736$ |
| | LDA | Max Separability | $\mathbf{0.051 \pm 0.002}$ |
| CIFAR-100 | Random | Random $\mathcal{N}(0,1)$ | $4.063 \pm 1.052$ |
| | PCA | Max Variance | $3.248 \pm 0.998$ |
| | LDA | Max Separability | $\mathbf{1.686 \pm 0.530}$ |

### 5.4. Stability and Efficiency Analysis

**Robustness to Feature Perturbations.** To validate robustness against Latent Conditional Invariance (LCI) perturbations, we inject zero-mean Gaussian noise into the target features, defined as $Z'_{tgt} = Z_{tgt} + \epsilon_i$ where $\epsilon_i \sim \mathcal{N}(0, \Sigma_{noise})$, with $\Sigma_{noise}$ being a diagonal matrix with elements $(r \cdot \sigma_j)^2$, $\sigma_j$ the empirical standard deviation of the $j$-th dimension, and $r \in [0,1]$ the noise scale. As Figure 3 illustrates, the MSE remains stable as $r$ increases. This resilience is grounded in our first-order moment matching: $\hat{b}_{noisy} = \hat{b}_{clean} + W^\top(\frac{1}{n_T}\sum_{i=1}^{n_T}\epsilon_i)$. By the Law of Large Numbers, the empirical mean of the zero-mean noise converges to zero, ensuring $\hat{b}_{noisy} \approx \hat{b}_{clean}$ and leaving the estimated label distribution $\hat{q}$ essentially unaffected.

**Calibration-free Stability.** RDALS is theoretically calibration-free. As shown in Figure 4, we benchmark against various strategies (TS, VS, NBVS, BCTS). While baselines exhibit significant performance variance, visualized as large vertical spreads depending on the calibration choice, RDALS consistently achieves the lowest estimation error without any tuning, demonstrating superior "plug-and-play" stability.

our LDA approach against Principal Components Analysis (PCA) and Random Projection to isolate the impact of subspace selection. To ensure a fair comparison, we enforce orthogonality across all methods, applying QR decomposition to the Random baseline. Table 3 shows that LDA significantly outperforms both alternatives (MSE: Random > PCA > LDA).

**Importance of Regularized Solver**. We further decouple the solver's impact by comparing our regularized SOCP against a Direct Inverse ($\hat{q} = (\hat{A}^\top\hat{A})^{-1}\hat{A}^\top\hat{b}$) baseline (Table 4). Results show that the SOCP solver drastically improves stability across all feature types. Crucially, the combination of LDA and SOCP yields the minimal error, demonstrating that discriminative alignment and algorithmic regularization are complementary and indispensable.

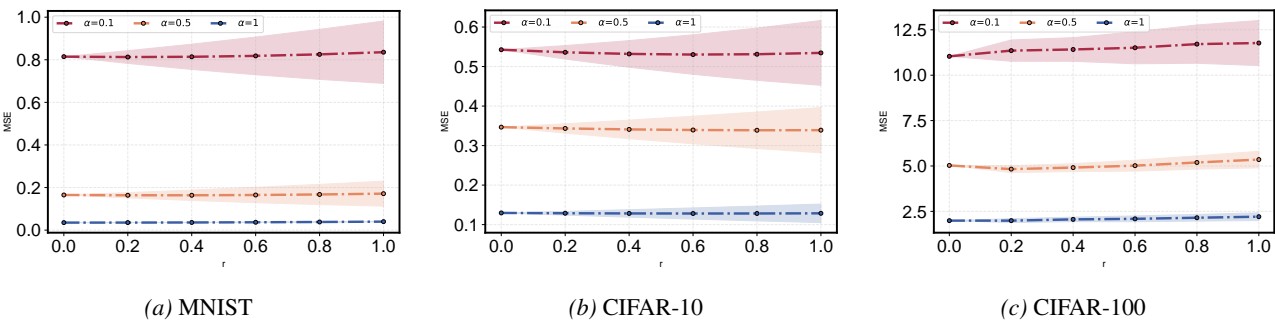

*(a)* MNIST      *(b)* CIFAR-10      *(c)* CIFAR-100

*Figure 3.* Robustness of RDALS against Latent Conditional Invariance (LCI) perturbations. The figure shows the estimation error (MSE) as we inject zero-mean Gaussian noise into the target features, where the noise scale is controlled by a ratio $r \in [0, 1]$.

*Table 4.* Ablation study decoupling the impact of feature projection and solver mechanism on MNIST, CIFAR-10, and CIFAR-100 ($\alpha = 1.0$). We compare the estimation error (MSE) across four combinations.

| Dataset | Projection | Solver Mechanism | |
|---|---|---|---|
| | | Direct Inverse | SOCP |
| MNIST | PCA | 14.117 | 2.962 |
| | LDA | 3.879 | **0.014** (Ours) |
| CIFAR-10 | PCA | 7.664 | 0.489 |
| | LDA | 4.785 | **0.056** (Ours) |
| CIFAR-100 | PCA | 22.632 | 3.298 |
| | LDA | 29.325 | **1.703** (Ours) |

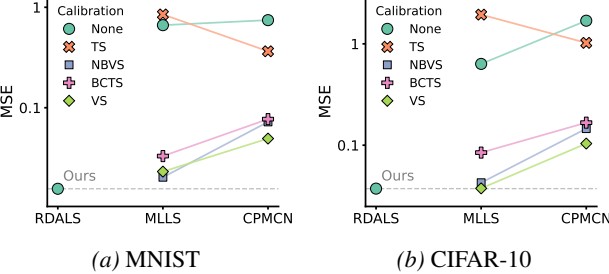

*(a)* MNIST      *(b)* CIFAR-10

*Figure 4.* Impact of calibration strategies on estimation error (MSE) under Dirichlet shift ($\alpha = 1$). The vertical dispersion indicates the baselines' high sensitivity to calibration hyperparameters, in contrast to the invariant and superior performance of RDALS.

**Resilience & Efficiency.** We extensively evaluate the robustness of RDALS under broader experimental settings. Firstly, our method exhibits **superior resilience** under severe label shifts (varying Dirichlet $\alpha$) in Figure 5, maintaining high precision even when baseline methods degrade due to extreme distribution divergence. Secondly, we verify that our improvements are agnostic to **model architecture** in Figure 6, consistently outperforming baselines across different backbones (ResNet-50, ViT-B-16). Next, RDALS demonstrates significant **computational efficiency** in Table 5, achieving approximately $2\times$ faster runtime compared

to competitive baselines. Finally, in Table 6 we analyze the method's robustness of regularized parameter $\lambda$ in Eq. 4. Detailed results and discussions for these evaluations are provided in Appendix A.5.

## 6. Conclusions

In this paper, we propose Regularized Discriminative Alignment for Label Shift (RDALS), a robust framework that addresses the fragility of standard estimators by leveraging latent conditional invariance. By integrating theoretically optimal LDA projections with a constrained SOCP solver, RDALS maximizes numerical stability and estimation accuracy, particularly in data-scarce and extreme-shift regimes. We provide rigorous finite-sample error bounds and generalization guarantees to validate our framework. Empirically, RDALS consistently outperforms state-of-the-art baselines across standard benchmarks, offering a computationally efficient and calibration-free solution for reliable domain adaptation in the wild. Future work will explore extending this framework to more relaxed test-time adaptation settings.

**Acknowledgment** Guanhua Fang's research is partly supported by National Key R&D Program of China (Grant No. 2024YFA1015700), National Natural Science Foundation of China (12301376), and Shanghai Educational Development Foundation (23CGA02). Wen Yu's research is supported by the National Natural Science Foundation of China Grants (12071088). Ming Zheng's research is supported by the National Natural Science Foundation of China Grants (12271106).

## Impact Statement

This work enhances model robustness under distribution shift, benefiting safety-critical applications like healthcare and finance. There are many potential societal consequences of our work, none of which we feel must be specifically highlighted here.

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

# Supplementary of "Regularized Discriminative Alignment for Deep Representations under Label Shift"

## A. Experimental Details and Additional Results

### A.1. Label Shift and Data Generation Details

To rigorously evaluate the robustness of our method, we simulate label shift by systematically manipulating the marginal label distributions of the source ($P_y$) and target ($Q_y$) domains while keeping the class-conditional distributions invariant.

**Dirichlet Shift Generation.** To simulate diverse distribution shifts, we generate label probability vectors using a Dirichlet distribution. Specifically, a probability vector $\mathbf{q}$ is sampled from $Dir(\alpha \mathbf{1}_K)$, where $\mathbf{1}_K$ is a vector of ones and $\alpha > 0$ is the concentration parameter. We conduct a fine-grained evaluation over the range $\alpha \in \{0.1, 0.3, 0.5, 0.7, 0.9, 1.0\}$. A smaller $\alpha$ leads to highly imbalanced distributions where most probability mass is concentrated on a few classes, representing severe shift scenarios.

**Tweak-One Shift Generation.** To test robustness against extreme outliers and "disappearing" classes, we employ the *Tweak-One* shift protocol. In this setting, we explicitly construct a probability distribution where a chosen class $k$ is assigned a fixed probability $\rho$ (i.e., $q_k = \rho$). The remaining probability mass $1 - \rho$ is distributed uniformly among the other $K - 1$ classes, such that $q_j = \frac{1-\rho}{K-1}$ for $j \neq k$. We explore an extreme range of $\rho \in \{0.01, 0.02, 0.03, 0.6, 0.7, 0.8\}$. The lower bound ($\rho = 0.01$) simulates the minority class scenario (rare events), while the upper bound ($\rho = 0.8$) simulates the majority class domination, challenging the estimator's ability to handle extreme imbalance.

**Data Resampling Protocol.** After defining the target (or source) label probability vector $\mathbf{q}$ (or $\mathbf{p}$) via the strategies above, we generate shift in source domain to simulate real world scenario, then we construct the actual datasets using stratified sampling with replacement. Specifically, to generate a dataset of size $N$ with a marginal distribution $\mathbf{q}$, we independently sample $N \cdot q_k$ instances for each class $k$ from the original pool. Sampling with replacement ensures that we can satisfy precise distributional constraints even when the required number of samples for a specific class exceeds its count in the original dataset.

**Data Efficiency and Splitting Strategy.** Standard baseline methods, including BBSL, RLLS, MLLS, and CPMCN, typically necessitate splitting the labeled source dataset $S$ (e.g., into an 80% training split and a 20% validation split). This hold-out validation set is structurally required to obtain unbiased confusion matrix estimates or to perform temperature scaling calibration to mitigate overfitting. In contrast, RDALS is theoretically grounded in the alignment of latent statistics and does not require post-hoc calibration. This allows our method to utilize the *full* source dataset to estimate the LDA parameters and latent moments, thereby maximizing data efficiency and estimation stability, particularly in data-scarce regimes.

### A.2. Model Architectures and Training Details

**Code Availability.** The RDALS source code is available on GitHub at `https://github.com/fduhcshi/RDALS`. The baseline implementations (BBSL, RLLS, MLLS, CPMCN) are adapted from their respective official repositories or standard benchmark libraries.

**Backbone Architectures.** We implement all deep learning backbones using the public *torchvision* library. Specifically, we employ:

- **ResNet-18**: Used for MNIST and CIFAR-10 experiments. We utilize the default weights pre-trained on ImageNet-1k.

- **ViT-B-16**: Used for CIFAR-100 experiments to handle the higher complexity of the 100-class classification task. This model is also initialized with ImageNet-1k pre-trained weights.

**Phase 1: Importance Weight Estimation (RDALS vs. Baselines).** A key distinction in our setup lies in the computational requirements for weight estimation:

- **RDALS (Ours):** Our method is computationally efficient and *training-free* during the estimation phase. We utilize the frozen backbone directly as a fixed feature extractor $\phi(\cdot)$. The summary functions are derived via LDA on the source features without any gradient-based fine-tuning.

- **Baselines:** Methods such as the standard matching approaches (BBSL, RLLS), and calibration-dependent approaches (MLLS, CPMCN) require a calibrated source classifier to estimate confusion matrices or density ratios. For a fair comparison, we train these initial classifiers by freezing the pre-trained backbone and exclusively training the linear classification head on the source dataset $S$.

**Phase 2: Downstream Adaptation.** After obtaining the estimated weights $\hat{w}$, we retrain the models on the weighted source dataset to evaluate adaptation performance. We adopt architecture-specific fine-tuning strategies to balance plasticity and stability:

- **ResNet-18 (MNIST/CIFAR-10):** We fine-tune the last residual block (Layer 4) and the classification head, keeping earlier layers frozen to preserve generic low-level features.

- **ViT-B-16 (CIFAR-100):** Given the data-hungry nature of Transformers and the limited size of CIFAR-100, we adopt a *Linear Probing* strategy, fine-tuning only the final classification head while keeping the entire backbone frozen to prevent overfitting.

**Optimization and Hyperparameters.** All training phases utilize Stochastic Gradient Descent (SGD) with a momentum of 0.9 and a batch size of 128. We adopt distinct schedules for different stages to ensure fair and stable comparisons:

- **Initial Classifiers for Baselines:** For the calibration or confusion matrix estimation in baseline methods, we train the initial source classifiers with a learning rate of 0.001 for 10 epochs. This setup aligns with the protocols described in prior work (e.g., RLLS), ensuring that the baselines are initialized under standard conditions.

- **Regularization for RDALS:** For the regularization parameter $\lambda$ in our SOCP objective (Eq. 4), we fix $\lambda = 0.001$ across all experiments, in Table 6 demonstrating the method's robustness without intensive hyperparameter tuning.

- **Downstream Weighted Classifiers:** For the final adaptation phase (training the re-weighted classifier), we set the learning rate to 0.001 across all datasets. Regarding training duration, we empirically observed that Accuracy and Macro-F1 scores typically plateau around 20 epochs for MNIST and CIFAR-10, and between 30–40 epochs for CIFAR-100.

*Note:* Since our primary objective is to evaluate the **relative** effectiveness of label shift estimators rather than chasing absolute state-of-the-art classification scores, we adopt this early stopping strategy. This prevents overfitting to the source set and ensures that the performance differences reflect the quality of the estimated importance weights rather than training noise.

### A.3. Evaluation Metrics

To provide a comprehensive assessment of both the estimation precision and the downstream adaptation effectiveness, we employ the following three metrics. All reported results are averaged over 100 independent trials to ensure statistical significance, with standard deviations provided in parentheses.

**1. Mean Squared Error (MSE).** We primarily assess the quality of importance weight estimation using the Mean Squared Error (MSE), defined as:

$$\text{MSE} = \frac{1}{K} \sum_{k=1}^{K} (\hat{w}_k - w_k)^2,$$

where $\hat{w}_k$ and $w_k$ represent the estimated and ground-truth importance weights for class $k$, respectively. A lower MSE indicates a more precise recovery of the target label distribution.

**2. Classification Accuracy (ACC).** To evaluate the effectiveness of the adaptation, we report the overall classification accuracy on the target domain. This metric reflects the global performance of the re-weighted classifier:

$$\text{ACC} = \frac{1}{n_T} \sum_{i=1}^{n_T} \mathbb{I}(\hat{y}_i = y_i),$$

where $\hat{y}_i$ is the predicted label for the $i$-th target sample, $y_i$ is the true label, and $\mathbb{I}(\cdot)$ is the indicator function.

**3. Macro-F1 Score.** Given that label shift naturally induces class imbalance (especially in our severe shift settings where $\alpha = 0.1$ or $\rho = 0.01$), Accuracy can be misleading if the model is biased towards majority classes. Therefore, we also report the **Macro-F1 Score**, which calculates the F1 score for each class independently and then takes the unweighted mean:

$$\text{Macro F1} = \frac{1}{K} \sum_{k=1}^{K} \frac{2 \cdot \text{Precision}_k \cdot \text{Recall}_k}{\text{Precision}_k + \text{Recall}_k}.$$

This metric treats all classes equally, regardless of their sample size.

### A.4. Extended Experiments Results

**Estimation performance of $\hat{w}$ under *Dirichlet Shift*.** We present the comprehensive estimation error (MSE) results of $\hat{w}$ under Dirichlet shift across a fine-grained range of shift intensities ($\alpha \in [0.1, 1.0]$) and varying target sample sizes ($N \in \{1000, 5000, 10000\}$) in **Table 7** (MNIST), **Table 8** (CIFAR-10), and **Table 9** (CIFAR-100). The results consistently demonstrate that RDALS outperforms state-of-the-art baselines with significantly lower error and variance, establishing its superiority particularly in challenging data-scarce regimes ($N = 1000$) and severe distribution shifts ($\alpha = 0.1$).

**Estimation performance of $\hat{w}$ under *Tweak-one Shift*.** We evaluate the robustness of RDALS against extreme class imbalance using the Tweak-one shift protocol, varying the target probability $\rho \in \{0.01, 0.02, 0.03, 0.6, 0.7, 0.8\}$. The estimation errors reported in **Table 10** (MNIST) and **Table 11** (CIFAR-10) confirm that RDALS maintains superior stability even when a class nearly vanishes ($\rho = 0.01$) or dominates the distribution. Note that we omit CIFAR-100 from this specific evaluation, as its high cardinality ($K = 100$) implies a uniform baseline probability of 0.01.

**Prediction performance of downstream classifier.** We report the post-adaptation classification Accuracy and Macro-F1 scores in **Table 12** (MNIST), **Table 13** (CIFAR-10), and **Table 14** (CIFAR-100). The results confirm that the precise weight estimation of RDALS directly translates into superior downstream adaptation, consistently outperforming baselines across all datasets.

**Results on the *Latent Conditional Invariance* assumption (Section 2.3).** We extended our evaluation to the CIFAR-10-C dataset, which introduces input corruptions alongside label shift. We conducted experiments with varying *Dirichlet Shift* parameters $\alpha$ and sample sizes, as detailed in Table 15. The results demonstrate that RDALS maintains superior estimation precision and classification accuracy compared to baselines, even when pixel-level invariance is compromised. This confirms that aligning distributions in the discriminative latent space provides a robust invariant proxy, effectively mitigating the impact of input perturbations.

### A.5. Details for Resilience & Efficiency in Section 5.4.

**Resilience to Severe Label Shift.** We evaluate the robustness of RDALS under extreme distribution shifts by varying the Dirichlet parameter $\alpha \in \{0.1, \ldots, 1.0\}$. As illustrated in Figure 5, RDALS consistently outperforms baselines across all shift intensities.

**Generalization Across Architectures.** To verify that our improvements are not specific to a single architecture, we extend our evaluations to diverse backbones, including ResNet-50 and ViT-B-16. As shown in Figure 6, RDALS consistently outperforms leading baselines (MLLS, CPMCN) across all tested architectures. While stronger architectures like Vision Transformers generally improve the quality of latent features, RDALS maintains a significant performance margin. This result confirms that our discriminative alignment provides a fundamental structural advantage that is independent of the specific feature extractor used.

**Computational Efficiency.** We assess the computational cost of different adaptation methods on the CIFAR-10 dataset, averaging results over 100 independent trials. To ensure a fair comparison, the recorded time strictly covers the *weight estimation phase*, recording immediately after feature extraction and concluding once the importance weights $\hat{w}$ are obtained. We exclude the time for feature extraction and downstream classifier training, as these steps are identical across all methods. As reported in Table 5, RDALS is approximately $2\times$ faster than the competitive baseline CPMCN. This efficiency stems from our calibration-free formulation, RDALS solves a disciplined convex optimization problem directly on the pre-extracted features, making it highly suitable for resource-constrained adaptation scenarios.

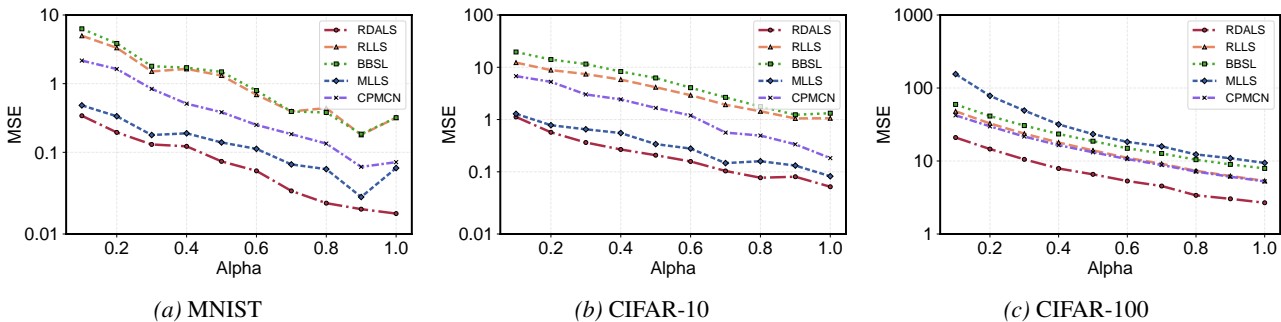

*Figure 5.* MSE of importance weight estimation under varying Dirichlet shift intensities ($\alpha \in (0, 1]$). We compare RDALS against baselines on MNIST, CIFAR-10, and CIFAR-100 with a fixed sample size of $N = 10,000$. A smaller $\alpha$ indicates a more severe label shift.

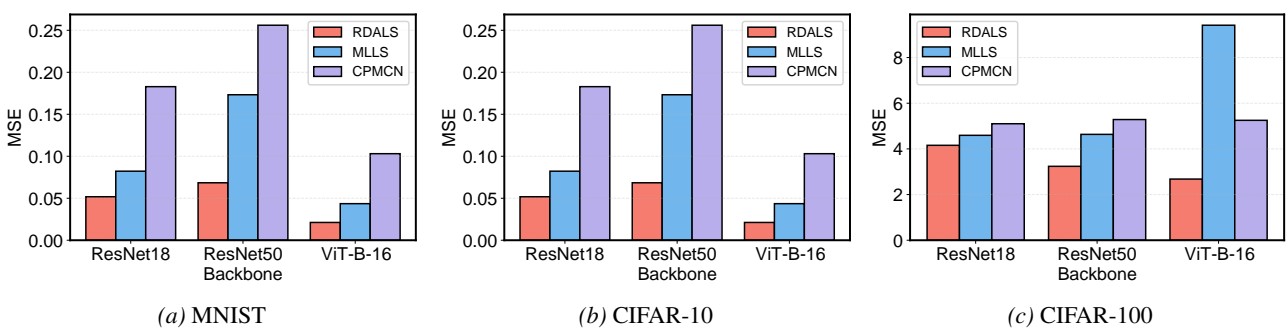

*Figure 6.* Comparison of weight estimation error (MSE) on MNIST, CIFAR-10, and CIFAR-100 using different backbones (ResNet-18, ResNet-50, and ViT-B-16). RDALS (red) consistently achieves the lowest error, demonstrating that the structural advantage of discriminative alignment generalizes across both CNN and Transformer-based architectures.

*Table 5.* Running time comparison (in seconds) on CIFAR-10 across varying sample sizes. Results are averaged over 100 independent trials and measure the estimation phase exclusively (post-feature extraction).

| Sample size | BBSL | RLLS | MLLS | CPMCN | RDALS (Ours) |
|---|---|---|---|---|---|
| 1000 | **31.50** | 32.08 | 32.18 | 39.55 | 36.30 |
| 5000 | 81.23 | 81.65 | 82.49 | 99.55 | **55.39** |
| 10000 | 137.17 | 138.11 | 138.69 | 167.82 | **74.18** |

*Table 6.* Sensitivity analysis of the regularization parameter $\lambda$. RDALS demonstrates robustness to variations in $\lambda$, maintaining stable performance across a wide range of values.

| Sample Size | $\lambda$ | | | | | |
|---|---|---|---|---|---|---|
| | $10^{-6}$ | $10^{-5}$ | $10^{-4}$ | $10^{-3}$ | $10^{-2}$ | $10^{-1}$ |
| 1000 | 6.766e-1 | 6.756e-1 | 7.302e-1 | 5.557e-1 | 8.442e-1 | 5.208e-1 |
| 10000 | 4.555e-2 | 4.576e-2 | 5.833e-2 | 4.910e-2 | 6.002e-2 | 4.857e-2 |

*Table 7.* **Estimation Error (MSE) on MNIST under Dirichlet Shift.** We evaluate weight estimation performance across varying shift intensities $\alpha \in \{0.1, \ldots, 1.0\}$ and target sample sizes $N \in \{1000, 5000, 10000\}$. For each entry, the first row reports the mean MSE, and the second row (in parentheses) reports the standard deviation averaged over 100 independent trials.

| Sample size | Method | 0.1 | 0.2 | 0.3 | 0.5 | 0.7 | 1 |
|---|---|---|---|---|---|---|---|
| | BBSL | 4.986e+1 | 3.622e+1 | 2.659e+1 | 1.502e+1 | 8.793e+0 | 7.043e+0 |
| | | (1.230e+1) | (1.138e+1) | (9.541e+0) | (6.973e+0) | (5.311e+0) | (3.948e+0) |
| | RLLS | 4.064e+1 | 2.918e+1 | 2.099e+1 | 1.198e+1 | 6.764e+0 | 5.229e+0 |
| | | (1.001e+1) | (9.203e+0) | (7.617e+0) | (5.735e+0) | (4.190e+0) | (3.144e+0) |
| 1000 | MLLS | 2.275e+1 | 1.339e+1 | 8.387e+0 | 5.391e+0 | 2.396e+0 | 1.656e+0 |
| | | (1.358e+1) | (9.389e+0) | (5.894e+0) | (5.182e+0) | (2.734e+0) | (1.910e+0) |
| | CPMCN | 1.628e+1 | 1.315e+1 | 1.049e+1 | 6.141e+0 | 3.780e+0 | 2.729e+0 |
| | | (1.050e+1) | (7.473e+0) | (5.344e+0) | (4.072e+0) | (3.228e+0) | (2.376e+0) |
| | Ours | **4.028e+0** | **2.866e+0** | **2.101e+0** | **1.277e+0** | **6.191e-1** | **4.803e-1** |
| | | **(1.837e+0)** | **(1.463e+0)** | **(1.316e+0)** | **(8.435e-1)** | **(6.002e-1)** | **(4.764e-1)** |
| | BBSL | 2.508e+1 | 1.210e+1 | 8.881e+0 | 4.101e+0 | 3.192e+0 | 1.168e+0 |
| | | (1.874e+1) | (7.213e+0) | (8.023e+0) | (4.634e+0) | (5.184e+0) | (2.293e+0) |
| | RLLS | 1.655e+1 | 1.026e+1 | 7.304e+0 | 3.111e+0 | 2.493e+0 | 1.025e+0 |
| | | (9.992e+0) | (5.663e+0) | (6.141e+0) | (3.120e+0) | (3.336e+0) | (1.919e+0) |
| 5000 | MLLS | 1.952e+0 | 9.493e-1 | 5.489e-1 | 1.726e-1 | 1.517e-1 | 6.162e-2 |
| | | (1.707e+0) | (7.924e-1) | (5.400e-1) | (1.690e-1) | (1.760e-1) | (8.334e-2) |
| | CPMCN | 4.427e+0 | 3.257e+0 | 1.450e+0 | 7.204e-1 | 4.210e-1 | 1.097e-1 |
| | | (3.262e+0) | (2.415e+0) | (1.400e+0) | (9.577e-1) | (6.109e-1) | (1.645e-1) |
| | Ours | **6.517e-1** | **4.890e-1** | **2.929e-1** | **1.381e-1** | **1.209e-1** | **5.690e-2** |
| | | **(3.506e-1)** | **(3.132e-1)** | **(2.153e-1)** | **(1.226e-1)** | **(1.202e-1)** | **(6.258e-2)** |
| | BBSL | 6.295e+0 | 3.854e+0 | 1.795e+0 | 1.486e+0 | 3.933e-1 | 3.195e-1 |
| | | (5.627e+0) | (4.522e+0) | (2.396e+0) | (2.493e+0) | (6.664e-1) | (6.995e-1) |
| | RLLS | 4.980e+0 | 3.325e+0 | 1.507e+0 | 1.313e+0 | 3.941e-1 | 3.195e-1 |
| | | (4.333e+0) | (3.679e+0) | (1.779e+0) | (2.079e+0) | (6.675e-1) | (6.995e-1) |
| 10000 | MLLS | 4.829e-1 | 3.352e-1 | 1.786e-1 | 1.392e-1 | 6.662e-2 | 5.895e-2 |
| | | (3.466e-1) | (3.048e-1) | (1.727e-1) | (1.892e-1) | (8.419e-2) | (2.977e-1) |
| | CPMCN | 2.168e+0 | 1.635e+0 | 8.400e-1 | 3.838e-1 | 1.840e-1 | 7.190e-2 |
| | | (2.070e+0) | (1.625e+0) | (7.923e-1) | (4.037e-1) | (2.203e-1) | (1.205e-1) |
| | Ours | **3.407e-1** | **1.950e-1** | **1.302e-1** | **7.375e-2** | **3.824e-2** | **2.339e-2** |
| | | **(1.687e-1)** | **(1.341e-1)** | **(1.033e-1)** | **(7.011e-2)** | **(3.664e-2)** | **(2.720e-2)** |

*Table 8.* **Estimation Error (MSE) on CIFAR-10 under Dirichlet Shift.** This table presents a comprehensive comparison of RDALS against baselines across different degrees of label shift and data availability. The first row indicates the mean MSE, while the value in parentheses represents the standard deviation over 100 trials.

| Sample size | Method | 0.1 | 0.2 | 0.3 | 0.5 | 0.7 | 1.0 |
|---|---|---|---|---|---|---|---|
| 1000 | BBSL | 5.107e+1 (1.045e+1) | 3.746e+1 (1.030e+1) | 2.764e+1 (9.783e+0) | 1.636e+1 (6.379e+0) | 1.093e+1 (6.326e+0) | 6.442e+0 (3.592e+0) |
| | RLLS | 4.177e+1 (8.620e+0) | 3.020e+1 (8.307e+0) | 2.179e+1 (7.863e+0) | 1.275e+1 (5.320e+0) | 8.425e+0 (5.221e+0) | 4.726e+0 (2.821e+0) |
| | MLLS | 1.866e+1 (7.990e+0) | 1.514e+1 (6.766e+0) | 1.246e+1 (6.610e+0) | 7.577e+0 (4.121e+0) | 4.021e+0 (3.396e+0) | 2.090e+0 (2.077e+0) |
| | CPMCN | 1.993e+1 (1.219e+1) | 1.932e+1 (8.416e+0) | 1.548e+1 (6.522e+0) | 1.038e+1 (4.611e+0) | 6.518e+0 (4.547e+0) | 3.704e+0 (2.683e+0) |
| | Ours | **1.067e+1** (**5.276e+0**) | **7.761e+0** (**4.523e+0**) | **5.603e+0** (**3.760e+0**) | **3.480e+0** (**2.836e+0**) | **1.652e+0** (**1.552e+0**) | **8.278e-1** (**7.828e-1**) |
| 5000 | BBSL | 3.786e+1 (1.181e+1) | 2.790e+1 (9.642e+0) | 1.929e+1 (9.778e+0) | 1.290e+1 (7.419e+0) | 6.832e+0 (4.691e+0) | 3.956e+0 (3.783e+0) |
| | RLLS | 3.038e+1 (8.033e+0) | 2.264e+1 (7.255e+0) | 1.552e+1 (7.374e+0) | 9.758e+0 (5.313e+0) | 5.624e+0 (3.756e+0) | 2.954e+0 (2.757e+0) |
| | MLLS | 4.676e+0 (**2.278e+0**) | 3.169e+0 (1.947e+0) | 2.009e+0 (1.507e+0) | 1.187e+0 (1.187e+0) | 4.782e-1 (5.820e-1) | 2.862e-1 (3.936e-1) |
| | CPMCN | 1.156e+1 (5.821e+0) | 7.932e+0 (4.319e+0) | 6.584e+0 (3.911e+0) | 4.029e+0 (3.345e+0) | 2.423e+0 (2.331e+0) | 8.040e-1 (1.251e+0) |
| | Ours | **4.107e+0** (5.731e+0) | **1.352e+0** (**1.029e+0**) | **8.602e-1** (**7.255e-1**) | **4.428e-1** (**3.524e-1**) | **2.697e-1** (**2.423e-1**) | **1.252e-1** (**1.426e-1**) |
| 10000 | BBSL | 1.961e+1 (1.471e+1) | 1.405e+1 (1.088e+1) | 1.153e+1 (1.062e+1) | 6.251e+0 (7.227e+0) | 2.658e+0 (3.778e+0) | 1.322e+0 (2.343e+0) |
| | RLLS | 1.228e+1 (6.115e+0) | 8.835e+0 (5.977e+0) | 7.403e+0 (5.075e+0) | 4.153e+0 (4.189e+0) | 1.921e+0 (2.598e+0) | 1.058e+0 (1.718e+0) |
| | MLLS | 1.284e+0 (**7.589e-1**) | 7.789e-1 (6.034e-1) | 6.505e-1 (5.826e-1) | 3.388e-1 (3.271e-1) | 1.467e-1 (1.620e-1) | 8.222e-2 (1.177e-1) |
| | CPMCN | 6.769e+0 (3.961e+0) | 5.268e+0 (3.740e+0) | 3.021e+0 (2.584e+0) | 1.669e+0 (1.517e+0) | 5.626e-1 (8.031e-1) | 1.829e-1 (2.867e-1) |
| | Ours | **1.121e+0** (1.509e+0) | **5.703e-1** (**3.749e-1**) | **3.635e-1** (**2.545e-1**) | **2.079e-1** (**1.586e-1**) | **1.038e-1** (**1.027e-1**) | **5.179e-2** (**5.912e-2**) |

*Table 9.* **Estimation Error (MSE) on CIFAR-100 under Dirichlet Shift.** Investigating performance in a high-dimensional output space ($K = 100$). We report the mean MSE in first row and standard deviation (in parentheses) for varying $\alpha$ and sample sizes $N$. Note that smaller $\alpha$ indicates more severe distribution shift.

| Sample size | Method | 0.1 | 0.2 | 0.3 | 0.5 | 0.7 | 1 |
|---|---|---|---|---|---|---|---|
| 5000 | BBSL | 6.017e+1 (1.616e+1) | 4.218e+1 (1.348e+1) | 3.140e+1 (1.120e+1) | 1.955e+1 (4.831e+0) | 1.263e+1 (3.579e+0) | 7.976e+0 (1.780e+0) |
| | RLLS | 4.810e+1 (1.078e+1) | 3.295e+1 (7.373e+0) | 2.419e+1 (7.326e+0) | 1.459e+1 (3.228e+0) | 9.005e+0 (2.041e+0) | 5.381e+0 (1.050e+0) |
| | MLLS | 1.779e+2 (2.500e+3) | 8.015e+1 (8.510e+2) | 4.168e+1 (2.016e+2) | 2.148e+1 (1.922e+1) | 1.433e+1 (1.340e+1) | 8.802e+0 (5.469e+0) |
| | CPMCN | 4.372e+1 (**3.835e+0**) | 2.983e+1 (3.543e+0) | 2.101e+1 (**2.823e+0**) | 1.291e+1 (**1.719e+0**) | 8.209e+0 (1.443e+0) | 5.043e+0 (9.481e-1) |
| | Ours | **2.257e+1** (4.298e+0) | **1.731e+1** (**3.262e+0**) | **1.312e+1** (2.900e+0) | **8.068e+0** (1.764e+0) | **5.291e+0** (**1.307e+0**) | **3.263e+0** (**7.785e-1**) |
| 10000 | BBSL | 5.958e+1 (3.688e+0) | 4.129e+1 (3.208e+0) | 3.056e+1 (3.288e+0) | 1.870e+1 (2.399e+0) | 1.269e+1 (1.736e+0) | 7.917e+0 (1.423e+0) |
| | RLLS | 4.799e+1 (**2.747e+0**) | 3.271e+1 (**2.454e+0**) | 2.369e+1 (2.624e+0) | 1.396e+1 (1.844e+0) | 9.164e+0 (1.368e+0) | 5.340e+0 (1.118e+0) |
| | MLLS | 1.551e+2 (3.936e+1) | 7.813e+1 (2.373e+1) | 4.915e+1 (1.625e+1) | 2.334e+1 (5.950e+0) | 1.586e+1 (4.248e+0) | 9.415e+0 (2.498e+0) |
| | CPMCN | 4.224e+1 (4.282e+0) | 2.984e+1 (2.706e+0) | 2.165e+1 (2.523e+0) | 1.317e+1 (1.700e+0) | 8.812e+0 (**1.305e+0**) | 5.251e+0 (1.063e+0) |
| | Ours | **2.097e+1** (4.271e+0) | **1.459e+1** (2.801e+0) | **1.053e+1** (**2.481e+0**) | **6.565e+0** (**1.661e+0**) | **4.541e+0** (1.410e+0) | **2.679e+0** (**9.319e-1**) |

*Table 10.* **Estimation Error (MSE) on MNIST under Tweak-one Shift.** We manually vary the probability of a single class ($\rho$) to simulate outlier scenarios ranging from rare events ($\rho = 0.01$) to majority dominance ($\rho = 0.8$). Results are reported across different sample sizes $N$, showing the mean MSE and standard deviation (in parentheses).

| Sample size | Method | 0.01 | 0.02 | 0.03 | 0.6 | 0.7 | 0.8 |
|---|---|---|---|---|---|---|---|
| 1000 | BBSL | 9.763e+0 (9.144e-1) | 2.311e+0 (1.251e-1) | 9.474e-1 (5.741e-2) | 2.888e+0 (1.774e+0) | 4.296e+0 (1.925e+0) | 1.339e+1 (2.666e+0) |
| | RLLS | 8.075e+0 (8.177e-1) | 1.591e+0 (2.184e-1) | 5.846e-1 (4.624e-2) | 1.654e+0 (9.826e-1) | 2.970e+0 (1.567e+0) | 8.874e+0 (1.698e+0) |
| | MLLS | 6.363e+0 (**4.668e-2**) | 6.959e-1 (7.033e-1) | 1.209e-1 (2.756e-2) | 3.433e-1 (1.563e-1) | 1.154e+0 (4.336e-1) | 5.363e+0 (2.361e+0) |
| | CPMCN | 8.188e+0 (4.692e-2) | 1.123e+0 (1.165e+0) | 4.483e-2 (2.587e-2) | 9.457e-1 (5.639e-1) | 1.189e+0 (1.312e+0) | 4.075e-1 (4.269e-1) |
| | Ours | **5.222e-1** (3.058e-1) | **4.521e-2** (**3.097e-2**) | **1.625e-2** (**1.049e-2**) | **5.639e-2** (**2.657e-2**) | **1.329e-1** (**5.828e-2**) | **3.962e-1** (**1.873e-1**) |
| 5000 | BBSL | 9.801e+0 (1.254e-2) | 4.059e-1 (7.482e-1) | 1.860e-2 (2.126e-2) | 7.590e-2 (3.840e-2) | 2.400e-1 (1.212e-1) | 1.702e+0 (1.314e+0) |
| | RLLS | 8.088e+0 (**1.048e-2**) | 3.336e-1 (4.815e-1) | 1.860e-2 (2.126e-2) | 7.590e-2 (3.840e-2) | 2.400e-1 (1.212e-1) | 1.633e+0 (1.165e+0) |
| | MLLS | 8.544e-1 (6.958e-1) | 4.460e-2 (4.325e-2) | 7.088e-3 (5.565e-3) | 1.893e-2 (8.864e-3) | 5.302e-2 (3.566e-2) | 2.110e-1 (9.624e-2) |
| | CPMCN | 9.759e-1 (8.999e-1) | 5.162e-2 (4.027e-2) | 1.045e-2 (6.668e-3) | 3.419e-2 (1.860e-2) | 8.120e-2 (4.290e-2) | 2.558e-1 (1.189e-1) |
| | Ours | **4.763e-2** (4.908e-2) | **4.651e-3** (**3.418e-3**) | **2.468e-3** (**1.238e-3**) | **1.593e-2** (**6.476e-3**) | **3.390e-2** (**1.155e-2**) | **1.168e-1** (**5.094e-2**) |
| 10000 | BBSL | 7.415e+0 (3.655e+0) | 5.585e-2 (7.196e-2) | 9.233e-3 (8.445e-3) | 2.560e-2 (1.312e-2) | 8.825e-2 (3.860e-2) | 4.634e-1 (2.508e-1) |
| | RLLS | 7.393e+0 (1.776e+0) | 5.585e-2 (7.196e-2) | 9.233e-3 (8.445e-3) | 2.560e-2 (1.312e-2) | 8.825e-2 (3.860e-2) | 4.630e-1 (2.507e-1) |
| | MLLS | 6.721e-1 (4.295e-1) | 2.255e-2 (1.537e-2) | 4.428e-3 (3.792e-3) | **1.026e-2** (4.477e-3) | 2.650e-2 (1.172e-2) | 8.938e-2 (3.952e-2) |
| | CPMCN | 5.853e-1 (3.729e-1) | 2.365e-2 (2.020e-2) | 5.288e-3 (4.163e-3) | 1.758e-2 (8.380e-3) | 3.825e-2 (1.849e-2) | 1.391e-1 (5.926e-2) |
| | Ours | **1.832e-2** (2.170e-2) | **2.628e-3** (**1.749e-3**) | **1.741e-3** (**7.375e-4**) | 1.028e-2 (**3.537e-3**) | **2.325e-2** (**8.722e-3**) | **7.417e-2** (**2.568e-2**) |

*Table 11.* **Estimation Error (MSE) on CIFAR-10 under Tweak-One Shift.** Comparison of weight estimation stability under extreme class imbalance. The first row in each entry represents the mean MSE, and the second row (in parentheses) denotes the standard deviation.

| Sample size | Method | 0.01 | 0.02 | 0.03 | 0.6 | 0.7 | 0.8 |
|---|---|---|---|---|---|---|---|
| 1000 | BBSL | 9.862e+0 | 2.473e+0 | 1.103e+0 | 4.287e+0 | 6.241e+0 | 1.786e+1 |
| | | (8.265e-1) | (3.248e-1) | (3.021e-1) | (2.089e+0) | (9.152e-1) | (6.924e-1) |
| | RLLS | 8.023e+0 | 1.703e+0 | 7.158e-1 | 2.511e+0 | 3.206e+0 | 1.101e+1 |
| | | (1.078e+0) | (5.013e-1) | (2.556e-1) | (1.368e+0) | (4.731e-1) | (**3.201e-1**) |
| | MLLS | 4.424e+0 | 6.831e-1 | 1.444e-1 | 4.638e-1 | 1.280e+0 | 4.612e+0 |
| | | (2.211e+0) | (4.454e-1) | (1.191e-1) | (2.142e-1) | (5.386e-1) | (1.939e+0) |
| | CPMCN | 5.503e+0 | 8.308e-1 | 2.660e-1 | 1.857e+0 | 1.194e+0 | **8.611e-1** |
| | | (3.239e+0) | (6.392e-1) | (2.025e-1) | (1.322e+0) | (1.921e+0) | (7.440e-1) |
| | Ours | **1.548e+0** | **1.807e-1** | **5.145e-2** | **1.431e-1** | **3.580e-1** | 2.479e+0 |
| | | (**7.092e-1**) | (**1.345e-1**) | (**3.353e-2**) | (**7.138e-2**) | (**2.020e-1**) | (1.930e+0) |
| 5000 | BBSL | 9.800e+0 | 1.641e+0 | 2.905e-1 | 2.823e-1 | 1.674e+0 | 1.414e+1 |
| | | (1.472e-2) | (8.133e-1) | (6.088e-1) | (1.780e-1) | (1.532e+0) | (1.400e+1) |
| | RLLS | 8.087e+0 | 1.222e+0 | 2.106e-1 | 2.750e-1 | 1.104e+0 | 6.526e+0 |
| | | (**1.268e-2**) | (5.286e-1) | (3.346e-1) | (1.593e-1) | (6.312e-1) | (2.921e+0) |
| | MLLS | 6.004e-1 | 4.889e-2 | 1.238e-2 | 3.725e-2 | 1.152e-1 | 5.291e-1 |
| | | (7.126e-1) | (4.689e-2) | (1.203e-2) | (1.452e-2) | (4.577e-2) | (2.561e-1) |
| | CPMCN | 3.359e+0 | 9.266e-2 | 2.327e-2 | 9.123e-2 | 1.378e-1 | 2.482e-1 |
| | | (3.735e+0) | (1.032e-1) | (2.368e-2) | (5.726e-2) | (9.935e-2) | (1.392e-1) |
| | Ours | **2.842e-1** | **3.051e-2** | **9.108e-3** | **2.697e-2** | **5.626e-2** | **2.156e-1** |
| | | (1.816e-1) | (**2.284e-2**) | (**6.662e-3**) | (**1.233e-2**) | (**2.278e-2**) | (**8.613e-2**) |
| 10000 | BBSL | 9.224e+0 | 2.387e-1 | 2.311e-2 | 7.051e-2 | 2.142e-1 | 2.094e+0 |
| | | (1.695e+0) | (3.760e-1) | (2.307e-2) | (3.887e-2) | (1.299e-1) | (2.130e+0) |
| | RLLS | 7.962e+0 | 2.095e-1 | 2.311e-2 | 7.051e-2 | 2.116e-1 | 1.495e+0 |
| | | (6.015e-1) | (3.142e-1) | (2.307e-2) | (3.887e-2) | (1.240e-1) | (1.034e+0) |
| | MLLS | 2.373e-1 | 2.422e-2 | **3.798e-3** | 1.479e-2 | 3.421e-2 | 1.957e-1 |
| | | (2.154e-1) | (2.337e-2) | (**3.292e-3**) | (6.984e-3) | (1.415e-2) | (8.333e-2) |
| | CPMCN | 3.379e-1 | 4.451e-2 | 8.336e-3 | 2.845e-2 | 5.006e-2 | 1.733e-1 |
| | | (3.534e-1) | (4.285e-2) | (5.818e-3) | (1.534e-2) | (3.019e-2) | (8.857e-2) |
| | Ours | **1.530e-1** | **1.325e-2** | 4.883e-3 | **1.352e-2** | **3.033e-2** | **9.509e-2** |
| | | (**1.116e-1**) | (**9.597e-3**) | (3.378e-3) | (**5.247e-3**) | (**1.189e-2**) | (**3.734e-2**) |

*Table 12.* **Downstream Classification Performance on MNIST (Dirichlet Shift).** We evaluate the adaptation quality by reporting the ACC (%) and Macro-F1 (%) of the re-weighted classifiers. Results are shown for varying shift intensities $\alpha$. For each entry, the first row denotes the mean over 100 trials, and the second row (in parentheses) denotes the standard deviation.

| Sample size | Method | $\alpha = 0.1$ | | $\alpha = 0.5$ | | $\alpha = 1$ | |
| --- | --- | --- | --- | --- | --- | --- | --- |
| | | ACC(%) | Macro F1(%) | ACC(%) | Macro F1(%) | ACC(%) | Macro F1(%) |
| 1000 | BBSL | 30.17 | 18.40 | 52.42 | 42.28 | 63.77 | 56.62 |
| | | (0.10) | (0.09) | (0.10) | (0.12) | (0.09) | (0.11) |
| | RLLS | 30.86 | 19.04 | 54.27 | 44.71 | 65.69 | 59.49 |
| | | (0.10) | (0.10) | (0.10) | (0.12) | (0.08) | (0.10) |
| | MLLS | 36.92 | 30.23 | 64.89 | 60.63 | 75.83 | 73.67 |
| | | (0.11) | (0.11) | (0.10) | (0.11) | (0.07) | (0.09) |
| | CPMCN | 35.61 | 28.31 | 61.60 | 56.11 | 74.17 | 71.26 |
| | | (0.11) | (0.11) | (0.11) | (0.12) | (0.06) | (0.08) |
| | Ours | **43.46** | **35.96** | **67.38** | **62.79** | **77.68** | **75.59** |
| | | (0.09) | (0.09) | (0.08) | (0.09) | (0.05) | (0.06) |
| 5000 | BBSL | 78.07 | 74.37 | 90.96 | 89.78 | 94.84 | 94.74 |
| | | (0.08) | (0.10) | (0.05) | (0.07) | (0.01) | (0.01) |
| | RLLS | 80.15 | 77.50 | 92.23 | 91.73 | 95.44 | 95.45 |
| | | (0.06) | (0.08) | (0.03) | (0.04) | (0.01) | (0.01) |
| | MLLS | 83.72 | 81.97 | 94.47 | 94.13 | 95.46 | 95.47 |
| | | (0.05) | (0.06) | (0.02) | (0.02) | (0.01) | (0.01) |
| | CPMCN | 83.67 | 82.00 | 94.51 | 94.17 | 95.48 | 95.49 |
| | | (0.04) | (0.05) | (0.01) | (0.01) | (0.01) | (0.01) |
| | Ours | **89.27** | **89.20** | **94.80** | **94.80** | **95.51** | **95.52** |
| | | (0.03) | (0.03) | (0.01) | (0.01) | (0.01) | (0.01) |
| 10000 | BBSL | 91.86 | 91.28 | 96.09 | 96.06 | 96.95 | 96.96 |
| | | (0.06) | (0.08) | (0.01) | (0.01) | (0.01) | (0.01) |
| | RLLS | 93.11 | 92.92 | 96.18 | 96.18 | 96.95 | 96.96 |
| | | (0.03) | (0.03) | (0.01) | (0.01) | (0.01) | (0.01) |
| | MLLS | 94.15 | 94.16 | 96.36 | 96.37 | 96.98 | 96.99 |
| | | (0.01) | (0.01) | (0.01) | (0.01) | (0.00) | (0.00) |
| | CPMCN | 94.08 | 94.09 | 96.36 | 96.37 | 96.99 | 96.99 |
| | | (0.01) | (0.01) | (0.01) | (0.01) | (0.00) | (0.00) |
| | Ours | **94.26** | **94.27** | **96.40** | **96.41** | **97.00** | **97.01** |
| | | (0.01) | (0.01) | (0.01) | (0.01) | (0.00) | (0.00) |

*Table 13.* **Downstream Classification Performance on CIFAR-10 (Dirichlet Shift).** We evaluate the adaptation quality by reporting the ACC (%) and Macro-F1 (%) of the re-weighted classifiers. Results are shown for varying shift intensities $\alpha$. For each entry, the first row denotes the mean over 100 trials, and the second row (in parentheses) denotes the standard deviation.

| Sample size | Method | $\alpha = 0.1$ | | $\alpha = 0.5$ | | $\alpha = 1$ | |
|---|---|---|---|---|---|---|---|
| | | ACC(%) | Macro F1(%) | ACC(%) | Macro F1(%) | ACC(%) | Macro F1(%) |
| 1000 | BBSL | 27.83 (0.08) | 15.66 (0.08) | 43.92 (0.07) | 33.26 (0.08) | 50.24 (0.07) | 41.83 (0.05) |
| | RLLS | 28.63 (0.08) | 16.70 (0.08) | 46.24 (0.07) | 36.63 (0.09) | 52.51 (0.06) | 45.31 (0.07) |
| | MLLS | 33.70 (0.10) | 25.04 (0.10) | 56.48 (0.08) | 50.41 (0.09) | 62.79 (0.05) | 58.39 (0.06) |
| | CPMCN | 32.47 (0.09) | 23.71 (0.09) | 52.16 (0.07) | 45.41 (0.08) | 58.82 (0.05) | 53.95 (0.06) |
| | Ours | **35.21** (0.10) | **27.91** (0.10) | **58.90** (0.08) | **53.66** (0.09) | **65.13** (0.05) | **61.49** (0.06) |
| 5000 | BBSL | 61.02 (0.09) | 58.93 (0.10) | 75.61 (0.03) | 74.88 (0.04) | 79.11 (0.02) | 78.67 (0.02) |
| | RLLS | 65.08 (0.08) | 63.92 (0.09) | 77.86 (0.03) | 77.44 (0.03) | 81.29 (0.02) | 80.94 (0.02) |
| | MLLS | 70.74 (0.06) | 70.44 (0.07) | 81.68 (0.02) | 81.55 (0.02) | 83.65 (0.01) | 83.57 (0.01) |
| | CPMCN | 69.18 (0.06) | 68.70 (0.07) | 81.09 (0.02) | 80.95 (0.02) | 83.54 (0.01) | 83.46 (0.01) |
| | Ours | **71.87** (0.06) | **71.66** (0.07) | **82.19** (0.02) | **82.10** (0.02) | **84.06** (0.01) | **83.99** (0.01) |
| 10000 | BBSL | 72.86 (0.08) | 71.84 (0.10) | 81.86 (0.03) | 81.49 (0.04) | 83.74 (0.01) | 83.53 (0.02) |
| | RLLS | 76.09 (0.03) | 74.87 (0.05) | 82.87 (0.02) | 82.56 (0.02) | 84.71 (0.01) | 84.61 (0.01) |
| | MLLS | **79.66** (0.02) | **79.46** (0.02) | 83.94 (0.01) | 83.87 (0.01) | 85.01 (0.01) | 84.97 (0.01) |
| | CPMCN | 78.78 (0.02) | 78.46 (0.02) | 83.74 (0.01) | 83.64 (0.01) | 85.02 (0.01) | 84.98 (0.01) |
| | Ours | 79.64 (0.02) | 79.41 (0.02) | **84.02** (0.01) | **83.97** (0.01) | **85.06** (0.01) | **85.02** (0.01) |

*Table 14.* **Downstream Classification Performance on CIFAR-100 (Dirichlet Shift).** We evaluate the adaptation quality by reporting the ACC (%) and Macro-F1 (%) of the re-weighted classifiers. Results are shown for varying shift intensities $\alpha$. For each entry, the first row denotes the mean over 100 trials, and the second row (in parentheses) denotes the standard deviation.

| Sample size | Method | $\alpha = 0.1$ | | $\alpha = 0.5$ | | $\alpha = 1$ | |
|---|---|---|---|---|---|---|---|
| | | ACC(%) | Macro F1(%) | ACC(%) | Macro F1(%) | ACC(%) | Macro F1(%) |
| 5000 | BBSL | 14.98 | 7.36 | 27.02 | 18.88 | 32.75 | 25.34 |
| | | (0.02) | (0.02) | (0.03) | (0.03) | (0.03) | (0.03) |
| | RLLS | 17.64 | 9.99 | 32.92 | 25.46 | 39.61 | 33.49 |
| | | (0.02) | (0.02) | (0.02) | (0.02) | (0.02) | (0.02) |
| | MLLS | 15.56 | 11.17 | 17.36 | 12.84 | 24.73 | 20.39 |
| | | (0.03) | (0.02) | (0.04) | (0.03) | (0.04) | (0.04) |
| | CPMCN | 18.97 | 11.24 | 31.17 | 23.68 | 38.83 | 32.37 |
| | | (0.03) | (0.02) | (0.03) | (0.03) | (0.03) | (0.03) |
| | Ours | **22.94** | **16.06** | **39.80** | **33.76** | **46.62** | **41.65** |
| | | (0.02) | (0.02) | (0.02) | (0.02) | (0.02) | (0.02) |
| 10000 | BBSL | 19.38 | 10.28 | 31.52 | 23.36 | 37.60 | 29.80 |
| | | (0.02) | (0.02) | (0.02) | (0.02) | (0.02) | (0.02) |
| | RLLS | 23.75 | 15.24 | 38.01 | 31.14 | 44.07 | 38.56 |
| | | (0.02) | (0.02) | (0.02) | (0.01) | (0.01) | (0.01) |
| | MLLS | 18.35 | 13.56 | 19.38 | 14.93 | 26.69 | 22.30 |
| | | (0.03) | (0.03) | (0.04) | (0.02) | (0.03) | (0.03) |
| | CPMCN | 24.96 | 16.29 | 37.83 | 30.68 | 43.68 | 37.79 |
| | | (0.01) | (0.01) | (0.01) | (0.01) | (0.01) | (0.01) |
| | Ours | **33.99** | **27.96** | **51.81** | **47.90** | **57.65** | **54.45** |
| | | (0.02) | (0.02) | (0.02) | (0.02) | (0.01) | (0.01) |

*Table 15.* Performance comparison of RDALS against baselines on CIFAR-10-C. We report the Mean Squared Error (MSE) for weight estimation and Accuracy/Macro-F1 for downstream classification. All results are averaged over 100 independent trials.

| Sample Size | Method | $\alpha = 0.1$ | | | $\alpha = 0.5$ | | | $\alpha = 1$ | | |
|---|---|---|---|---|---|---|---|---|---|---|
| | | MSE | ACC(%) | Macro F1(%) | MSE | ACC(%) | Macro F1(%) | MSE | ACC(%) | Macro F1(%) |
| 1000 | BBSL | 5.088e+1 (1.007e+1) | 25.69 (0.06) | 14.28 (0.06) | 1.676e+1 (7.055e+0) | 37.12 (0.06) | 27.05 (0.07) | 6.379e+0 (3.759e+0) | 41.91 (0.07) | 33.75 (0.08) |
| | RLLS | 4.118e+1 (8.199e+0) | 26.60 (0.06) | 15.61 (0.06) | 1.309e+1 (5.732e+0) | 38.82 (0.06) | 29.77 (0.07) | 4.729e+0 (2.967e+0) | 43.81 (0.06) | 36.95 (0.08) |
| | MLLS | 1.352e+2 (7.308e+1) | 19.15 (0.06) | 13.11 (0.06) | 6.251e+1 (4.780e+1) | 29.41 (0.08) | 25.40 (0.09) | 2.343e+1 (2.637e+1) | 37.52 (0.08) | 33.66 (0.09) |
| | CPMCN | **2.199e+1** (1.262e+1) | 24.80 (0.07) | 18.49 (0.08) | 1.247e+1 (7.048e+0) | 34.08 (0.07) | 28.66 (0.08) | 7.278e+0 (5.724e+0) | 39.54 (0.07) | 34.26 (0.08) |
| | RDALS | 2.463e+1 (9.329e+0) | **31.89** (0.05) | **24.53** (0.06) | **8.250e+0** (5.640e+0) | **42.04** (0.05) | **36.13** (0.06) | **2.829e+0** (2.494e+0) | **47.44** (0.05) | **42.82** (0.06) |
| 5000 | BBSL | 4.121e+1 (1.399e+1) | 45.97 (0.07) | 37.16 (0.09) | 1.224e+1 (6.188e+0) | 62.74 (0.06) | 59.20 (0.07) | 4.314e+0 (4.312e+0) | 68.76 (0.05) | 67.13 (0.07) |
| | RLLS | 3.346e+1 (9.238e+0) | 50.68 (0.06) | 44.88 (0.08) | 1.064e+1 (5.348e+0) | 65.39 (0.04) | 63.42 (0.05) | 3.354e+0 (3.379e+0) | 69.81 (0.04) | 68.71 (0.06) |
| | MLLS | 5.505e+1 (3.698e+1) | 50.96 (0.07) | 47.93 (0.09) | 3.643e+1 (3.169e+1) | 62.11 (0.07) | 61.04 (0.09) | 1.108e+1 (1.529e+1) | 68.00 (0.05) | 67.83 (0.06) |
| | CPMCN | 2.126e+1 (1.435e+1) | **57.15** (0.07) | **55.39** (0.09) | 1.104e+1 (8.470e+0) | 62.98 (0.08) | 61.00 (0.11) | 6.946e+0 (1.072e+1) | 68.15 (0.05) | 67.52 (0.06) |
| | RDALS | **2.069e+1** (1.025e+1) | 56.29 (0.05) | 52.76 (0.07) | **4.373e+0** (3.515e+0) | **67.70** (0.03) | **66.92** (0.04) | **1.516e+0** (1.230e+0) | **70.22** (0.02) | **69.90** (0.03) |
| 10000 | BBSL | 3.263e+1 (2.276e+1) | 60.31 (0.07) | 56.38 (0.09) | 8.056e+0 (6.573e+0) | 70.71 (0.05) | 69.46 (0.06) | 1.385e+0 (1.622e+0) | 75.67 (0.02) | 75.62 (0.02) |
| | RLLS | 2.193e+1 (1.091e+1) | 63.55 (0.05) | 61.40 (0.06) | 6.819e+0 (5.278e+0) | 71.90 (0.04) | 71.25 (0.05) | 1.341e+0 (1.489e+0) | **75.71** (0.02) | **75.66** (0.02) |
| | MLLS | 3.210e+1 (1.722e+1) | 64.00 (0.05) | 62.87 (0.06) | 1.648e+1 (1.389e+1) | 70.63 (0.05) | 70.48 (0.05) | 4.108e+0 (4.726e+0) | 74.47 (0.03) | 74.50 (0.03) |
| | CPMCN | 1.958e+1 (1.597e+1) | **66.92** (0.05) | **66.27** (0.06) | 1.009e+1 (8.478e+0) | 71.11 (0.04) | 70.67 (0.06) | 3.046e+0 (3.018e+0) | 74.36 (0.03) | 74.16 (0.04) |
| | RDALS | **1.685e+1** (9.960e+0) | 65.10 (0.05) | 63.18 (0.07) | **3.869e+0** (2.669e+0) | **72.76** (0.02) | **72.71** (0.02) | **1.216e+0** (8.653e-1) | 74.59 (0.02) | 74.60 (0.02) |

## B. Technical Proofs

This section consists of two conceptual parts. In the first part, we show that, within an appropriate class of linear discriminative functions on the representation space and under a normalization that removes scale effects, maximizing the numerical stability of the transformed moment system is achieved by the multiclass LDA subspace. This establishes LDA as the optimal choice of linear moments for label-shift estimation. In the second part, we derive prediction error bounds by controlling the deviations of the estimated moment matrix and moment vector from their population counterparts, propagating these deviations through the stability of the transformed linear system to obtain an explicit bound on $\|\hat{q} - q^*\|$ and hence $\|\hat{w} - w^*\|_\infty$. We then plug the resulting weight-estimation error into a standard excess-risk decomposition to conclude a generalization bound. The following figure 7 provides a proof roadmap of prediction error bounds and the generalization bound, illustrating the logical dependencies among the key assumptions, intermediate propositions, and main results.

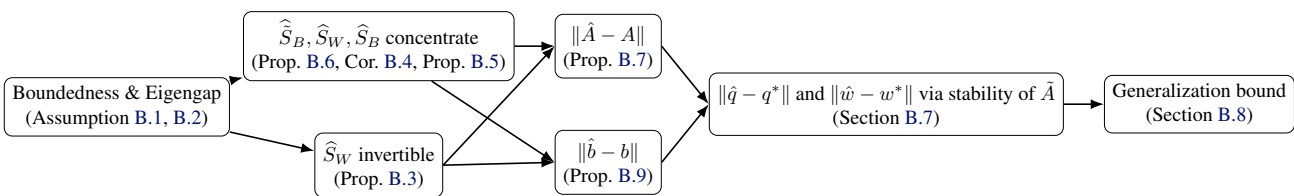

*Figure 7.* Road map of the non-asymptotic analysis.

### B.1. Assumptions

We now introduce two regularity conditions.

**Assumption B.1** (Boundedness). Assume that the representation vector $Z \in \mathbb{R}^d$ satisfies $\|Z\|_2 \leq R_Z$ almost surely. Moreover, there exists a constant $c_W > 0$ such that the within-class scatter matrix $S_W = \sum_{j=1}^K p_j \Sigma_j$ satisfies $\lambda_{\min}(S_W) \geq c_W > 0$.

The boundedness condition $\|Z\|_2 \leq R_Z$ is a standard regularity assumption in non-asymptotic analysis. The lower eigenvalue bound $\lambda_{\min}(S_W) \geq c_W > 0$ enforces non-degeneracy of the within-class scatter, guaranteeing that matrix $S_W^{-1/2}$ is well-defined.

**Assumption B.2** (Eigengap). Let $\tilde{S}_B := S_W^{-1/2} S_B S_W^{-1/2}$ and $\widehat{\tilde{S}}_B := \widehat{S}_W^{-1/2} \widehat{S}_B \widehat{S}_W^{-1/2}$. Assume an eigengap condition

$$\lambda_1(\tilde{S}_B) > \lambda_2(\tilde{S}_B) > ... > \lambda_K(\tilde{S}_B) = 0. \tag{5}$$

Moreover, let $\gamma := \lambda_{K-1}(\tilde{S}_B) - \lambda_K(\tilde{S}_B)$.

Assumption B.2 requires a spectral gap to enable Davis–Kahan perturbation bounds for controlling the projection error $\|\hat{W} - W\|_{2,*}$.

### B.2. Optimality of Linear Discriminative Functions for Label Shift

Let $p = (p_1, \ldots, p_K)^\top$ with $p_j > 0$ and $\sum_{j=1}^K p_j = 1$, and let $D = \mathrm{diag}(p)$. Define $v = D^{1/2} \mathbf{1}_K$. Since $\|v\|_2^2 = \mathbf{1}_K^\top D \mathbf{1}_K = 1$, $v$ is a unit vector. Choose any matrix $C_p \in \mathbb{R}^{K \times (K-1)}$ whose columns form an orthonormal basis of the orthogonal complement of $v$, so that

$$v^\top C_p = 0^\top, \qquad C_p^\top C_p = I_{K-1}.$$

For any $q \in \Delta^{K-1}$, define

$$\tilde{q} := C_p^\top D^{-1/2}(q - p).$$

Since $\mathbf{1}_K^\top (q - p) = 0$, we have

$$v^\top D^{-1/2}(q - p) = \mathbf{1}_K^\top (q - p) = 0.$$

Thus $D^{-1/2}(q - p)$ lies in the column space of $C_p$, and hence

$$D^{-1/2}(q - p) = C_p \tilde{q}.$$

Equivalently,

$$q = p + D^{1/2} C_p \tilde{q}.$$

This gives a one-to-one parametrization of the simplex hyperplane in contrast coordinates. Substituting this expression into $b = Aq$ yields

$$b - Ap = AD^{1/2} C_p \tilde{q}.$$

Therefore, with

$$\tilde{b} := b - Ap, \qquad \tilde{A} := AD^{1/2} C_p,$$

the original constrained system is equivalent to the square contrast system

$$\tilde{b} = \tilde{A}\tilde{q}.$$

Once we focus on the linear functions defined on the representation space $\mathcal{Z}$:

$$h_m(z) = w_m^\top z, \quad m = 1, \dots, K - 1,$$

and let

$$W := [w_1, \dots, w_{K-1}] \in \mathbb{R}^{d \times (K-1)},$$

and $h(z) := W^\top z \in \mathbb{R}^{K-1}$. Combining with the previous notations, we have $\tilde{A} = W^\top \mu D^{1/2} C_p$. Let $\mu := [\mu_1, \dots, \mu_K] \in \mathbb{R}^{d \times K}$, then we have

$$\begin{aligned}
\tilde{A}\tilde{A}^\top &= W^\top \left( \mu D^{1/2} C_p C_p^\top D^{1/2} \mu^\top \right) W \\
&= W^\top \mu \left( D - pp^\top \right) \mu^\top W \\
&= W^\top S_B W,
\end{aligned}$$

where the last equality holds since

$$S_B = \mu D \mu^\top - \mu pp^\top \mu^\top.$$

Hence the smallest singular value of $\tilde{A}$ is directly related to the smallest eigenvalue of the between-class divergence matrix $S_B$ in the subspace spanned by columns of $W$. However, the eigenvalues of $\tilde{A}\tilde{A}^\top$ depends on the scale of $W$. Note that $S_W$ is positive definite. To eliminate the effect of scale, we consider the following normalized feasibility matrix:

$$\mathcal{W} := \{W \in \mathbb{R}^{d \times (K-1)} : W^\top S_W W = I_{K-1}\}.$$

By letting $V := S_W^{1/2} W$ and $\tilde{S}_B = S_W^{-1/2} S_B S_W^{-1/2}$, then the problem of maximizing the numerical stability of the linear system can be formulated as

$$\begin{aligned}
\arg\max_{W \in \mathcal{W}} \sigma_{\min}(\tilde{A})^2 &= \arg\max_{W \in \mathcal{W}} \lambda_{\min}\left(W^\top S_B W\right) \\
&= \arg\max_{V^\top V = I_{K-1}} \lambda_{\min}\left(V^\top \tilde{S}_B V\right),
\end{aligned} \tag{6}$$

where $\sigma_{\min}$ denotes the smallest singular value and $\lambda_{\min}$ denotes the smallest eigenvalue. The rightmost term in Eq. (6) indicates that the optimal value is the $(K-1)$-th largest eigenvalue of $\tilde{S}_B$ and the optimal $V$ is spanned by the top $(K-1)$ eigenvectors of $\tilde{S}_B$ by the Courant-Fischer min-max theorem. Namely,

$$\max_{W \in \mathcal{W}} \sigma_{\min}(\tilde{A})^2 = \lambda_{K-1}(\tilde{S}_B),$$

and the optimal $W^*$ satisfies

$$\text{col}(W^*) = \text{col}(S_W^{-1/2}U),$$

where $U = [u_1, \ldots, u_{K-1}] \in \mathbb{R}^{d \times (K-1)}$ with $u_i$ being the eigenvector of $\tilde{S}_B$ corresponding to its $i$-th largest eigenvalue $\lambda_i(\tilde{S}_B)$, $i = 1, \ldots, K-1$. Equivalently, the columns of $W^*$ span the top $(K-1)$ generalized eigenspace of $S_B u = \lambda S_W u$, i.e., the multiclass LDA subspace.

## B.3. Invertibility of $\widehat{S}_W$

The concentration analysis for $A$ and $b$ conditions on the event $\mathcal{E}_W := \{\lambda_{\min}(\widehat{S}_W) \geq c_W/2\}$. This followong proposition provides a lower bound for $\lambda_{\min}(\widehat{S}_W)$ under Assumption B.1. Throughout the analysis we treat the source label marginal $p \in \Delta^{K-1}$, where $p_j := P_y(y = j)$, as a fixed quantity and takes values $n_{S,j}/n_S$, $j \in [K]$.

Note that sample splitting is *not* required by the method itself. In practice we use the full source sample $S$ for both steps, which typically reduces variance and improves empirical performance. While the resulting estimators are no longer exactly independent, the dependence arises only through the shared data and is expected to be weak in large samples. Consequently, we view the sample-splitting analysis as providing a conservative guarantee that captures the dominant stability mechanism (via $\sigma_{\min}(\tilde{A})$), whereas the non-splitting implementation is a natural plug-in version.

**Proposition B.3.** *Suppose Assumption B.1 holds. Let $I_1$ denote the index set of the source split $S_1$ and define*

$$I_{1,j} := \{i \in I_1 : y_i = j\}, \qquad n_{1,j} := |I_{1,j}|, \qquad n_{1,\min} := \min_{j \in [K]} n_{1,j}.$$

*Let $p_j = P_y(y = j)$ and define the empirical within-class scatter $\widehat{S}_W := \sum_{j=1}^K p_j \widehat{\Sigma}_j$, where*

$$\widehat{\mu}_j := \frac{1}{n_{1,j}} \sum_{i \in I_{1,j}} Z_i, \qquad \widehat{\Sigma}_j := \frac{1}{n_{1,j}} \sum_{i \in I_{1,j}} (Z_i - \widehat{\mu}_j)(Z_i - \widehat{\mu}_j)^\top.$$

*Fix $\delta \in (0,1)$ and set*

$$L := \log\left(\frac{4k(d+1)}{\delta}\right).$$

*If $n_{1,\min}$ is large enough such that*

$$8R_Z^2 \sqrt{\frac{2L}{n_{1,\min}}} + \frac{16R_Z^2}{3} \frac{L}{n_{1,\min}} + 4R_Z^2 \left(\sqrt{\frac{2L}{n_{1,\min}}} + \frac{2}{3}\frac{L}{n_{1,\min}}\right)^2 \leq \frac{c_W}{2}, \tag{7}$$

*then with probability at least $1 - \delta$,*

$$\lambda_{\min}(\widehat{S}_W) \geq \frac{c_W}{2}.$$

*Proof.* We complete our proof by the following steps. *Using Weyl's inequality to get an upper bound.* Recall $S_W = \sum_{j=1}^K p_j \Sigma_j$ and $\widehat{S}_W = \sum_{j=1}^K p_j \widehat{\Sigma}_j$. By Weyl's inequality,

$$|\lambda_{\min}(\widehat{S}_W) - \lambda_{\min}(S_W)| \leq \|\widehat{S}_W - S_W\|_{2,*}. \tag{8}$$

Therefore, since $\lambda_{\min}(S_W) \geq c_W$ by Assumption B.1, it suffices to prove

$$\|\widehat{S}_W - S_W\|_{2,*} \leq \frac{c_W}{2}. \tag{9}$$

Using the triangle inequality and $\sum_{j=1}^K p_j = 1$,

$$\|\widehat{S}_W - S_W\|_{2,*} = \left\|\sum_{j=1}^K p_j(\widehat{\Sigma}_j - \Sigma_j)\right\|_{2,*} \leq \sum_{j=1}^K p_j \|\widehat{\Sigma}_j - \Sigma_j\|_{2,*} \leq \max_{j \in [K]} \|\widehat{\Sigma}_j - \Sigma_j\|_{2,*}. \tag{10}$$

Hence it suffices to control $\|\widehat{\Sigma}_j - \Sigma_j\|_{2,*}$ uniformly over $j$.

*Decomposition of $\widehat{\Sigma}_j - \Sigma_j$.* Fix a class $j \in [K]$ and write $n := n_{1,j}$. Let $\mu_j := \mathbb{E}[Z \mid Y = j]$ and $\Sigma_j := \mathbb{E}[(Z - \mu_j)(Z - \mu_j)^\top \mid Y = j]$. Define

$$\bar{\Sigma}_j := \frac{1}{n} \sum_{i \in I_{1,j}} (Z_i - \mu_j)(Z_i - \mu_j)^\top.$$

Expanding $(Z_i - \widehat{\mu}_j) = (Z_i - \mu_j) - (\widehat{\mu}_j - \mu_j)$ and using $\frac{1}{n} \sum_{i \in I_{1,j}} (Z_i - \mu_j) = \widehat{\mu}_j - \mu_j$, we obtain the exact identity

$$\widehat{\Sigma}_j = \bar{\Sigma}_j - (\widehat{\mu}_j - \mu_j)(\widehat{\mu}_j - \mu_j)^\top. \tag{11}$$

Subtracting $\Sigma_j$ from both sides yields

$$\widehat{\Sigma}_j - \Sigma_j = (\bar{\Sigma}_j - \Sigma_j) - (\widehat{\mu}_j - \mu_j)(\widehat{\mu}_j - \mu_j)^\top, \tag{12}$$

and hence, by the triangle inequality,

$$\|\widehat{\Sigma}_j - \Sigma_j\|_{2,*} \leq \|\bar{\Sigma}_j - \Sigma_j\|_{2,*} + \|\widehat{\mu}_j - \mu_j\|_2^2. \tag{13}$$

*Concentration of $\|\bar{\Sigma}_j - \Sigma_j\|_{2,*}$.* Conditional on $\{Y = j\}$, the vectors $\{Z_i\}_{i \in I_{1,j}}$ are i.i.d. Let

$$Y_i := (Z_i - \mu_j)(Z_i - \mu_j)^\top - \Sigma_j \in \mathbb{R}^{d \times d}, \qquad i \in I_{1,j}.$$

Then $\mathbb{E}[Y_i \mid Y = j] = 0$. Moreover, by Assumption B.1, $\|Z_i - \mu_j\|_2 \leq \|Z_i\|_2 + \|\mu_j\|_2 \leq R_Z + \mathbb{E}[\|Z\|_2 \mid Y = j] \leq 2R_Z$ almost surely, hence

$$\|(Z_i - \mu_j)(Z_i - \mu_j)^\top\|_{2,*} = \|Z_i - \mu_j\|_2^2 \leq 4R_Z^2. \tag{14}$$

Also, since $\Sigma_j = \mathbb{E}[(Z - \mu_j)(Z - \mu_j)^\top \mid Y = j]$ and $\|(Z - \mu_j)(Z - \mu_j)^\top\|_{2,*} \leq 4R_Z^2$ almost surely, we have

$$\|\Sigma_j\|_{2,*} \leq \mathbb{E}[\|(Z - \mu_j)(Z - \mu_j)^\top\|_{2,*} \mid Y = j] \leq 4R_Z^2. \tag{15}$$

Combining (14)–(15) gives a uniform bound

$$\|Y_i\|_{2,*} \leq \|(Z_i - \mu_j)(Z_i - \mu_j)^\top\|_{2,*} + \|\Sigma_j\|_{2,*} \leq 8R_Z^2. \tag{16}$$

For the variance proxy, using $\|\mathbb{E}[Y_i^2 \mid Y = j]\|_{2,*} \leq \mathbb{E}[\|Y_i\|_{2,*}^2 \mid Y = j]$ and (16),

$$\left\| \sum_{i \in I_{1,j}} \mathbb{E}[Y_i^2 \mid Y = j] \right\|_{2,*} \leq \sum_{i \in I_{1,j}} \mathbb{E}[\|Y_i\|_{2,*}^2 \mid Y = j] \leq 64nR_Z^4. \tag{17}$$

Apply Theorem 1.4 of Tropp (2012) to the self-adjoint sum $\sum_{i \in I_{1,j}} Y_i$ in dimension $d$. For any $\eta \in (0,1)$, with probability at least $1 - \eta$,

$$\|\bar{\Sigma}_j - \Sigma_j\|_{2,*} = \left\| \frac{1}{n} \sum_{i \in I_{1,j}} Y_i \right\|_{2,*} \leq 8R_Z^2 \sqrt{\frac{2\log(2d/\eta)}{n}} + \frac{8R_Z^2}{3} \frac{\log(2d/\eta)}{n}. \tag{18}$$

*Concentration of $\|\widehat{\mu}_j - \mu_j\|_2$.* Define $\xi_i := Z_i - \mu_j \in \mathbb{R}^d$ for $i \in I_{1,j}$. Then $\mathbb{E}[\xi_i \mid Y = j] = 0$ and $\|\xi_i\|_2 \leq 2R_Z$ almost surely. Consider the self-adjoint dilation

$$\tilde{\xi}_i := \begin{bmatrix} 0 & \xi_i^\top \\ \xi_i & 0 \end{bmatrix} \in \mathbb{R}^{(d+1) \times (d+1)}.$$

Then $\|\tilde{\xi}_i\|_{2,*} = \|\xi_i\|_2 \leq 2R_Z$ and $\mathbb{E}[\tilde{\xi}_i \mid Y = j] = 0$. Moreover, $\tilde{\xi}_i^2 = \mathrm{diag}(\xi_i^\top \xi_i, \xi_i \xi_i^\top)$ implies $\|\mathbb{E}[\tilde{\xi}_i^2 \mid Y = j]\|_{2,*} \leq \mathbb{E}[\|\xi_i\|_2^2 \mid Y = j] \leq 4R_Z^2$. Applying Theorem 1.4 of Tropp (2012) in dimension $d + 1$ yields: for any $\eta \in (0,1)$, with probability at least $1 - \eta$,

$$\|\widehat{\mu}_j - \mu_j\|_2 = \left\| \frac{1}{n} \sum_{i \in I_{1,j}} \xi_i \right\|_2 \leq 2R_Z \sqrt{\frac{2\log(2(d+1)/\eta)}{n}} + \frac{2R_Z}{3} \frac{\log(2(d+1)/\eta)}{n}. \tag{19}$$

*Uniform control over classes.* Set $\eta := \delta/K$. By a union bound over $j \in [K]$, with probability at least $1 - \delta$, the bounds (18) and (19) hold simultaneously for all classes. Using $n \geq n_{1,\min}$ and $\log(2d/\eta) \leq \log(4k(d+1)/\delta) = L$ and $\log(2(d+1)/\eta) \leq L$, we obtain for all $j \in [K]$,

$$\|\bar{\Sigma}_j - \Sigma_j\|_{2,*} \leq 8R_Z^2 \sqrt{\frac{2L}{n_{1,\min}}} + \frac{8R_Z^2}{3} \frac{L}{n_{1,\min}},$$

and

$$\|\widehat{\mu}_j - \mu_j\|_2 \leq 2R_Z \left( \sqrt{\frac{2L}{n_{1,\min}}} + \frac{L}{3n_{1,\min}} \right).$$

Hence, by (10),

$$\|\widehat{S}_W - S_W\|_{2,*} \leq \max_{j \in [K]} \|\widehat{\Sigma}_j - \Sigma_j\|_{2,*} \leq 8R_Z^2 \sqrt{\frac{2L}{n_{1,\min}}} + \frac{16R_Z^2}{3} \frac{L}{n_{1,\min}} + 4R_Z^2 \left( \sqrt{\frac{2L}{n_{1,\min}}} + \frac{2}{3} \frac{L}{n_{1,\min}} \right)^2.$$

Under the sample size condition (7), the right-hand side is at most $c_W/2$, which verifies (9). Finally, combining (8) and (9) yields $\lambda_{\min}(\widehat{S}_W) \geq \lambda_{\min}(S_W) - \|\widehat{S}_W - S_W\|_{2,*} \geq c_W - c_W/2 = c_W/2$.

$\square$

## B.4. Concentration of $\widehat{S}_W$ and $\widehat{\widehat{S}}_B$

For later use, define the deviation level for the within-class scatter

$$\Delta_{S_W}(\delta) := 8R_Z^2 \sqrt{\frac{2L}{n_{1,\min}}} + \frac{8R_Z^2}{3} \frac{L}{n_{1,\min}} + 4R_Z^2 \left( \sqrt{\frac{2L}{n_{1,\min}}} + \frac{L}{3n_{1,\min}} \right)^2. \tag{20}$$

Also define the explicit deviation level for the source-split class means

$$\Delta_\mu(\delta) := 2R_Z \left( \sqrt{\frac{2L}{n_{1,\min}}} + \frac{L}{3n_{1,\min}} \right), \qquad L := \log\left( \frac{4k(d+1)}{\delta} \right). \tag{21}$$

**Corollary B.4** (Concentration of $\widehat{S}_W$). *Under Assumption B.1, with probability at least $1 - \delta$,*

$$\|\widehat{S}_W - S_W\|_{2,*} \leq \Delta_{S_W}(\delta). \tag{22}$$

*Proof.* This is exactly the bound established at the end of the proof of Proposition B.3. $\square$

**Proposition B.5** (Concentration of $\widehat{S}_B$). *Suppose Assumption B.1 holds and define*

$$\widehat{S}_B := \sum_{j=1}^{K} p_j (\widehat{\mu}_j - \widehat{\mu})(\widehat{\mu}_j - \widehat{\mu})^\top, \qquad \widehat{\mu} := \sum_{j=1}^{K} p_j \widehat{\mu}_j.$$

*Then with probability at least $1 - \delta$,*

$$\|\widehat{S}_B - S_B\|_{2,*} \leq 8R_Z \Delta_\mu(\delta) + 4\Delta_\mu(\delta)^2. \tag{23}$$

*Proof.* By the union-bound step used in Proposition B.3 with $\eta = \delta/K$, we have with probability at least $1 - \delta$ that

$$\max_{j \in [K]} \|\widehat{\mu}_j - \mu_j\|_2 \leq \Delta_\mu(\delta). \tag{24}$$

Work on the event (24). Define

$$\bar{\mu} := \sum_{j=1}^{K} p_j \mu_j, \qquad a_j := \mu_j - \bar{\mu}, \qquad \widehat{a}_j := \widehat{\mu}_j - \widehat{\mu}.$$

Let $\delta_{\mu_j} := \widehat{\mu}_j - \mu_j$ and note that $\widehat{\bar{\mu}} - \bar{\mu} = \sum_{j=1}^{K} p_j\,\delta_{\mu_j}$, then

$$\|\widehat{\bar{\mu}} - \bar{\mu}\|_2 \le \max_{j\in[K]} \|\delta_{\mu_j}\|_2 \le \Delta_\mu(\delta).$$

Hence, for each $j$,

$$\widehat{a}_j - a_j = (\widehat{\mu}_j - \mu_j) - (\widehat{\bar{\mu}} - \bar{\mu}) = \delta_{\mu_j} - (\widehat{\bar{\mu}} - \bar{\mu}), \quad \text{so} \quad \|\widehat{a}_j - a_j\|_2 \le \|\delta_{\mu_j}\|_2 + \|\widehat{\bar{\mu}} - \bar{\mu}\|_2 \le 2\Delta_\mu(\delta).$$

Moreover, Assumption B.1 implies $\|\mu_j\|_2 \le R_Z$ for all $j$, and thus

$$\|a_j\|_2 = \|\mu_j - \bar{\mu}\|_2 \le \|\mu_j\|_2 + \|\bar{\mu}\|_2 \le R_Z + \sum_{\ell=1}^{K} p_\ell \|\mu_\ell\|_2 \le 2R_Z.$$

Using $xx^\top - yy^\top = (x-y)y^\top + y(x-y)^\top + (x-y)(x-y)^\top$, we get

$$\|\widehat{a}_j\widehat{a}_j^\top - a_j a_j^\top\|_{2,*} \le 2\|a_j\|_2\,\|\widehat{a}_j - a_j\|_2 + \|\widehat{a}_j - a_j\|_2^2 \le 2\cdot(2R_Z)\cdot(2\Delta_\mu(\delta)) + (2\Delta_\mu(\delta))^2 = 8R_Z\Delta_\mu(\delta) + 4\Delta_\mu(\delta)^2.$$

Finally,

$$\|\widehat{S}_B - S_B\|_{2,*} = \Big\| \sum_{j=1}^{K} p_j\left(\widehat{a}_j\widehat{a}_j^\top - a_j a_j^\top\right) \Big\|_{2,*} \le \sum_{j=1}^{K} p_j \|\widehat{a}_j\widehat{a}_j^\top - a_j a_j^\top\|_{2,*} \le 8R_Z\Delta_\mu(\delta) + 4\Delta_\mu(\delta)^2,$$

which proves (23) on (24). Since (24) holds with probability at least $1 - \delta$, the proposition follows. $\qquad\square$

## B.5. Concentration inequalities for $A$

Recall that the population moment matrix is

$$A_{mj} = \mathbb{E}_P\big[h_m(Z) \mid Y = j\big], \qquad m = 1, \ldots, K-1,\ j \in [K],$$

where $h(z) = W^\top z$ and $W = S_W^{-1/2}U$ is the (population) LDA projection. In the algorithm, we estimate $W$ using the split $S_1$ and then form linear moments $\hat{h}(z) = \hat{W}^\top z$. To separate the sampling fluctuations from the projection error, define the *ideal (conditional) moment matrix*

$$\check{A}_{mj} := \mathbb{E}_P\big[\hat{h}_m(Z) \mid Y = j\big], \quad m = 1, \ldots, K-1,\ j \in [K]. \tag{25}$$

Then the total error admits the exact decomposition

$$\|\hat{A} - A\|_{2,*} \le \|\hat{A} - \check{A}\|_{2,*} + \|\check{A} - A\|_{2,*}. \tag{26}$$

*Notation and conditioning.* Let $S_1$ and $S_2$ be the source splits, and write

$$\mathcal{F}_1 := \sigma(S_1)$$

for the sigma-field generated by $S_1$. The random matrix $\hat{W}$ is measurable with respect to $\mathcal{F}_1$. Moreover, conditional on $\mathcal{F}_1$, the samples in $S_2$ remain i.i.d. within each class. Let

$$n_{2,j} := \#\{i \in I_2 : y_i = j\}, \qquad n_{2,\min} := \min_{j\in[K]} n_{2,j}.$$

*Term $\|\hat{A} - \check{A}\|_{2,*}$.* By definition,

$$\hat{A}_{:j} - \check{A}_{:j} = \frac{1}{n_{2,j}} \sum_{i\in I_2:y_i=j} \left(\hat{h}(Z_i) - \mathbb{E}_P[\hat{h}(Z) \mid Y = j]\right) \in \mathbb{R}^{K-1}, \qquad j \in [K], \tag{27}$$

where $\hat{A}_{:j}$ denotes the $j$-th column of $\hat{A}$. Equivalently,

$$\hat{A} - \check{A} = \sum_{j=1}^{K} \sum_{i \in I_2 : y_i = j} X_i^{(j)}, \qquad X_i^{(j)} := \frac{1}{n_{2,j}} \left( \hat{h}(Z_i) - \mathbb{E}_P[\hat{h}(Z) \mid Y = j] \right) e_j^\top \in \mathbb{R}^{(K-1) \times K}. \tag{28}$$

Conditional on $\mathcal{F}_1$, the collection $\{X_i^{(j)}\}$ is independent and satisfies

$$\mathbb{E}\big[X_i^{(j)} \mid \mathcal{F}_1\big] = 0. \tag{29}$$

To apply a self-adjoint matrix concentration inequality, define the dilation

$$\tilde{X}_i^{(j)} := \begin{bmatrix} 0 & X_i^{(j)} \\ (X_i^{(j)})^\top & 0 \end{bmatrix} \in \mathbb{R}^{(2k-1) \times (2k-1)}.$$

Then $\tilde{X}_i^{(j)}$ is self-adjoint, and (by direct multiplication)

$$(\tilde{X}_i^{(j)})^2 = \begin{bmatrix} X_i^{(j)}(X_i^{(j)})^\top & 0 \\ 0 & (X_i^{(j)})^\top X_i^{(j)} \end{bmatrix}, \qquad \|\tilde{X}_i^{(j)}\|_{2,*} = \|X_i^{(j)}\|_{2,*}.$$

Moreover,

$$\|\hat{A} - \check{A}\|_{2,*} = \Big\| \sum_{j=1}^{K} \sum_{i \in I_2 : y_i = j} \tilde{X}_i^{(j)} \Big\|_{2,*}. \tag{30}$$

*Uniform bound.* Under Assumption B.1, $\|Z\|_2 \le R_Z$ almost surely and hence, for any fixed matrix $M$, $\|M^\top Z\|_2 \le \|M\|_{2,*} \|Z\|_2$. Since $\hat{h}(z) = \hat{W}^\top z$ and $\|\hat{U}\|_{2,*} = 1$, we have

$$\|\hat{W}\|_{2,*} = \|\widehat{S}_W^{-1/2} \hat{U}\|_{2,*} \le \|\widehat{S}_W^{-1/2}\|_{2,*}. \tag{31}$$

On the event

$$\mathcal{E}_W := \{\lambda_{\min}(\widehat{S}_W) \ge c_W/2\}, \tag{32}$$

we have $\|\widehat{S}_W^{-1/2}\|_{2,*} \le \sqrt{2/c_W}$, hence

$$\sup_{\|z\|_2 \le R_Z} \|\hat{h}(z)\|_2 \le R_Z \|\hat{W}\|_{2,*} \le R_Z \sqrt{\frac{2}{c_W}} =: B_h. \tag{33}$$

Therefore, for all $i, j$ on $\mathcal{E}_W$,

$$\|X_i^{(j)}\|_{2,*} = \frac{1}{n_{2,j}} \big\| \hat{h}(Z_i) - \mathbb{E}_P[\hat{h}(Z) \mid Y = j] \big\|_2 \le \frac{1}{n_{2,j}} \left( \|\hat{h}(Z_i)\|_2 + \|\mathbb{E}_P[\hat{h}(Z) \mid Y = j]\|_2 \right) \le \frac{2B_h}{n_{2,j}}. \tag{34}$$

In particular,

$$\max_{i,j} \|\tilde{X}_i^{(j)}\|_{2,*} = \max_{i,j} \|X_i^{(j)}\|_{2,*} \le \frac{2B_h}{n_{2,\min}} =: R. \tag{35}$$

*Variance proxy.* Using $\|(\tilde{X}_i^{(j)})^2\|_{2,*} = \|X_i^{(j)}\|_{2,*}^2$ and (34), we obtain on $\mathcal{E}_W$,

$$\sigma^2 := \Big\| \sum_{j=1}^{K} \sum_{i \in I_2 : y_i = j} \mathbb{E}\big[(\tilde{X}_i^{(j)})^2 \mid \mathcal{F}_1\big] \Big\|_{2,*} \le \sum_{j=1}^{K} \sum_{i \in I_2 : y_i = j} \mathbb{E}\big[\|\tilde{X}_i^{(j)}\|_{2,*}^2 \mid \mathcal{F}_1\big] \le \sum_{j=1}^{K} n_{2,j} \left(\frac{2B_h}{n_{2,j}}\right)^2 = 4B_h^2 \sum_{j=1}^{K} \frac{1}{n_{2,j}}. \tag{36}$$

*Matrix Bernstein.* Conditional on $\mathcal{F}_1$, the self-adjoint summands $\{\tilde{X}_i^{(j)}\}$ are independent, mean-zero by (29), and satisfy the uniform bound (35) and variance proxy (36) on $\mathcal{E}_W$. Applying Theorem 1.4 of (Tropp, 2012) (matrix Bernstein) with dimension $d_0 = 2k - 1$, we get that for any $\delta \in (0, 1)$, on $\mathcal{E}_W$,

$$\|\hat{A} - \check{A}\|_{2,*} \le \sqrt{2\sigma^2 \log \frac{2d_0}{\delta}} + \frac{R}{3} \log \frac{2d_0}{\delta} \tag{37}$$

with conditional probability at least $1 - \delta$ (given $\mathcal{F}_1$). Plugging (35)–(36) into (37) yields the following bound

$$\|\hat{A} - \check{A}\|_{2,*} \leq 2B_h \left( \sqrt{2 \log \frac{2(2k-1)}{\delta} \sum_{j=1}^{K} \frac{1}{n_{2,j}}} + \frac{1}{3n_{2,\min}} \log \frac{2(2k-1)}{\delta} \right) =: \Delta_{A,\mathrm{est}}(\delta). \tag{38}$$

*Term* $\|\check{A} - A\|_{2,*}$. This term captures the error induced by using the estimated projection $\hat{h}$ rather than the population projection $h$. Write $\Delta_W := \hat{W} - W$ and recall $\mu_j := \mathbb{E}_P[Z \mid Y = j]$ and $\mu = [\mu_1, \ldots, \mu_K] \in \mathbb{R}^{d \times K}$. Since $h(z) = W^\top z$ and $\hat{h}(z) = \hat{W}^\top z$, for each $j \in [K]$,

$$\check{A}_{:j} - A_{:j} = \mathbb{E}_P[(\hat{W} - W)^\top Z \mid Y = j] = (\Delta_W)^\top \mu_j.$$

Stacking the columns gives the exact matrix identity

$$\check{A} - A = (\Delta_W)^\top \mu. \tag{39}$$

Therefore, by submultiplicativity of the operator norm,

$$\|\check{A} - A\|_{2,*} \leq \|\Delta_W\|_{2,*} \|\mu\|_{2,*}. \tag{40}$$

Under Assumption B.1, $\|\mu_j\|_2 \leq \mathbb{E}[\|Z\|_2 \mid Y = j] \leq R_Z$, hence

$$\|\mu\|_{2,*} \leq \|\mu\|_F = \left( \sum_{j=1}^{K} \|\mu_j\|_2^2 \right)^{1/2} \leq \sqrt{K}\, R_Z. \tag{41}$$

Combining (40)–(41) yields

$$\|\check{A} - A\|_{2,*} \leq \sqrt{K}\, R_Z \|\hat{W} - W\|_{2,*}. \tag{42}$$

*Controlling* $\|\hat{W} - W\|_{2,*}$. Using $\hat{W} = \widehat{S}_W^{-1/2}\hat{U}$ and $W = S_W^{-1/2}U$, add and subtract $S_W^{-1/2}\hat{U}$ to obtain the identity

$$\hat{W} - W = (\widehat{S}_W^{-1/2} - S_W^{-1/2})\hat{U} + S_W^{-1/2}(\hat{U} - U). \tag{43}$$

Taking operator norms and using $\|\hat{U}\|_{2,*} = 1$ gives

$$\|\hat{W} - W\|_{2,*} \leq \|\widehat{S}_W^{-1/2} - S_W^{-1/2}\|_{2,*} + \|S_W^{-1/2}\|_{2,*} \|\hat{U} - U\|_{2,*}. \tag{44}$$

Since $\lambda_{\min}(S_W) \geq c_W$, we have

$$\|S_W^{-1/2}\|_{2,*} = \lambda_{\min}(S_W)^{-1/2} \leq c_W^{-1/2}. \tag{45}$$

*Perturbation of the inverse square-root.* Assume additionally that $\mathcal{E}_W$ in (32) holds. The scalar function $f(x) = x^{-1/2}$ is differentiable on $[c_W/2, \infty)$ with $\sup_{x \geq c_W/2} |f'(x)| = \frac{1}{2}(c_W/2)^{-3/2} = \frac{\sqrt{2}}{c_W^{3/2}}$. By mean-value bound for operator-Lipschitz functions on a compact interval, it follows that

$$\|\widehat{S}_W^{-1/2} - S_W^{-1/2}\|_{2,*} \leq \frac{\sqrt{2}}{c_W^{3/2}} \|\widehat{S}_W - S_W\|_{2,*} \qquad \text{on } \mathcal{E}_W. \tag{46}$$

*Eigenvector perturbation.* Under Assumption B.2, Lemma 1 of Cai & Zhang (2018) together with Theorem V.3.6 of Stewart & Sun (1990) implies a Davis–Kahan-type $\sin\Theta$ perturbation bound. Specifically, there exists a diagonal sign matrix $S \in \mathbb{R}^{(K-1)\times(K-1)}$, whose diagonal entries belong to $\{\pm 1\}$, such that

$$\|\hat{U} - US\|_{2,*} \leq \frac{2\sqrt{2}}{\gamma} \|\widehat{\tilde{S}}_B - \tilde{S}_B\|_{2,*}. \tag{47}$$

The sign matrix $S$ accounts for the intrinsic sign ambiguity of the eigenvectors. A closely related form of this perturbation bound is also given in Corollary 3 of Yu et al. (2015). Since the top $K - 1$ eigenvalues are simple under Assumption B.2,

the sign matrix $S$ accounts only for the inherent sign ambiguity of eigenvectors. Without loss of generality, we absorb this sign change into $U$ and take $S = I$ in the sequel.

Finally, combining (44), (45), (46) and (47), and absorbing the sign matrix into the definition of $U$ gives, on $\mathcal{E}_W$,

$$\|\hat{W} - W\|_{2,*} \leq \frac{\sqrt{2}}{c_W^{3/2}} \|\widehat{S}_W - S_W\|_{2,*} + \frac{2\sqrt{2}}{\gamma\sqrt{c_W}} \|\widehat{\tilde{S}}_B - \tilde{S}_B\|_{2,*}. \tag{48}$$

Substituting (48) into (42) yields a reduction of $\|\check{A} - A\|_{2,*}$ to deviations of $\widehat{S}_W, \widehat{S}_B$.

**Proposition B.6** (Concentration of $\widehat{\tilde{S}}_B$)**.** *Recall* $\tilde{S}_B := S_W^{-1/2} S_B S_W^{-1/2}$ *and* $\widehat{\tilde{S}}_B := \widehat{S}_W^{-1/2} \widehat{S}_B \widehat{S}_W^{-1/2}$. *Assume additionally that the event* $\mathcal{E}_W := \{\lambda_{\min}(\widehat{S}_W) \geq c_W/2\}$ *holds. Then deterministically on* $\mathcal{E}_W$,

$$\|\widehat{\tilde{S}}_B - \tilde{S}_B\|_{2,*} \leq \frac{\sqrt{2}}{c_W} \|\widehat{S}_B - S_B\|_{2,*} + \frac{(8 + 4\sqrt{2})R_Z^2}{c_W^2} \|\widehat{S}_W - S_W\|_{2,*}. \tag{49}$$

*Consequently, combining Proposition B.4 and Proposition B.5, on* $\mathcal{E}_W$ *we have with probability at least* $1 - 2\delta$ *that*

$$\|\widehat{\tilde{S}}_B - \tilde{S}_B\|_{2,*} \leq \frac{\sqrt{2}}{c_W} \left(8R_Z\Delta_\mu(\delta) + 4\Delta_\mu(\delta)^2\right) + \frac{(8 + 4\sqrt{2})R_Z^2}{c_W^2} \Delta_{S_W}(\delta). \tag{50}$$

*Proof.* Starting from the add–subtract decomposition

$$\widehat{\tilde{S}}_B - \tilde{S}_B = (\widehat{S}_W^{-1/2} - S_W^{-1/2})\, \widehat{S}_B\, \widehat{S}_W^{-1/2} + S_W^{-1/2}(\widehat{S}_B - S_B)\widehat{S}_W^{-1/2} + S_W^{-1/2} S_B(\widehat{S}_W^{-1/2} - S_W^{-1/2}).$$

We emphasize a minor notational distinction between the transformed matrices $\tilde{S}_B, \tilde{S}_W$ and the original scatter matrices $S_B, S_W$. Taking operator norms and using submultiplicativity yields

$$\|\widehat{\tilde{S}}_B - \tilde{S}_B\|_{2,*} \leq \|\widehat{S}_W^{-1/2} - S_W^{-1/2}\|_{2,*} \|\widehat{S}_B\|_{2,*} \|\widehat{S}_W^{-1/2}\|_{2,*} + \|S_W^{-1/2}\|_{2,*} \|\widehat{S}_B - S_B\|_{2,*} \|\widehat{S}_W^{-1/2}\|_{2,*} \tag{51}$$

$$+ \|S_W^{-1/2}\|_{2,*} \|S_B\|_{2,*} \|\widehat{S}_W^{-1/2} - S_W^{-1/2}\|_{2,*}. \tag{52}$$

We bound each factor explicitly. First, by Assumption B.1, for every $j$ we have $\|\widehat{\mu}_j\|_2 \leq R_Z$ and $\|\widehat{\bar{\mu}}\|_2 \leq R_Z$, hence $\|\widehat{\mu}_j - \widehat{\bar{\mu}}\|_2 \leq 2R_Z$ and therefore

$$\|\widehat{S}_B\|_{2,*} \leq \sum_{j=1}^{K} p_j \|\widehat{\mu}_j - \widehat{\bar{\mu}}\|_2^2 \leq 4R_Z^2. \tag{53}$$

The same argument gives $\|S_B\|_{2,*} \leq 4R_Z^2$. Second, $\|S_W^{-1/2}\|_{2,*} \leq c_W^{-1/2}$ by Assumption B.1, and on $\mathcal{E}_W$ we have

$$\|\widehat{S}_W^{-1/2}\|_{2,*} \leq \sqrt{\frac{2}{c_W}}. \tag{54}$$

Third, on $\mathcal{E}_W$, the inverse-square-root perturbation bound (46) implies

$$\|\widehat{S}_W^{-1/2} - S_W^{-1/2}\|_{2,*} \leq \frac{\sqrt{2}}{c_W^{3/2}} \|\widehat{S}_W - S_W\|_{2,*}. \tag{55}$$

Plugging (53)–(55) into (52) yields

$$\|\widehat{\tilde{S}}_B - \tilde{S}_B\|_{2,*} \leq 4R_Z^2 \left(\sqrt{\frac{2}{c_W}} + \frac{1}{\sqrt{c_W}}\right) \|\widehat{S}_W^{-1/2} - S_W^{-1/2}\|_{2,*} + \frac{\sqrt{2}}{c_W} \|\widehat{S}_B - S_B\|_{2,*}.$$

Using (55) and simplifying constants gives (49). Finally, combining (49) with (22) and (23) and taking a union bound over those two concentration events yields (50) with probability at least $1 - 2\delta$. $\qquad\square$

Fix $\delta \in (0, 1)$ and set the split

$$\delta_0 := \delta/5. \tag{56}$$

Define the following fully explicit deviation level for $\|\hat{W} - W\|_{2,*}$:

$$\Delta_W(\delta) := \frac{\sqrt{2}}{c_W^{3/2}} \Delta_{S_W}(\delta) + \frac{2\sqrt{2}}{\gamma\sqrt{c_W}} \left\{ \frac{\sqrt{2}}{c_W} \left(8R_Z \Delta_\mu(\delta) + 4\Delta_\mu(\delta)^2\right) + \frac{(8 + 4\sqrt{2})R_Z^2}{c_W^2} \Delta_{S_W}(\delta) \right\}. \tag{57}$$

Also define the fully explicit deviation level for the moment matrix error:

$$\Delta_A(\delta) := \Delta_{A,\text{est}}(\delta) + \sqrt{K} R_Z \Delta_W(\delta), \tag{58}$$

where $\Delta_{A,\text{est}}(\delta)$ is the Bernstein bound in (38) with $B_h$. In particular, (38) gives

$$\Delta_{A,\text{est}}(\delta) := 2B_h \left( \sqrt{2\log \frac{2(2k-1)}{\delta} \sum_{j=1}^{K} \frac{1}{n_{2,j}}} + \frac{1}{3n_{2,\min}} \log \frac{2(2k-1)}{\delta} \right). \tag{59}$$

**Proposition B.7** (Concentration inequality for $\|\hat{A} - A\|_{2,*}$). *Assume Assumption B.1 and B.2 hold. Suppose the sample-size condition* (7) *holds with $\delta$ replaced by $\delta_0 = \delta/5$ in* (56) *so that $\mathbb{P}(\mathcal{E}_W) \geq 1 - \delta_0$ by Proposition B.3. Then, with probability at least $1 - \delta$ over all samples,*

$$\|\hat{A} - A\|_{2,*} \leq \Delta_A(\delta_0), \tag{60}$$

*where $\Delta_A$ is defined in* (58).

*Proof.* By (26), on $\mathcal{E}_W$ we have $\|\hat{A} - A\|_{2,*} \leq \|\hat{A} - \check{A}\|_{2,*} + \|\check{A} - A\|_{2,*}$. Conditional on $\mathcal{F}_1 = \sigma(S_1)$, matrix Bernstein gives $\|\hat{A} - \check{A}\|_{2,*} \leq \Delta_{A,\text{est}}(\delta_0)$ with conditional probability at least $1 - \delta_0$ using the deterministic envelope $\bar{B}_h$ on $\mathcal{E}_W$. Moreover, (42) and (48) yield on $\mathcal{E}_W$, $\|\check{A} - A\|_{2,*} \leq \sqrt{K} R_Z \|\hat{W} - W\|_{2,*}$ and $\|\hat{W} - W\|_{2,*} \leq \frac{\sqrt{2}}{c_W^{3/2}} \|\hat{S}_W - S_W\|_{2,*} + \frac{2\sqrt{2}}{\gamma\sqrt{c_W}} \|\hat{\tilde{S}}_B - \tilde{S}_B\|_{2,*}$. Finally, on $\mathcal{E}_W$, Proposition B.4 and Proposition B.6 imply the reduction $\|\hat{W} - W\|_{2,*} \leq \Delta_W(\delta_0)$ with probability at least $1 - 2\delta_0$. A union bound over the four events $\mathcal{E}_W$, $\{\|\hat{A} - \check{A}\|_{2,*} \leq \Delta_{A,\text{est}}(\delta_0)\}$, $\{\|\hat{S}_W - S_W\|_{2,*} \leq \Delta_{S_W}(\delta_0)\}$, and $\{\|\hat{\tilde{S}}_B - \tilde{S}_B\|_{2,*} \leq \text{RHS of } (50) \text{ with } \delta = \delta_0\}$ shows that all bounds hold simultaneously with probability at least $1 - 5\delta_0 = 1 - \delta$. On this intersection, the displayed bound (60) follows directly from the definitions (57)–(58). □

### B.6. Concentration inequalities for $b$

Recall that $b \in \mathbb{R}^{K-1}$ is the target-domain moment vector defined by

$$b := \mathbb{E}_Q[h(Z)], \qquad h(z) = W^\top z, \qquad W = S_W^{-1/2} U.$$

In the algorithm, we use the estimated projection $\hat{h}(z) = \hat{W}^\top z$ and form

$$\hat{b} := \mathbb{Q}_{n_T}[\hat{h}(Z)] = \frac{1}{n_T} \sum_{i=1}^{n_T} \hat{h}(Z_i^{\text{tgt}}), \qquad \check{b} := \mathbb{E}_Q[\hat{h}(Z)].$$

Then the total error admits the exact decomposition

$$\|\hat{b} - b\|_2 \leq \|\hat{b} - \check{b}\|_2 + \|\check{b} - b\|_2. \tag{61}$$

*Term $\|\hat{b} - \check{b}\|_2$.* Let $S_1$ be the source split used to construct $\hat{W}$ and define $\mathcal{F}_1 := \sigma(S_1)$. Then $\hat{h}$ is $\mathcal{F}_1$-measurable and, conditional on $\mathcal{F}_1$, the target samples $\{Z_i^{\text{tgt}}\}_{i=1}^{n_T}$ are i.i.d. from $Q$. Define the event

$$\mathcal{E}_W := \{\lambda_{\min}(\hat{S}_W) \geq c_W/2\}, \tag{62}$$

which is controlled in Proposition B.3. On $\mathcal{E}_W$, we have

$$\|\hat{W}\|_{2,*} = \|\hat{S}_W^{-1/2}\hat{U}\|_{2,*} \leq \|\hat{S}_W^{-1/2}\|_{2,*} \leq \sqrt{\frac{2}{c_W}},$$

and thus, using $\|Z\|_2 \leq R_Z$ almost surely (by Assumption B.1),

$$\sup_{\|z\|_2 \leq R_Z} \|\hat{h}(z)\|_2 \leq R_Z \|\hat{W}\|_{2,*} \leq R_Z \sqrt{\frac{2}{c_W}} =: B_h. \tag{63}$$

Now define the centered i.i.d. vectors

$$\xi_i := \hat{h}(Z_i^{\text{tgt}}) - \mathbb{E}_Q[\hat{h}(Z)] = \hat{h}(Z_i^{\text{tgt}}) - \check{b} \in \mathbb{R}^{K-1}, \qquad i = 1, \ldots, n_T.$$

Then $\mathbb{E}[\xi_i \mid \mathcal{F}_1] = 0$ and, on $\mathcal{E}_W$,

$$\|\xi_i\|_2 \leq \|\hat{h}(Z_i^{\text{tgt}})\|_2 + \|\check{b}\|_2 \leq B_h + \mathbb{E}_Q[\|\hat{h}(Z)\|_2] \leq B_h + B_h = 2B_h. \tag{64}$$

Moreover,

$$\hat{b} - \check{b} = \frac{1}{n_T} \sum_{i=1}^{n_T} \xi_i. \tag{65}$$

**Proposition B.8** (Concentration inequality for $\|\hat{b} - \check{b}\|_2$ )**.** *Suppose Assumption B.1 holds. On the event $\mathcal{E}_W$ in* (62)*, for any $\delta \in (0,1)$, with conditional probability at least $1 - \delta$,*

$$\|\hat{b} - \check{b}\|_2 \leq 2B_h \sqrt{\frac{2 \log\left(\frac{2k}{\delta}\right)}{n_T}} + \frac{2B_h}{3} \frac{\log\left(\frac{2k}{\delta}\right)}{n_T} =: \Delta_{b,\text{est}}(\delta), \tag{66}$$

*where $B_h$ is defined in* (63)*.*

*Proof.* Condition on $\mathcal{F}_1$. Fix $\delta \in (0,1)$. Consider the self-adjoint dilation of each $\xi_i$:

$$\tilde{\xi}_i := \begin{bmatrix} 0 & \xi_i^\top \\ \xi_i & 0 \end{bmatrix} \in \mathbb{R}^{K \times K}.$$

Then $\tilde{\xi}_i$ is self-adjoint, $\mathbb{E}[\tilde{\xi}_i \mid \mathcal{F}_1] = 0$, and $\|\tilde{\xi}_i\|_{2,*} = \|\xi_i\|_2$. By (64), on $\mathcal{E}_W$ we have the uniform bound

$$\|\tilde{\xi}_i\|_{2,*} \leq 2B_h =: L. \tag{67}$$

Moreover, $\tilde{\xi}_i^2 = \text{diag}(\xi_i^\top \xi_i, \, \xi_i \xi_i^\top)$ implies $\|\tilde{\xi}_i^2\|_{2,*} = \|\xi_i\|_2^2$. Using $\|\mathbb{E}[\tilde{\xi}_i^2 \mid \mathcal{F}_1]\|_{2,*} \leq \mathbb{E}[\|\tilde{\xi}_i^2\|_{2,*} \mid \mathcal{F}_1]$ and (64), on $\mathcal{E}_W$ we obtain

$$\sigma^2 := \left\| \sum_{i=1}^{n_T} \mathbb{E}[\tilde{\xi}_i^2 \mid \mathcal{F}_1] \right\|_{2,*} \leq \sum_{i=1}^{n_T} \mathbb{E}[\|\xi_i\|_2^2 \mid \mathcal{F}_1] \leq n_T \, (2B_h)^2 = 4n_T B_h^2. \tag{68}$$

Apply Theorem 1.4 of (Tropp, 2012) (matrix Bernstein) to the self-adjoint sum $\sum_{i=1}^{n_T} \tilde{\xi}_i$ in dimension $K$. With conditional probability at least $1 - \delta$, on $\mathcal{E}_W$,

$$\left\| \sum_{i=1}^{n_T} \tilde{\xi}_i \right\|_{2,*} \leq \sqrt{2\sigma^2 \log\left(\frac{2k}{\delta}\right)} + \frac{L}{3} \log\left(\frac{2k}{\delta}\right). \tag{69}$$

Since $\|\sum_i \tilde{\xi}_i\|_{2,*} = \|\sum_i \xi_i\|_2$ and $\hat{b} - \check{b} = \frac{1}{n_T} \sum_i \xi_i$ by (65), dividing (69) by $n_T$ and substituting (67)–(68) yields (66). Finally, the unconditional probability statement follows by integrating over $\mathcal{F}_1$. $\qquad\square$

*Term $\|\check{b} - b\|_2$.* By definition,
$$\check{b} - b = \mathbb{E}_Q[\hat{h}(Z) - h(Z)] = \mathbb{E}_Q[(\hat{W} - W)^\top Z].$$

Therefore, by Jensen's inequality and submultiplicativity of the operator norm,

$$\|\check{b} - b\|_2 \leq \mathbb{E}_Q[\|(\hat{W} - W)^\top Z\|_2] \leq \|\hat{W} - W\|_{2,*} \, \mathbb{E}_Q[\|Z\|_2] \leq R_Z \|\hat{W} - W\|_{2,*}, \tag{70}$$

where the last inequality uses $\|Z\|_2 \leq R_Z$ almost surely.

Combining (61), (66), and (70) gives, on $\mathcal{E}_W$,

$$\|\hat{b} - b\|_2 \leq \Delta_{b,\mathrm{est}}(\delta) + R_Z \|\hat{W} - W\|_{2,*}$$

with conditional probability at least $1 - \delta$.

Fix $\delta \in (0,1)$ and reuse the bookkeeping split $\delta_0 := \delta/5$ from (56). Recall the explicit projection deviation $\Delta_W(\cdot)$ from (57). Define the sampling deviation for $\hat{b}$:

$$\Delta_{b,\mathrm{est}}(\delta) := 2B_h \sqrt{\frac{2\log\left(\frac{2k}{\delta}\right)}{n_T}} + \frac{2B_h}{3} \frac{\log\left(\frac{2k}{\delta}\right)}{n_T}. \tag{71}$$

Also define the fully explicit total deviation for the target moment vector:

$$\Delta_b(\delta) := \Delta_{b,\mathrm{est}}(\delta) + R_Z \Delta_W(\delta). \tag{72}$$

**Proposition B.9** (Concentration inequality for $\|\hat{b} - b\|_2$). *Assume Assumption B.1 and the eigengap condition (5) with $\gamma > 0$. Suppose the sample-size condition (7) holds with $\delta$ replaced by $\delta_0 = \delta/5$ (so that $\mathbb{P}(\mathcal{E}_W) \geq 1 - \delta_0$ by Proposition B.3). Then, with probability at least $1 - \delta$ over all samples,*

$$\|\hat{b} - b\|_2 \leq \Delta_b(\delta_0), \tag{73}$$

*where $\Delta_b$ is defined in (72).*

*Proof.* By (61), on $\mathcal{E}_W$ we have $\|\hat{b} - b\|_2 \leq \|\hat{b} - \check{b}\|_2 + \|\check{b} - b\|_2$. Conditional on $\mathcal{F}_1 = \sigma(S_1)$, Proposition B.8 yields $\|\hat{b} - \check{b}\|_2 \leq \Delta_{b,\mathrm{est}}(\delta_0)$ with conditional probability at least $1 - \delta_0$. Moreover, (70) implies $\|\check{b} - b\|_2 \leq R_Z \|\hat{W} - W\|_{2,*}$. Finally, on $\mathcal{E}_W$, Proposition B.4 and Proposition B.6, together with (48), imply $\|\hat{W} - W\|_{2,*} \leq \Delta_W(\delta_0)$ with probability at least $1 - 2\delta_0$. A union bound over the four events $\mathcal{E}_W$, $\{\|\hat{b} - \check{b}\|_2 \leq \Delta_{b,\mathrm{est}}(\delta_0)\}$, $\{\|\widehat{S}_W - S_W\|_{2,*} \leq \Delta_{S_W}(\delta_0)\}$, and $\{\|\widehat{S}_B - \tilde{S}_B\|_{2,*} \leq \text{RHS of (50) with } \delta = \delta_0\}$ shows that all bounds hold simultaneously with probability at least $1 - (\delta_0 + \delta_0 + \delta_0 + 2\delta_0) = 1 - 5\delta_0 = 1 - \delta$. On this intersection, we have $\|\hat{b} - b\|_2 \leq \Delta_{b,\mathrm{est}}(\delta_0) + R_Z \Delta_W(\delta_0) = \Delta_b(\delta_0)$, which proves (73). $\qquad\square$

## B.7. Concentration inequalities for importance weights

This subsection combines the concentration results for $(\hat{A}, \hat{b})$ obtained in Sections B.5–B.6 to derive a high-probability bound for $\|\hat{q} - q^*\|_2$, and $\|\hat{w} - w^*\|_\infty$.

*Adding and subtracting.* Recall $q^* = (Q_y(1), \ldots, Q_y(K))^\top \in \Delta^{K-1}$ and $Aq^* = b$. Define

$$\Upsilon(q) := \|Aq - b\|_2, \qquad q \in \Delta^{K-1}.$$

Let $\mathcal{E}_W := \{\lambda_{\min}(\widehat{S}_W) \geq c_W/2\}$ as in (32). By Section B.5, on $\mathcal{E}_W$ we have $\|\hat{A} - A\|_{2,*} \leq \Delta_A(\delta/5)$ with probability at least $1 - \delta$. By Section B.6, on $\mathcal{E}_W$ we have $\|\hat{b} - b\|_2 \leq \Delta_b(\delta/5)$ with probability at least $1 - \delta$.

Consider the regularized estimator

$$\hat{q} \in \arg\min_{q \in \Delta^{K-1}} \left\{\|\hat{A}q - \hat{b}\|_2 + \lambda\|q\|_2\right\}. \tag{74}$$

For any $q \in \Delta^{K-1}$, by adding and subtracting terms,

$$Aq - b = (\hat{A}q - \hat{b}) + (A - \hat{A})q + (\hat{b} - b),$$

and hence by the triangle inequality and submultiplicativity,

$$\Upsilon(q) \leq \|\hat{A}q - \hat{b}\|_2 + \|\hat{A} - A\|_{2,*} \|q\|_2 + \|\hat{b} - b\|_2. \tag{75}$$

By optimality of $\hat{q}$ in (74), for all $q \in \Delta^{K-1}$,

$$\|\hat{A}\hat{q} - \hat{b}\|_2 + \lambda\|\hat{q}\|_2 \leq \|\hat{A}q - \hat{b}\|_2 + \lambda\|q\|_2. \tag{76}$$

Taking $q = q^*$ and using $Aq^* = b$ yields

$$\|\hat{A}\hat{q} - \hat{b}\|_2 \leq \|\hat{A}q^* - \hat{b}\|_2 + \lambda(\|q^*\|_2 - \|\hat{q}\|_2) \tag{77}$$
$$\leq \|\hat{A}q^* - \hat{b}\|_2 + \lambda\|q^*\|_2. \tag{78}$$

Applying (75) with $q = \hat{q}$ and then (78) gives

$$\Upsilon(\hat{q}) \leq \|\hat{A}q^* - \hat{b}\|_2 + \lambda\|q^*\|_2 + \|\hat{A} - A\|_{2,*}\|\hat{q}\|_2 + \|\hat{b} - b\|_2. \tag{79}$$

Since $\hat{q} \in \Delta^{K-1}$, we have $\|\hat{q}\|_2 \leq \|\hat{q}\|_1 = 1$. Moreover,

$$\hat{A}q^* - \hat{b} = (\hat{A} - A)q^* + (b - \hat{b}),$$

so

$$\|\hat{A}q^* - \hat{b}\|_2 \leq \|\hat{A} - A\|_{2,*}\|q^*\|_2 + \|\hat{b} - b\|_2. \tag{80}$$

Combining (79)–(80) and then restricting to the event

$$\{\lambda \geq \|\hat{A} - A\|_{2,*}\} \tag{81}$$

yields

$$\Upsilon(\hat{q}) = \|A(\hat{q} - q^*)\|_2 \leq (\|\hat{A} - A\|_{2,*} + \lambda)\|q^*\|_2 + 2\|\hat{b} - b\|_2. \tag{82}$$

*Replacing $\sigma_{\min}(A)$ by the transformed linear system.* Recall the reparameterization from Section 2: for any $q \in \Delta^{K-1}$ there exists a unique $\tilde{q} \in \mathbb{R}^{K-1}$ such that $q = p + D^{1/2}C_p\tilde{q}$, where $D = \text{diag}(p)$ and $C_p \in \mathbb{R}^{K \times (K-1)}$ has orthonormal columns spanning $\text{span}\{D^{1/2}\mathbf{1}_K\}^\perp$. In particular, $\hat{q} - q^* = D^{1/2}C_p(\hat{\tilde{q}} - \tilde{q}^*)$. Define the square transformed matrix

$$\tilde{A} := AD^{1/2}C_p \in \mathbb{R}^{(K-1)\times(K-1)} \tag{83}$$

so that $A(\hat{q} - q^*) = \tilde{A}(\hat{\tilde{q}} - \tilde{q}^*)$. Let $p_{\max} := \max_{j \in [K]} p_j$. Since $C_p^\top C_p = I_{K-1}$, we have $\|D^{1/2}C_p\|_{2,*}^2 = \lambda_{\max}(C_p^\top D C_p) \leq p_{\max}$, hence $\|\hat{q} - q^*\|_2 \leq \sqrt{p_{\max}}\|\hat{\tilde{q}} - \tilde{q}^*\|_2$. Therefore,

$$\|A(\hat{q} - q^*)\|_2 = \|\tilde{A}(\hat{\tilde{q}} - \tilde{q}^*)\|_2 \geq \sigma_{\min}(\tilde{A})\|\hat{\tilde{q}} - \tilde{q}^*\|_2 \geq \frac{\sigma_{\min}(\tilde{A})}{\sqrt{p_{\max}}}\|\hat{q} - q^*\|_2. \tag{84}$$

Therefore, on $\mathcal{E}_W \cap \{\lambda \geq \|\hat{A} - A\|_{2,*}\}$, we obtain

$$\|\hat{q} - q^*\|_2 \leq \frac{\sqrt{p_{\max}}}{\sigma_{\min}(\tilde{A})} \left((\|\hat{A} - A\|_{2,*} + \lambda)\|q^*\|_2 + 2\|\hat{b} - b\|_2\right). \tag{85}$$

In particular, if we choose

$$\lambda \geq \Delta_A(\delta/10), \tag{86}$$

then with probability at least $1 - \delta$ by a union bound, together with Proposition B.3),

$$\|\hat{q} - q^*\|_2 \leq \epsilon_{q^*}(n_S, n_T, \delta, \lambda) := \frac{\sqrt{p_{\max}}}{\sigma_{\min}(\tilde{A})} \left((\Delta_A(\delta/10) + \lambda)\|q^*\|_2 + 2\Delta_b(\delta/10)\right). \tag{87}$$

Define the true and estimated importance weights

$$w^*(j) := \frac{q_j^*}{p_j}, \qquad \hat{w}(j) := \frac{\hat{q}_j}{p_j}, \qquad j \in [K],$$

where $p_j = P_y(y = j)$ and $p_{\min} := \min_{j \in [K]} p_j > 0$. Then

$$\|\hat{w} - w^*\|_\infty = \max_{j \in [K]} \frac{|\hat{q}_j - q_j^*|}{p_j} \leq \frac{1}{p_{\min}}\|\hat{q} - q^*\|_\infty \leq \frac{1}{p_{\min}}\|\hat{q} - q^*\|_2. \tag{88}$$

In particular, on the high-probability event of (87),

$$\|\hat{w} - w^*\|_\infty \leq \frac{1}{p_{\min}}\epsilon_{q^*}(n_S, n_T, \delta, \lambda). \tag{89}$$

### B.8. Generalization bound.

Without loss of generality, we assume the loss is bounded as $0 \le \ell \le 1$. We analyze weighted ERM on an independent source sample $S \sim P$ of size $n := n_S$. Define the empirical weighted risk on $S$:

$$\widehat{\mathcal{R}}_{P,\mathrm{tr}}^{w}(f) := \frac{1}{n} \sum_{i=1}^{n} w(Y_i)\, \ell(f(X_i), Y_i),$$

and let

$$\hat{f}_{\hat{w}} := \arg\min_{f \in \mathcal{F}} \widehat{\mathcal{R}}_{P,\mathrm{tr}}^{\hat{w}}(f), \qquad f^* := \arg\min_{f \in \mathcal{F}} \mathcal{R}_Q(f).$$

Using label shift, $\mathcal{R}_Q(f) = \mathbb{E}_P[w^*(Y)\ell(f(X), Y)]$. Add and subtract terms:

$$\mathcal{R}_Q(\hat{f}_{\hat{w}}) - \mathcal{R}_Q(f^*) = \underbrace{\mathcal{R}_Q(\hat{f}_{\hat{w}}) - \widehat{\mathcal{R}}_{P,\mathrm{tr}}^{w^*}(\hat{f}_{\hat{w}})}_{(b_1)} + \underbrace{\widehat{\mathcal{R}}_{P,\mathrm{tr}}^{w^*}(\hat{f}_{\hat{w}}) - \widehat{\mathcal{R}}_{P,\mathrm{tr}}^{\hat{w}}(\hat{f}_{\hat{w}})}_{(a_1)} \tag{90}$$

$$+ \underbrace{\widehat{\mathcal{R}}_{P,\mathrm{tr}}^{\hat{w}}(\hat{f}_{\hat{w}}) - \widehat{\mathcal{R}}_{P,\mathrm{tr}}^{\hat{w}}(f^*)}_{\le 0} + \underbrace{\widehat{\mathcal{R}}_{P,\mathrm{tr}}^{\hat{w}}(f^*) - \widehat{\mathcal{R}}_{P,\mathrm{tr}}^{w^*}(f^*)}_{(a_2)} + \underbrace{\widehat{\mathcal{R}}_{P,\mathrm{tr}}^{w^*}(f^*) - \mathcal{R}_Q(f^*)}_{(b_2)}. \tag{91}$$

*Controlling $(a_1)$ and $(a_2)$ using $\epsilon_{q^*}$.* Since $0 \le \ell \le 1$ and for any fixed $f$,

$$\left| \widehat{\mathcal{R}}_{P,\mathrm{tr}}^{w^*}(f) - \widehat{\mathcal{R}}_{P,\mathrm{tr}}^{\hat{w}}(f) \right| \le \frac{1}{n} \sum_{i=1}^{n} |w^*(Y_i) - \hat{w}(Y_i)|\, |\ell(f(X_i), Y_i)| \le \|\hat{w} - w^*\|_\infty.$$

Therefore, $|a_1| \le \|\hat{w} - w^*\|_\infty$ and $|a_2| \le \|\hat{w} - w^*\|_\infty$. On the event (89),

$$|a_1| + |a_2| \le 2\|\hat{w} - w^*\|_\infty \le \frac{2}{p_{\min}}\, \epsilon_{q^*}(n_S, n_T, \delta, \lambda). \tag{92}$$

*Controlling $(b_1)$ and $(b_2)$ by a uniform deviation bound.* Let $\mathcal{G} := \{(x,y) \mapsto w^*(y)\ell(f(x), y) : f \in \mathcal{F}\}$ and denote its Rademacher complexity on $n$ samples by $\mathrm{Rad}_n(\mathcal{G})$. Since $0 \le \ell \le 1$ and $\|w^*\|_\infty \le 1/p_{\min}$, we have $|g(x,y)| \le \|w^*\|_\infty$ for all $g \in \mathcal{G}$. By Assumption 2.1 and the McDiarmid inequality as appendix B.4 in (Azizzadenesheli et al., 2019), for any $\delta \in (0,1)$, with probability at least $1 - \delta$ over $S$,

$$\sup_{f \in \mathcal{F}} \left| \mathcal{R}_Q(f) - \widehat{\mathcal{R}}_{P,\mathrm{tr}}^{w^*}(f) \right| \le 2\,\mathrm{Rad}_n(\mathcal{G}) + \|w^*\|_\infty \sqrt{\frac{\log(2/\delta)}{n}}. \tag{93}$$

Consequently, on the same event, $|b_1| \le$ RHS of (93) and $|b_2| \le$ RHS of (93), hence

$$|b_1| + |b_2| \le 4\,\mathrm{Rad}_n(\mathcal{G}) + 2\|w^*\|_\infty \sqrt{\frac{\log(2/\delta)}{n}}. \tag{94}$$

Combining (91), (92), and (94), and using $\|w^*\|_\infty \le 1/p_{\min}$, we obtain that with probability at least $1 - 2\delta$,

$$\mathcal{R}_Q(\hat{f}_{\hat{w}}) - \mathcal{R}_Q(f^*) \le \frac{2}{p_{\min}}\, \epsilon_{q^*}(n_S, n_T, \delta, \lambda) + 4\,\mathrm{Rad}_n(\mathcal{G}) + \frac{2}{p_{\min}} \sqrt{\frac{\log(2/\delta)}{n}}. \tag{95}$$

