# OpenReview forum: "Regularized Discriminative Alignment for Deep Representations under Label Shift"
_ICML.cc/2026/Conference — ICML 2026 regular_

### Official Review · Reviewer_cf6e · 2026-03-04

**Soundness:** 3
**Presentation:** 3
**Significance:** 3
**Originality:** 4
**Overall Recommendation:** 4
**Confidence:** 2

**Summary:**

This paper addresses the problem of robust label shift adaptation in high-dimensional and high-noise environments. It innovatively reconsiders the method of label shift correction in the latent feature space rather than the raw input space. The paper proposes RDALS, which integrates pre-trained representations, LDA-based discriminative projection, and a regularized constrained solver for stable importance weight estimation. The authors provide theoretical guarantees including finite-sample bounds and a generalization bound for the reweighted classifier. Empirical results on standard benchmarks show consistent improvements over prior methods.

**Compliance With Llm Reviewing Policy:**

Affirmed.

**Final Justification:**

After considering the rebuttal and subsequent discussion, I find the core claims and empirical results to be reliable and meaningful. I therefore maintain my original recommendation.

**Key Questions For Authors:**

1.	How sensitive is the method to violations of the latent conditional invariance assumption in real-world shifts?
2.	Does the LDA subspace remain stable when the source sample size is small relative to feature dimension?
3.	How does the method perform when the backbone representation is fine-tuned instead of frozen?
4.	Can the approach be extended beyond linear projections without losing theoretical guarantees?
5.	Is the performance gain mainly due to better conditioning of the linear system or due to dimensionality reduction?

**Limitations:**

yes

**Strengths And Weaknesses:**

Strengths
1.	The paper addresses a practically important problem in domain adaptation and clearly motivates limitations of existing label shift estimators.
2.	The idea of performing moment matching in a discriminative latent subspace is intuitive and well aligned with modern representation learning practice.
3.	The theoretical analysis is relatively complete, including finite-sample error bounds and excess risk guarantees.
4.	The empirical evaluation is extensive and includes ablation studies that clearly isolate the contributions of projection and solver design.

Weaknesses
1.	The use of LDA restricts the approach to linear discriminative subspaces and may limit expressivity in more complex representation geometries.
2.	The theoretical analysis relies on eigengap and boundedness assumptions that may not hold in deep feature spaces.
3.	Experiments are limited to simulated label shift on standard vision datasets and do not include real-world naturally shifted datasets.
4.	The comparison focuses mainly on classical label shift methods and does not explore broader test-time adaptation or semi-supervised baselines.

---

> ### Author Rebuttal · Authors · 2026-03-30
>
> We thank the reviewer for these very insightful questions.
>
> > Q1: How sensitive is the method to violations of the latent conditional invariance assumption in real-world shifts?
>
> We conducted the new experiments on the real-world PACS dataset. It contains images across four distinct stylistic domains: Photo, Art painting, Cartoon, and Sketch. We randomly select one domain to serve as the source and another as the target, and the result table at <https://anonymous.4open.science/r/RDALS_rebuttal-66A5/PACS_dataset.pdf> shows our method maintains strong performance. Regarding potential numerical challenges in high-dimensional feature spaces, the main issue is that the empirical within-class scatter matrix may become singular or ill-conditioned. To address this safely, we simply apply a shrinkage regularization by adding a small $\epsilon I$ before performing the LDA. This perfectly ensures the numerical stability without adding huge computational costs.
>
> We conducted an additional experiment where the latent conditional invariance assumption is violated. The detailed results can be found at <https://anonymous.4open.science/r/RDALS_rebuttal-66A5/noise_r.pdf>. Specifically, we added zero-mean Gaussian noise into the extracted target features, defined as $Z'\_{\text{tgt}} = Z\_{\text{tgt}} + \epsilon\_i$, where $\epsilon\_i \sim \mathcal{N}(0, \Sigma\_{noise})$. We set $\Sigma\_{noise}$ as a diagonal matrix with elements $(r \cdot \sigma\_j)^2$, where $r \in [0, 1]$ controls the noise scale. The experimental results show that our MSE barely increases as the noise ratio $r$ increases.
>
> > Q2: Does the LDA subspace remain stable when the source sample size is small relative to feature dimension?
>
> The LDA subspace remains stable even when the sample size is smaller than the feature dimension ($N < d$). Algorithmically, as noted in the Remark of Section 3.3, we apply a shrinkage regularization ($\hat{S}\_W + \epsilon I$). This mathematically prevents the within-class scatter matrix from becoming singular, ensuring absolute numerical stability during projection. Empirically, our newly added sample size ablation at <https://anonymous.4open.science/r/RDALS_rebuttal-66A5/sample_size.pdf> confirms this: even in the highly data-scarce regime ($N=1000$), the estimation error (MSE) of RDALS remains low and stable, consistently outperforming all the baselines.
>
> > Q3: How does the method perform when the backbone representation is fine-tuned?
>
> We adopt a frozen backbone to keep the framework lightweight and decoupled from source-data training. Fine-tuning the backbone is a natural and promising extension, but it changes the problem setting considerably. Optimizing the representation $\phi$ jointly requires access to labeled source data and a suitable training objective, which introduces additional computational cost and couples the estimation procedure to the source training. One can design a task-specific loss to train $\phi$ end-to-end so that the learned representation is better aligned with the moment-matching conditions underlying our estimator. We view this as a valuable direction for future application and appreciate the reviewer for highlighting it.
>
> > Q4: Can it be extended beyond linear projections without losing theoretical guarantees?
>
> We address the two separable components of this concern.
>
> 1. One could consider richer nonlinear summary functions $h: \mathcal{Z}\to\mathbb{R}^{K-1}$.  However, the appropriate optimality criterion for nonlinear $h$ is no longer captured by $\sigma\_{\min}(\tilde A)$ alone, and the corresponding optimization problem does not admit a clean closed-form solution analogous to LDA.  Characterizing the optimal nonlinear summary function is an interesting direction that we consider beyond the scope of the current work, and we flag it explicitly as a direction for future investigation.
>
> 2.The generalization bound in Theorem 4.6 does *not* rely on linearity of the projection.  Concretely, if one replaces the LDA projection with a nonlinear feature map, the same proof strategy applies, and the excess-risk bound retains exactly the same form as $\sigma\_{\min}(\tilde{A})$ evaluated at the chosen $h$.
>
> > Q5: Performance gain due to better conditioning or dimensionality reduction?
>
> As shown in Table 2, naive dimensionality reduction (e.g., PCA) yields poor performance; the gain stems specifically from the discriminative dimensionality reduction of LDA. According to Theorem 4.1, this specific projection mathematically optimizes the condition number of the moment-matching linear system. Therefore, the fundamental driver of the performance gain is the **improved conditioning**. To further verify this mechanism, we have supplemented the Table 2 ablation experiments on both the MNIST and CIFAR-100 datasets, with detailed results confirming this causal relationship available at <https://anonymous.4open.science/r/RDALS_rebuttal-66A5/ablation.pdf>.

---

> > ### Author Rebuttal · Reviewer_cf6e · 2026-04-02
> >
> > Thanks for your response. I decide to keep the score.

---

> > > ### Author Response · Authors · 2026-04-03
> > >
> > > Thank you once again for your continued engagement and careful assessment! We are delighted to hear that our rebuttal has successfully addressed all of your initial concerns. Given that these issues have been fully resolved, we would sincerely appreciate it if you could consider raising your score to reflect this. Thank you for your time and effort in evaluating our work.

---

### Official Review · Reviewer_vDWY · 2026-03-12

**Soundness:** 3
**Presentation:** 1
**Significance:** 3
**Originality:** 3
**Overall Recommendation:** 4
**Confidence:** 3

**Summary:**

The paper introduces Regularized Discriminative Alignment for Label Shift (RDALS), a framework designed to correct label shift in domain adaptation. The method employs Linear Discriminant Analysis (LDA) to project these representations into a lower-dimensional, highly discriminative subspace.  By doing so, it constructs a moment-matching linear system that theoretically maximizes numerical stability. The importance weights are then recovered using a constrained optimization (SOCP) objective that enforces simplex constraints and includes a regularization term. The authors provide finite-sample error bounds and a generalization bound, alongside empirical evaluations on MNIST, CIFAR-10, and CIFAR-100, demonstrating improvements over baselines like BBSL, RLLS, MLLS, and CPMCN, particularly in extreme shift and data-scarce settings.

**Compliance With Llm Reviewing Policy:**

Affirmed.

**Final Justification:**

I will keep my score unchanged.

**Key Questions For Authors:**

1. How sensitive is the Latent Conditional Invariance assumption to the choice of the pre-trained backbone, and does the method degrade if the backbone was trained on a vastly different domain than the target task?
2. Given the severe readability issues in Section 4, how do you plan to restructure the theoretical presentation to make the core intuition accessible to a broader ICML audience? (A satisfactory revision plan here is critical).
3. How does RDALS behave in an open-set label shift scenario, where the target domain contains classes not present in the source domain?

**Limitations:**

The authors focus strictly on the label shift assumption ($P(y)$ changes, $$P(z|y)$$ remains invariant). While they test on CIFAR-10-C to simulate input perturbations, they do not adequately discuss the method's limitations when the Latent Conditional Invariance assumption breaks down entirely (e.g., severe covariate shift co-occurring with label shift). Adding a dedicated limitation paragraph addressing the boundaries of the pre-trained feature assumption would strengthen the manuscript.

**Strengths And Weaknesses:**

Strengths:
- Theoretical Rigor: The paper establishes a strong theoretical link between the choice of summary functions and the numerical stability of the linear system. Proving the optimality of the LDA subspace for this specific problem is a neat and valuable contribution.
- Empirical Robustness: The method demonstrates clear performance gains in challenging scenarios, such as the Dirichlet shift with $\alpha=0.1$ and the Tweak-One shift.

Weaknesses:
- Poor Presentation and Readability: The paper struggles significantly with clarity and information hierarchy. Section 4 is overwhelmingly dense, transitioning abruptly into heavy mathematical notation without sufficient intuitive scaffolding.
- Overwhelming Appendices: The supplementary material contains massive, dense tables of raw numbers that are difficult to parse.
- Assumption Sensitivity: While "Latent Conditional Invariance" is more practical than pixel-level invariance, the paper lacks a deep discussion on how varying the pre-trained architecture impacts this assumption.

---

> ### Author Rebuttal · Authors · 2026-03-30
>
> We thank the reviewer for these very insightful questions.
>
> > Q1: How sensitive is the Latent Conditional Invariance assumption to the choice of the pre-trained backbone, and does the method degrade if the backbone was trained on a different domain than the target task?
>
> Firstly, regarding backbone sensitivity, Appendix Figure 5 demonstrates that RDALS consistently achieves the lowest error across various architectures (ResNet-18, ResNet-50, and ViT-B-16). Secondly, regarding the pre-training domain gap, our new experiments on the PACS dataset at <https://anonymous.4open.science/r/RDALS_rebuttal-66A5/PACS_dataset.pdf> show good performance even when the downstream tasks vastly differ from the standard ImageNet pre-training domain. Mechanistically, our LDA module fits on the downstream source data, acting as a task-specific filter that discards irrelevant pre-training biases and preserves LCI. However, we acknowledge that for entirely disjoint domains where the frozen features lose all linear separability, the method would inevitably degrade. We have explicitly added this boundary condition to the Limitations section.
>
> > Q2: How do you plan to restructure the theoretical presentation to make the core intuition accessible to a broader ICML audience?
>
> We fully agree that the readability of Section 4 is a key concern, and we regard this revision as one of the highest priorities in our revision. Below we outline our concrete plan to restructure the theoretical presentation.
>
> 1. **To simplify notation.**
> Upon reflection, several pieces of notation introduced in the current Section 4 are required only in the proofs and need not appear in the main text.
> In particular, the empirical-process shorthand $\mathbb{P}\_{n\_{S\_2,P}}$ and $\mathbb{P}\_{n\_T,Q}$ for $\hat{A}$ and $\hat{b}$ are convenient for the proofs but add cognitive load for a reader who only wants to understand the method. We will replace these with explicit sample average notation directly and defer the empirical-process formalism to the appendix.
> Similarly, the construction of the contrast matrix $C\_p$ and the reparametrization $\tilde q$ will be moved to the appendix; in the main text we will state the identification argument informally.
>
> 2. **Moving technical assumptions and results to the appendix.**
> To reduce density, we will relocate Assumptions 4.2 and 4.3 to the appendix. Similarly, the construction of the contrast matrix $C\_p$ and the reparametrization $\tilde q$ will be moved to the appendix; in the main text we will state the identification argument informally.
>
> 3. **Added intuition in Sections 4.1 and 4.2.**
> We will augment the two subsections with brief intuitive commentary.
>
>     (i) **Section 4.1 (Estimating Equations).** After the linear system $b = Aq$ is introduced, we will add one or two sentences explaining why the conditioning of $\tilde A = AD^{1/2}C\_p$ matters in practice: Intuitively, $\tilde{A}$ encodes how each unit of label-shift propagates into a change in the observed moments $\tilde{b}$.  When $\tilde{A}$ is well-conditioned, even a small moment discrepancy $\tilde{b}$ can be reliably "decoded" back into the underlying shift $\tilde{q}$; conversely, a small $\sigma\_{\min}(\tilde{A})$ means the system is nearly singular and estimation errors in $\hat{b}$ and $\hat{A}$ are heavily amplified.
>
>     (ii) **Section 4.2 (LDA Optimality).** Before stating Theorem 4.1, we will add a short paragraph to explain the geometric intuition: LDA finds the directions that maximally separate class conditional means relative to within-class spread, and Theorem 4.1 shows that these are precisely the directions that maximize $\sigma\_{\min}(\tilde A)$, i.e., they make the linear system for $q$ as well-conditioned as possible.
>
>     We believe that these changes will substantially improve the accessibility of Section 4 for a broad ICML audience while preserving the full mathematical rigor of our results.
>
> > Q3: How does RDALS behave in an open-set label shift scenario?
>
> Thank you for raising this interesting scenario. Open-set label shift, where the target domain contains classes absent from the source domain, falls outside the scope of the present work. Our framework, like the broader label shift literature (e.g., BBSE, RLLS), assumes a closed-set setting in which the source and target label spaces are identical. Handling unseen target classes is fundamentally an out-of-distribution (OOD) detection problem. We consider this an important direction for future work and will add a brief remark to the limitations section to clarify this scope boundary.
>
> >Limitations: Latent Conditional Invariance assumption breaks down entirely.
>
> We completely agree that our method will inevitably degrade when severe covariate shift co-occurs and fundamentally breaks the LCI assumption of the pre-trained features. We will explicitly discuss these boundary conditions and the limitations of our framework under such extreme, unrecoverable shifts.

---

> > ### Author Rebuttal · Reviewer_vDWY · 2026-04-05
> >
> > My concerns have been adequately addressed.

---

> > > ### Author Response · Authors · 2026-04-05
> > >
> > > Thank you again for your thoughtful engagement and careful review. We are very pleased to learn that our responses have fully resolved your concerns. In light of this, we would sincerely appreciate it if you could consider raising your score to reflect this. We sincerely appreciate the time and effort you have devoted to evaluating our work.

---

### Official Review · Reviewer_ASZn · 2026-03-13

**Soundness:** 3
**Presentation:** 3
**Significance:** 3
**Originality:** 3
**Overall Recommendation:** 4
**Confidence:** 4

**Summary:**

This paper addresses the label shift problem, where the marginal label distribution changes between source and target domains while class-conditional distributions remain invariant. The authors propose RDALS, a framework that operates in the latent space of pre-trained backbones rather than raw input space. The method applies Linear Discriminant Analysis (LDA) to project features into a discriminative subspace, constructs a moment-matching linear system, and solves for the target label distribution via a regularized SOCP formulation with simplex constraints. The paper provides theoretical results including LDA optimality for maximizing system stability, finite-sample error bounds, and generalization bounds. Experiments under Dirichlet and Tweak-One shifts show improvements over baselines.

**Compliance With Llm Reviewing Policy:**

Affirmed.

**Ethical Review Concerns:**

I do not identify any ethical concerns that require an additional ethics review.

**Final Justification:**

The detailed clarifications on assumptions, scope, and sample-splitting have solidified the work's technical soundness. Consequently, these responses have strengthened my inclination to recommend acceptance.

**Key Questions For Authors:**

see weakness section

**Limitations:**

The manuscript does not include a section that explicitly discusses the limitations of the proposed methodology. Including such a section would improve the overall completeness of the work.

**Strengths And Weaknesses:**

**Strengths**
- **Clean and principled formulation**
The pipeline (pre-trained features → LDA projection → moment matching → SOCP solver) is well-motivated at each step. Theorem 4.1 provides a clear justification for the choice of summary functions over arbitrary alternatives.
- **Comprehensive theoretical analysis**
The paper provides a complete chain of theoretical results: LDA optimality (Theorem 4.1), finite-sample concentration bounds (Theorem 4.4, Corollary 4.5), and a generalization bound (Theorem 4.6). The proof roadmap in Figure 6 is helpful for navigating.
- **Calibration-free design**
RDALS does not require calibration of a source classifier, which is a meaningful practical advantage. Figure 3 effectively illustrates the sensitivity of baselines to calibration strategy choice.
***
**Weaknesses**
- **Assumption 2.2 (Latent Conditional Invariance) is assumed, not verified.**
The entire method hinges on P(z|y) = Q(z|y) holding in the latent space, yet this is simply asserted for pre-trained backbones with only a t-SNE visualization (Figure 1) as informal evidence. The CIFAR-10-C experiment (Table 14) is a step toward validation, but CIFAR-10-C corruptions are relatively mild. It remains unclear how the method behaves when latent invariance is substantially violated. The paper would benefit from quantitative measurement of how much the latent conditional invariance is violated and how RDALS degrades as a function of this violation.
- **Limited scope of experimental benchmarks**
MNIST, CIFAR-10, and CIFAR-100 are all synthetic label shift settings constructed by resampling from the same underlying dataset. There are no experiments on naturally occurring label shift scenarios. Given that the paper motivates itself with real-world applications, this gap weakens the empirical contribution.
- **Sample splitting in theory but not in practice**
The theoretical analysis relies on sample splitting, but the actual implementation uses the full source data for both. While the authors acknowledge this discrepancy, it means the finite-sample bounds do not formally apply to the algorithm as implemented. This is a nontrivial gap between theory and practice.
- **Minor writing issues**
L62, L74) guranties -> guarantees
L375) marco-f1 -> macro-f1

---

> ### Author Rebuttal · Authors · 2026-03-30
>
> We thank the reviewer for these very insightful questions.
>
> >Q1: Assumption 2.2 (Latent Conditional Invariance) is assumed, not verified.
>
> Assumption 2.2 is crucial for constructing the estimating equations (1). Suppose the LCI assumption is violated, i.e., for each latent variable $z \in \mathcal{Z}$ and label $y \in \mathcal{Y}$, we have $P(z|y) \neq Q(z|y)$, which leads to $Q(x|y) \neq P(x|y)$. By adding and substrating,
>
> $$\\mathbb{E}_Q\\left[h_m(Z) \\mid Y=j\\right]=\\mathbb{E}_P\\left[h_m(Z) \\mid Y=j\\right]+\\left(\\mathbb{E}_Q-\\mathbb{E}_P\\right)\\left[h_m(Z) \\mid Y=j\\right].$$
>
> Then we have the estimating equation $Aq = b_{noisy} := b - \varepsilon,$ where $\varepsilon := (\varepsilon_1,\ldots,\varepsilon_{K-1})^{\top}$ with its $m$-th element to be $\mathbb{E}_Q\left[h_m(Z)\left(1-\frac{P(Z \mid Y)}{Q(Z \mid Y)}\right)\right]$ for $m = 1, \ldots, K - 1$. Inspired by this, we proposed the following noise injection experiment.
>
> We provided an additional experiment where the LCI assumption is violated. The detailed results can be found at  <https://anonymous.4open.science/r/RDALS_rebuttal-66A5/noise_r.pdf> . Specifically, we injected zero-mean Gaussian noise into the extracted target features, defined as $Z^{\prime}\_{\text{tgt}} = Z_{\text{tgt}} + \epsilon_i$ , where $\epsilon_i \sim \mathcal{N}(0, \Sigma_{noise})$. We set $\Sigma_{noise}$ as a diagonal matrix with elements $(r \cdot \sigma_j)^2$, where $\sigma_j$ is the empirical standard deviation of the $j$-th feature dimension, and $r \in [0, 1]$ controls the noise scale. The experimental results show that our estimation error (MSE) barely increases as the noise ratio $r$ increases. The mathematical mechanism behind this robustness is our first-order moment matching. RDALS solves for the label distribution via $\hat{A}\hat{q} = \hat{b}$, where the target moment is $\hat{b} = \frac{1}{n_T}\sum_{i=1}^{n_T} W^\top z_i$. With the injected noise, the perturbed empirical moment becomes:
> $$\\hat{b}\_{noisy} = \\frac{1}{n\_T}\\sum\_{i=1}^{n\_T} W^\\top(z\_i + \\epsilon\_i) = \\hat{b}\_{clean} + W^\\top \\left(\\frac{1}{n\_T}\\sum\_{i=1}^{n\_T} \\epsilon\_i\\right).$$
> By the LLN, the empirical mean of the zero-mean noise converges to zero. Consequently, $\hat{b}\_{noisy} \approx \hat{b}\_{clean}$, leaving the estimated $\hat{q}$ unaffected.
>
> >Q2: Limited scope of experimental benchmarks
>
> We have supplemented experiments on the real-world PACS dataset. It contains images across four distinct stylistic domains: Photo, Art painting, Cartoon, and Sketch. We randomly select one domain to serve as the source and another as the target, and result table at <https://anonymous.4open.science/r/RDALS_rebuttal-66A5/PACS_dataset.pdf> show our method maintains strong performance. Regarding potential numerical challenges in high-dimensional feature spaces, the main issue is that the empirical within-class scatter matrix may become singular or ill-conditioned. To address this safely, we simply apply a shrinkage regularization by adding a small $\epsilon I$ before performing LDA. This perfectly ensures numerical stability without adding huge computational costs.
>
> >Q3: Sample splitting in theory but not in practice
>
> The sample splitting in our analysis is a standard mathematical device to establish conservative finite-sample bounds by ensuring statistical independence. However, as noted in Appendix B.2, utilizing the full source dataset in practice acts as a natural plug-in estimator to maximize data efficiency. It is worth noting that this gap between theoretical splitting and practical full-data implementation is a standard convention adopted by the existing label shift literature. We will clarify this relationship in the camera ready version to better bridge our conservative theoretical bounds with standard practical implementations.
>
> >Q4: Minor writing issues
>
> Thank you for catching these typographical errors. We have corrected "guaranties" to "guarantees" and "marco-f1" to "macro-f1" in the camera ready version.
>
>
> >Limitations: The manuscript does not include a section that explicitly discusses the limitations of the proposed methodology. Including such a section would improve the overall completeness of the work.
>
> We thank the reviewer for the constructive feedback and agree that discussing limitations improves the overall completeness of our work. In the final camera-ready version, we will explicitly discuss the limitations of our methodology, particularly its dependence on representative target samples and the current scope of evaluated datasets.

---

> > ### Author Rebuttal · Reviewer_ASZn · 2026-04-03
> >
> > I appreciate the authors' clarifications on their assumptions and the comprehensive data provided for the experimental scope and sample splitting. My concerns have been fully resolved, and I will maintain my positive evaluation of this work.

---

> > > ### Author Response · Authors · 2026-04-03
> > >
> > > We sincerely appreciate your thorough review and constructive comments. We are pleased that our rebuttal has adequately addressed all of your previous concerns. Given that these key issues have been fully resolved, we would sincerely appreciate it if you could consider raising your score to reflect this.  Thank you for your time and expertise.

---

### Official Review · Reviewer_sUpn · 2026-03-15

**Soundness:** 3
**Presentation:** 3
**Significance:** 3
**Originality:** 3
**Overall Recommendation:** 4
**Confidence:** 1

**Summary:**

This paper studies the problem of learning under label shift, where the marginal label distribution changes between source and target domains while the class-conditional distributions remain invariant. The authors propose RDALS, a framework that aligns source and target distributions in the learned representation space rather than the raw input space. The method constructs a moment-matching linear system based on LDA to estimate importance weights and perform discriminative alignment between domains. The authors also provide theoretical analysis including finite-sample error bounds for importance weight estimation and generalization bounds for the adapted classifier. Experiments on several benchmark datasets demonstrate improved robustness and accuracy compared with existing label-shift adaptation methods.

**Compliance With Llm Reviewing Policy:**

Affirmed.

**Key Questions For Authors:**

1. The paper claims that the proposed alignment constructs a discriminative space that better preserves conditional invariance than methods operating in raw input space. Could the authors provide empirical evidence supporting this claim, such as visualization, invariance metrics, or controlled comparisons?

2. The ablation studies are relatively limited and appear to be conducted on a single dataset with the MSE metric. Could the authors provide additional ablation experiments across multiple datasets and include classification accuracy or other task-relevant metrics?

3. The current experiments focus on relatively small datasets (e.g., MNIST and CIFAR). How does the method scale to larger and more realistic benchmarks? Are there any computational or numerical challenges when applying the method to high-dimensional feature spaces?

4. The method relies on unlabeled target data to estimate the linear matching system. How sensitive is the approach to cases where the available target data does not accurately represent the true target distribution?

5. Could the authors clarify the conditions under which the theoretical guarantees hold and whether these assumptions are satisfied in the experimental settings?

**Limitations:**

Partially. The paper discusses some assumptions related to the label shift setting and the availability of unlabeled target data. However, the limitations could be discussed more explicitly. In particular, the dependence on representative target samples and the evaluation on relatively small datasets should be acknowledged as potential limitations.

**Strengths And Weaknesses:**

Strength：

- The paper proposes aligning source and target distributions in the learned representation space rather than in the raw input space, which is a reasonable perspective for label shift adaptation when deep features capture more stable structure across domains.
- The proposed framework constructs a moment-matching linear system based on LDA to estimate importance weights and perform discriminative alignment. The formulation is clearly described.
- The paper provides theoretical analysis including finite-sample error bounds for importance weight estimation and generalization bounds for the adapted classifier, which helps support the technical validity of the proposed approach.
- Experiments are conducted under several label shift scenarios and show improved robustness and accuracy compared with existing baselines.

Weaknesses:
- The paper claims that the proposed alignment constructs a discriminative space that better preserves conditional invariance compared with methods operating in raw space. However, this claim is not directly supported by specific experiments or analyses.
- The ablation experiments are relatively limited. Some ablations are conducted only on a single dataset and rely mainly on the MSE metric. Additional ablations across multiple datasets and using task-level metrics such as classification accuracy would strengthen the empirical evidence.
- Experiments are conducted mainly on datasets such as MNIST, CIFAR-10, and CIFAR-100. It remains unclear how the method scales to larger and more complex datasets.
- The method relies on unlabeled target samples to estimate the linear matching system. In practice, if the available target data does not accurately represent the target distribution, the estimation may become unreliable.
- The paper would benefit from a clearer discussion of potential failure modes or situations where the proposed method may not perform well.

---

> ### Author Rebuttal · Authors · 2026-03-30
>
> We thank the reviewer for insightful questions.
>
> > Q1:Could the authors provide empirical evidence supporting the claim that the latent space better preserves conditional invariance, such as visualizations or controlled comparisons?
>
> We have provided visual and empirical evidence in the paper to support this claim. Specifically, Figure 1 provides a t-SNE visualization comparing CIFAR-10 and CIFAR-10-C. It shows that the raw input space breaks the label shift assumption, whereas our pre-trained latent space successfully preserves class alignment, which supports our LCI assumption. Furthermore, Table 14 in appendix presents quantitative comparisons on the CIFAR-10-C dataset. The results show that our method  achieves the good MSE and downstream classification accuracy. This clearly shows that aligning distributions in the latent space is more robust against input perturbations than operating in the raw input space.
>
> >Q2: Could the authors provide additional ablation experiments across multiple datasets and include classification accuracy and other task-relevant metrics?
>
> We add more ablation experiments on MNIST and CIFAR-100 at <https://anonymous.4open.science/r/RDALS_rebuttal-66A5/ablation.pdf>. Regarding the choice of evaluation metric, we focus on the MSE of the $\hat{w}$ in our ablation studies because the downstream task training protocol is identical across all methods. Once $\hat{w}$ is obtained, every method trains the downstream classifier using the same importance-weighted empirical risk minimization loss. Therefore, isolating the MSE of $\hat{w}$ provides the most direct and fair measure of each component's effectiveness.
>
> >Q3: How does the method scale to larger and more realistic benchmarks? Are there any computational or numerical challenges when applying the method to high-dimensional feature spaces?
>
> We conducted the new experiments on the real-world PACS dataset. It contains images across four distinct stylistic domains: Photo, Art painting, Cartoon, and Sketch. We randomly select one domain to serve as the source and another as the target, and the result table at <https://anonymous.4open.science/r/RDALS_rebuttal-66A5/PACS_dataset.pdf> shows our method maintains strong performance. Regarding potential numerical challenges in high-dimensional feature spaces, the main issue is that the empirical within-class scatter matrix may become singular or ill-conditioned. We simply apply a shrinkage regularization by adding a small $\epsilon I$ before performing the LDA. This perfectly ensures the numerical stability without adding huge computational costs.
>
>
> >Q4: How sensitive is the approach to cases where the available target data does not accurately represent the true target distribution?
>
> To begin with, our method does not require the unlabeled target data to perfectly represent the target distribution. The only quantity estimated from the target data is the moment vector $\hat{b}\_m = \frac{1}{n_T}\sum_{i=1}^{n_T} \hat{h}_m(Z_i)$, which is a scalar summary statistic rather than a full density estimate. By the LLN, $\hat{b}$ is an unbiased estimator of $b = \mathbb{E}_Q[h(Z)]$ as long as the target samples are drawn i.i.d. from $Q$, regardless of how complex or high-dimensional $Q$ itself is.
>
> A scenario in which the method could be adversely affected when $\hat{b}$ is a biased estimate of $b$, and the estimation error $\|\hat{b} - b\|_2$ would not disappear with $n_T$. Handling such distributional mismatch within the target set is an interesting direction for future work, but is outside the scope of the current paper, which focuses on the standard label-shift setting where target samples are i.i.d. from $Q$. We will add a remark clarifying this point in the camera-ready manuscript.
>
> >Q5: Could the authors clarify the conditions under which theoretical guarantees hold and whether assumptions are satisfied in the experimental settings?
>
> Both theoretical assumptions naturally hold in our experiments. For Assumption 4.2 (Boundedness), pre-trained models use normalization layers that natively bound features, and our shrinkage regularization ($\hat{S}\_W + \epsilon I$) explicitly guarantees $\lambda_{min} > 0$. For Assumption 4.3 (Eigengap), the expressive pre-trained features ensure the $K$ distinct class centers are highly separable and linearly independent, naturally spanning a $(K-1)$-dimensional subspace with strictly positive top $K-1$ eigenvalues.
>
> >Limitations: Could the limitations be discussed more explicitly, particularly the dependence on representative target samples and the evaluation on relatively small datasets?
>
> We have strengthened our empirical study to address these concerns. First, we added experiments on PACS dataset in Q3. Second, we conducted a new experiment across varying sample sizes at <https://anonymous.4open.science/r/RDALS_rebuttal-66A5/sample_size.pdf>. The results show that RDALS consistently outperforms baselines and exhibits exceptional stability.

---

> > ### Author Rebuttal · Reviewer_sUpn · 2026-04-06
> >
> > My concerns have been well addressed.

---

> > > ### Author Response · Authors · 2026-04-06
> > >
> > > Thank you for your time and effort in evaluating our work. We are very pleased to see that the rebuttal has addressed your concerns. Given that these issues have been fully resolved, we would sincerely appreciate it if you could consider raising your score to reflect this.  Thank you once again for your thoughtful review and continued engagement with our paper.

---

### Decision · Program_Chairs · 2026-04-30

**Decision:**

Accept (regular)

**Comment:**

This paper now receives scores of 4, 4, 4, and 4, with an average of 4. And all the reviewers acknowledge that their concerns are fully resolved. Therefore, I am happy to give an Accept recommendation for this paper.